

# High resolution *in situ* measurements of phytoplankton photosynthesis and abundance in the Dutch North Sea

Hedy M. Aardema[1], Machteld Rijkeboer[1], Alain Lefebvre[2], Arnold Veen[1], and Jacco C. Kromkamp[3]

[1]Laboratory for Hydrobiological Analysis, Rijkswaterstaat (RWS), Zuiderwagenplein 2, 8224 AD Lelystad, The Netherlands
[2]Ifremer, Laboratoire Environnement et Ressources, BP 699, 62321 Boulogne sur Mer, France
[3]Department of Estuarine and Delta Systems, NIOZ Royal Netherlands Institute for Sea Research and Utrecht University, P.O. box 140, 4400 AC Yerseke, The Netherlands

*Correspondence to*: Hedy M. Aardema (hedy.aardema@mpic.de),
Present address: Department of Climate Geochemistry, Max Planck Institute for Chemistry, Hahn-Meitner-Weg 1, 55128 Mainz, Germany

**Abstract.** Marine waters can be highly heterogeneous both on a spatial and temporal scale, yet monitoring is currently
mainly limited to low resolution methods. This study explores the use of two high resolution methods to study phytoplankton dynamics; Fast Repetition Rate fluorometry (FRRf) to study phytoplankton photosynthesis and scanning flowcytometry (FCM) to study phytoplankton biomass and composition. Measurements were conducted during four cruises on the Dutch North Sea in April, May, June and August of 2017. Both FRRf and FCM data show spatial heterogeneity with monthly variation. Automated unsupervised Hidden Markov Model (uHMM) spatial clustering resulted in identification of regions
with distinct phytoplankton communities. Manual adjustments were necessary to optimize visualization of some distinct phytoplankton communities. Stepwise multiple linear regression (n=61) revealed that photophysiology (alpha), phytoplankton biomass (total red fluorescence) and abiotic predictors (Turbidity, DIN, time of the day and temperature) determined integrated water column gross primary productivity. Apart from spatial heterogeneity, the diurnal trend is a significant predictor exposing clear trends with other photophysiological parameters. Consequently, spatial patterns are
difficult as temporal and spatial patterns occur simultaneously. Nevertheless, high-resolution monitoring is a very useful supplement in addition to regular low-resolution monitoring.

**KEY WORDS:** Fast Repetition Rate fluorometry, flow cytometry, phytoplankton photosynthesis, spatial variability





## 1 Introduction

Due to high anthropogenic pressure, the North Sea has undergone considerable biogeochemical and biological changes in the past decades (Burson et al., 2016; Capuzzo et al., 2015 and 2017). Nutrient concentrations have shifted from a situation with increased input by agricultural run-off and wastewater to imbalance in the nutrient stoichiometry due to mitigation efforts (Burson et al., 2016; Philippart et al., 2000). Additionally, water clarity in large parts of the North Sea decreased during the 20th century (Capuzzo et al., 2015). These abiotic changes affect primary productivity and community composition shifts throughout the trophic levels, with large implications for ecosystem structure and fisheries production (Capuzzo et al., 2017; Burson et al., 2016). Good biological monitoring of the North Sea is required for good management. A "robust North Sea" in which ecological processes and biodiversity can thrive will ensure sustainable use, which is of major socio-economic importance; the North Sea is in close proximity to densely populated areas with high recreational value, crossed with major shipping lanes, serving as intensive fishing ground, used for sand extraction and is used for the on-going energy transition involving the creation of many offshore windmill farms.

In the future, large changes are expected due to climate change and coinciding ocean acidification, sea level rise, and increasing temperatures. Already, the North Sea is warming more rapid than most other seas (Philippart et al., 2011). These changing environmental conditions will have a big impact on marine biogeochemistry and thereby on phytoplankton community composition and primary productivity (Sarmiento et al., 2004; Behrenfeld et al., 2006; Marinov et al., 2010; Schiebel et al., 2017). Changes in phytoplankton community composition and primary productivity affect the entire ecosystem and global biogeochemical cycles (Montes-Hugo et al., 2009; Falkowski et al., 1998). Although phytoplankton community composition and productivity can be highly variable on a spatial and temporal scale, governmental monitoring still consists mainly of low-resolution measurements (Baretta-Bekker et al., 2009; Kromkamp & van Engeland, 2010; Cloern, 1996; Cloern et al., 2014; Rantajarvi et al., 1998). In spite of this, the amount of low-resolution sampling arrays has been cut back considerably since 1984 (Fig. 1; Baretta-Bekker et al., 2008).

Modern automated through-flow systems offer the opportunity to record phytoplankton composition, abundance and photosynthetic activity with high spatial and temporal resolution. This could potentially be an effective addition to current monitoring programs. These methods are not able to replace low resolution measurements such as species identification by microscope, but their higher spatial and temporal resolution and potentially shorter analysis time make it easier to identify short-lived events and can serve as early warning system. Additionally, they are able to give extra information on photophysiology, which can improve understanding of ecosystem dynamics. Two non-invasive, high-resolution instruments that can be used in marine ecosystem monitoring programs are scanning flowcytometers (FCM) for information on phytoplankton abundance and community composition and Fast Repetition Rate fluorometers (FRRf) to give information on phytoplankton photophysiology. Scanning flowcytometry is a method for counting and pulse-shape recording of



phytoplankton cells resulting in a high number of parameters on size, fluorescence and scattering properties per algal cell. These characteristics allow for division into groups based on pigment characteristics and size classes (Thyssen et al., 2015). Ideally these groups reflect functional groups, such as calcifiers, silicifiers, DMS producers (such as *Phaeocystis*) and nitrogen fixers to aid in better functional understanding of ecosystem dynamics (le Quéré et al., 2005). Because the

interaction of phytoplankton with their environment is always a sum of the community composition and their physiology, inclusion of phytoplankton physiology can improve understanding and interpretation of ecosystem dynamics. For instance, if waters become more turbid, phytoplankton can acclimate by increasing their effective absorption cross section, but it could also lead to a shift in community composition toward species with higher light use efficiency (Moore et al., 2006). The FRRf uses active fluorescence to gain insight into phytoplankton photophysiology. This technique is an alternative to the

traditional production-light curves (PE-curves) by measuring the electron transport rate (or gross photosynthesis) at increasing ambient light levels (Suggett et al., 2009; Silsbe et al., 2012). Electron transport rate per unit volume is estimated by a series of single turnover light flashes that cumulatively close all photosystems (Kromkamp et al., 2003; Suggett et al., 2003). This single turnover technique allows for calculation of the effective absorption cross-section and, in combination with an instrument specific calibration coefficient, the absorption coefficient and amount of reaction centres per volume

(Kolber et al, 1998; Kromkamp et al., 2003; Oxborough et al., 2012; Silsbe et al., 2015). Electron transport rate per unit volume is used to estimate gross primary productivity (Kromkamp et al., 2008; Smyth et al., 2004; Suggett et al., 2009).

In this study, we provide an example of how high-resolution methods can serve in biological monitoring. During four cruises in different seasons, aiming to find four different stages in the seasonal phytoplankton growth period, continuous

measurements were conducted with an FCM, FRRf, and Ferrybox to retrieve a wide range of data. An overview will be given on the information acquired with these measurements. Additionally, we will use a model to visualize spatial heterogeneity and identify regions based on the photophysiological characteristics and presence of five phytoplankton groups separated based on pigment ratio and size. Lastly, gross primary productivity is calculated and compared to high resolution data and low resolution data to get a better understanding on the key drivers for gross primary productivity in the Dutch

North Sea.





## 2 Methods

### 2.1 Study site and sampling

The Dutch North Sea is a shallow tidal shelf sea in the southern part of the North Sea. Atlantic water enters the North Sea in the south from the Channel but the majority of the Atlantic curves around Scotland and flows southwards and eastwards,

where it meets the Channel water and the freshwater from the rivers forming the Frisian Front. For a detailed description on the North Sea physical oceanography see Sundermann and Pohlman (2011). Along the Dutch coast, high river input from especially the Rhine River decrease the salinity and loads the coastal zone with high nutrient concentrations (Burson et al., 2016). Anthropogenic pressure is high in the Dutch North Sea resulting in a history of large shifts in nutrient concentrations and water clarity (Capuzzo et al., 2015; Burson et al., 2016).

The monitoring of the Dutch North Sea is performed by the Dutch government (Rijkswaterstaat) in a monitoring program called MWTL (Monitoring Waterstaatkundige Toestand des Lands, freely translated as 'Monitoring of the status of the governmental waters of the country'). The location of the sampling stations of the program are organized along transects (Fig. 1). Notice the decrease in the number of sampling stations between 1983 (small brown dots) and 2014 (larger blue filled circles). The stations are sampled between March and October with a frequency of every two or four weeks dependent

on the transect.



**Figure 1: Sampling locations of the MWTL monitoring program in 1984 (small black dots) and 2014 (larger circles). The location of the Dutch Delta and the transect names are given (Terschelling, Noordwijk and Walcheren).**



In 2017, four 4-day sampling surveys were conducted for the JERICO-NEXT project on board the RV *Zirfaea* during their regular monitoring cruises on the Dutch North Sea. To assess the heterogeneity of the Dutch North Sea and the benefits associated with high resolution monitoring the four cruises were conducted in different months (April, May, June and August), thereby aiming to cover different seasons and stages of the phytoplankton bloom (Baretta-Bekker et al., 2009). During these cruises the high-resolution methods (FRRf, FCM and Ferrybox) were combined with lower resolution methods on nutrient concentrations and vertical light extinction using vertical deployments of a rosette frame equipped with a CTD and Niskin bottles.

On the RV *Zirfaea* the water inlet was situated approximately 3.5 m below sea surface level. From the water inlet the sample water, with a flow rate of approximately 24 litre per minute, was split towards 1) a flow-through -4H-JENA Ferrybox (-4H-JENA engineering GmbH, Germany) equipped with an FSI Excell® Thermosalinograph (Sea-Bird Scientific, USA) to measure temperature and salinity and a SCUFA™ Submersible Fluorometer (Turner Designs Inc., USA), and 2) at a flow rate of 1 L per minute towards a 230 $cm^3$ flow through sampling container where water was cleared from bubbles and sand. The time from water inlet to sampling chamber was approximately 2 minutes. A FastOcean Fast Repetition Rate fluorometer (FRRf) with Act2-based laboratory flow through system (Chelsea Technologies Group Ltd, UK) and a Cytosense scanning flowcytometer (Cytobuoy BV, the Netherlands) automatically sampled from the sampling unit every 30 minutes. Since the average speed of the ship was 8 knots, the average spatial resolution of FCM and FRRf measurements was on average 7.5 kilometres. The Ferrybox sensors stored data every minute. At discrete stations (10 to 15 per cruise) water samples were collected for nutrient analysis with a rosette sampler. Simultaneously, the rosette sampler recorded the irradiance extinction in the water column with a QSP-200L Log Quantum Scalar Irradiance Sensor (Biospherical Instruments Inc., USA). The diffuse attenuation coefficient $K_d$ ($m^{-1}$) was calculated as the linear regression of the natural logarithm of irradiance corrected for changes in surface irradiance (PAR; 400-700 nm) versus water depth.

## 2.2 Chemical analyses

Samples for nutrient analysis were filtered over Whatmann GF/F filters and kept frozen until analyses. The analyses of ammonium ($NH_4^+$), nitrite ($NO_2^-$), nitrate ($NO_3^-$), ortho-phosphate ($PO_4$) and silicate (Si) concentrations were conducted by the Rijkswaterstaat laboratory (the Netherlands) according to RWS internal analysis protocol A1.004 using a San plus Analyzer (Skalar Analytical B.V., the Netherlands). Chlorophyll *a* concentration (hereafter Chl *a*) was sampled by filtering over Whatmann GF/C filters and freezing the filter at -80 ℃. Thereafter the analysis was conducted by the MUMM laboratory (Belgium) using High-Performance Liquid Chromatography (HPLC) according to RWS analysis protocol A200. Quality control was performed by the RWS laboratory (The Netherlands).



### 2.3 High frequency methods

### 2.3.1 Variable fluorescence

Variable fluorescence was measured with a FastOcean Fast Repetition Rate fluorometer (FRRf) and Act2-based laboratory system (Chelsea Technologies Group Ltd, UK). Temperature was controlled by connecting a Lauda ecoline cooler

(LAUDA-Brinkmann, LP., USA) to the water jacket of the Act2 system.

The acquisition protocol consisted of 100 excitation flashes with a flash pitch of 2 µs and 40 relaxation flashes with a flash duration of 60 µs. Excitation flashes were performed with the blue LED (450 nm) and strength of the LEDs was automatically adjusted to the phytoplankton concentration by the manufacturer' FAstPro software. A loop of simultaneous blue and green flashes (450 nm+530nm) was performed after the acquisition loop of only blue LEDs in case the blue LEDs

were not able to reach saturation (for instance with high cyanobacteria concentrations), but as this was not the case, only the parameters measured by blue LEDs were used for further calculation. The sequence was repeated 20 times with a sequence interval of 100 ms. The sample was refreshed before each fluorescent light curve (FLC) by flushing for 60 seconds and kept well-mixed by " flushing" for 200 ms between acquisition loops.

The FLC protocol consisted of 14 light steps of 100 s, of which the light intensity was automatically adjusted to get the

optimal FLC shape based on the previous light curve. A pre-illumination step (55 seconds on 12 µmol photons m$^{-2}$ s$^{-1}$) was included before the FLC to low light acclimate the phytoplankton and to relax NPQ of diatoms and other chlorophyll *a -c* algae as they stay in the light activated state in the dark. After each light step, measurements were made in the dark for 18s to retain a value for $F_0$'. The data was corrected for the background fluorescence by taking sample blanks multiple times per day by filtration over a 0.45 µm filter and subtracting the last determined background fluorescence from the sample

fluorescence.

An overview of the derived photosynthetic parameters can be found in Table 1. To derive values for the maximum photosynthetic electron transport rate ($P_{max}$), minimum saturating irradiance ($E_k$) and the light utilisation efficiency (α) the relative electron transport rate (rETR) of the samples was fitted to the exponential model of (Webb et al. 1974), after

normalizing the data to the irradiance as described by (Silsbe and Kromkamp 2012):

$$F_q'/F_m' \ = \ \frac{P_{max}\left(1-\exp\left(\frac{\alpha}{E_k}\right)\right)}{E} \tag{1}$$

where E is the irradiance in µmol photons m$^{-2}$ s$^{-1}$, $F_q$'/$F_m$' the effective PSII quantum efficiency, α is the initial slope of the rETR vs irradiance curve and $E_k$ is the light saturation parameter (in µmol photons m$^{-2}$ s$^{-1}$). The relative maximum rate of

photosynthetic electron transport ($P_{max}$) was calculated as:

$$P_{max} \ = \ E_k \times \ \alpha \tag{2}$$





**Table 1: The derived photosynthetic parameters used in this study (see Oxborough et al. (2012) and Silsbe et al. (2015) for more information).**

| | Description | unit |
|---|---|---|
| C | Fraction of RCIIs in the open state | Dimensionless |
| $F^{(\cdot)}$ | Fluorescence at zero[th] flashlet of an ST measurement when $C > 0$ [(under ambient light)] | Dimensionless |
| $F_m^{(\cdot)}$ | Fluorescence when $C = 1$[(under ambient light)] | Dimensionless |
| $F_q^{(\cdot)}$ or $F_v$ | $\Delta F^{(\cdot)}$, variable fluorescence | Dimensionless |
| $F_q^{(\cdot)}/F_m^{(\cdot)}$ | Fluorescence parameter providing an estimate of PSII efficiency under ambient light[(under ambient light)] | Dimensionless |
| $F_v/F_m$ | Quantum efficiency of PSII | Dimensionless |
| $\sigma_{PSII}$ | Absorption cross section of PSII photochemistry | $nm^2$ $PSII^{-1}$ |
| [RCII] | Concentration of functional RCII | nmol RCII $m^{-3}$ |
| $a_{LHII}$ | Absorption coefficient of PSII light harvesting | $m^{-1}$ |
| $\alpha$ | Light utilisation efficiency | $\mu$mol electrons ($\mu$mol photons)$^{-1}$ |
| $E_k$ | Minimum saturating irradiance of fluorescence light curve | $\mu$mol photons $m^{-2}$ $s^{-1}$ |
| $P_{max}$ | Maximum photosynthetic electron transport rate | $\mu$mol electrons $m^{-2}$ $s^{-1}$ |
| $JV_{PII}$ | PSII flux per unit volume | mol electrons (PSII $m^{-3}$) $d^{-1}$ |
| GPP | Gross Primary Productivity | mg C $m^{-2}$ $h^{-1}$ |
| $n_{PSII}$ | Number of [RCII] per mole Chl *a* | mol RCII $mol^{-1}$ chla |
| $1/\tau$ | Rate of re-opening of a closed RCII with an empty $Q_B$ site | $ms^{-1}$ |
| $K_a$ | Instrument type-specific constant allowing for direct calculation of [RCII] and $JV_{PII}$ from FRR data | $m^{-1}$ |

The PSII flux in $\mu$mol electrons $m^{-3}$ $h^{-1}$ was calculated as the product of the effective PSII efficiency ($F_q'/F_m'$), the optical absorption cross section of the light harvesting pigments of PSII ($a_{LHII}$) and the irradiance (E):

$$JV_{PII} = F_q'/F_m' * a_{LHII} * E \tag{3}$$

10     where

$$F_q'/F_m' = \frac{F_m' - F'}{F_m'} \tag{4}$$

and

$$a_{LHII} \ (in \ m^{-1}) = \frac{F_0 * F_m}{F_m - F_0} * K_a \tag{5}$$





$K_a$ (m$^{-1}$) is an instrument specific factor necessary for obtaining absolutes rate of photosynthetic transport (see Oxborough et al. (2012) and Silsbe et al. (2015) for more information). The amount of reaction centres per cubic metre ([RCII]) was calculated as

$$[RCII] \ (in \ nmol \ m^{-3}) = K_a * \frac{F_0}{\sigma_{PSII}} \tag{6}$$

and the approximate number of reaction centres per Chl $a$ (note that the Chl $a$ concentration is estimated based on minimum fluorescence value ($F_0$), and is therefore a mere estimation). If [RCII] is known the number of PSII units per mole Chl $a$ ($n_{PSII}$) can be calculated:

$$n_{PSII} = \frac{[RCII]}{[Chl \ a]} \tag{7}$$

$Q_A$ reoxidation or rate of re-opening of a closed RCII was calculated as 1 divided by the time constant of re-opening of a closed RCII with an empty $Q_B$ site ($\tau_{ES}$) in ms$^{-1}$.

Gross Primary Productivity (GPP) was estimated by integrating surface productivity over water depth. Volumetric $P_{max}$ and $\alpha$ were derived by fitting JVPII in μmol photons m$^{-3}$ h$^{-1}$ to equation 1 (the exponential model of Webb et al., 1974) and these parameters used to integrate productivity over depth where the light extinction in the water column was estimated as

$$E(z) = E_{surface} * e^{-K_d*z} \tag{9}$$

with E(z) being the irradiance at depth $z$. The value for surface irradiance ($E_{surface}$) was held constant over the month and calculated as the monthly average light intensity over 2010 to 2016 measured at the roof of the NIOZ building in Yerseke using a LI-190 quantum PAR sensor. Hourly data were averaged and stored using a LI1000 datalogger. The light extinction coefficient, $K_d$, was calculated based on a vertical irradiance profile obtained with a QSP-200L Log Quantum Scalar Irradiance Sensor (Biospherical Instruments Inc., USA) which was conducted approximately 10 times per cruise (see Methods-Study Site and sampling). In order to interpolate between these profiles a correlation with the turbidity (in NTU, as measured by the Ferrybox) as predictor was determined based on linear regression: $\ln(K_d)=0.785*\ln(Turbidity)-1.324$ (n=71, $R^2$=0.77, p<0.01). The calculated water column productivity was converted to carbon units by assuming 6 moles of electrons were required to fix one mole of carbon, based on a study in the adjacent Oosterschelde and Westerschelde estuaries (Kromkamp et al., in prep.).



### 2.3.2 CytoSense scanning flowcytometry

Single cell measurements of the phytoplankton community were carried out using a bench top scanning flowcytometer (Cytobuoy BV, the Netherlands) equipped with two lasers (488 nm and 552nm). Both lasers (60mW each) were continuously focussed on the same spot in the middle of the flow-through chamber having a height of ca. 5 um and a width

of 300 um. The speed of the particles is ca. 2.2 m s$^{-1}$. With this configuration Forward light Scatter (FWS) and Sideward light Scatter (SWS) of all particles were measured. The system contained 3 fluorescence channels; FLY of 550-600 nm (Phycoerythrin); FLO of 600-650 nm (Chlorophyll *b* and Phycocyanin) and FLR >650 nm for chlorophyll *a* and *c* detection. Per cell the pulse shape recording and the parameters (FWS, SWS, FLR, FLO and FLY) plus their affiliates (length, total and maximum values) are recorded and saved. The FCM was equipped with a double set of detectors (PMT's) for each of the

three channels to increase the dynamic range (Rutten, 2015). The instrument was checked daily for drift using 3 µm Cyto-Cal$^{TM}$ 488 nm alignments beads (Thermo Fisher Scientific Inc., USA). Additionally, the FCM was equipped with an Image-in-flow camera to take pictures of the nano- and micro-phytoplankton, this allows for linking pulse shape recordings to microscopy results and thereby identification of represented phytoplankton groups in respective clusters.

Phytoplankton cells were clustered based on the pulse shape recording of the individually scanned phytoplankton. In this paper we mainly discriminate the phytoplankton groups based on their size (pico, nano and micro) and Orange/Red fluorescence ratio (hereafter O/R ratio; Table 2; Easyclus software 1.26, ThomasRuttenProjects, The Netherlands). Size was calculated based on the length FWS. Length FWS was found to be a good estimate of the length of the particles because due to the speed acceleration of the particles in the sheath fluid of the FCM the organisms will flow along their long axis. We

obtained a linear relation between Length FWS and measured length of diverse phytoplankton species, having an angle of inclination of almost 1 and R$^2$=0.99. For organisms smaller than 5 µm there may be some deviation from this relationship due to the width of the laser beam (which is 5 µm).

**Table 2: The phytoplankton groups distinguished in the current study.**

|              | Length FWS | Main corresponding taxonomic group |
|--------------|-----------|-------------------------------------|
| Pico-Red     | <4 µm*    | Pico-eukaryotes                     |
| Pico-Synecho | <4 µm*    | e.g. Synechococcus                  |
| Nano-Crypto  | 4-20 µm   | Cryptophycea                        |
| Nano-Red     | 4-20 µm   | Diatoms, Haptophytes                |
| Micro-Red    | >20 µm    | Diatoms, Haptophytes                |

*In june <6 µm



## 2.4 Data analysis

Outliers were removed after visual inspection of pairplots made with the pairplot function of the HihgstatLib.V4 script (Zuur et al., 2009) and the fitted $F_q'/F_m'$-E curves. A minimum fluorescence signal was set for calculations of photosynthetic parameters, below this blank corrected instrument-specific fluorescence signal $F_q'/F_m'$ often reached above the biologically

unlikely limit of 0.65. The datasets of the high resolution measurements (FRRf, FCM and Ferrybox) were linked using corresponding timestamps. When multiple measurements were performed within one FLC, the average was used.

Spatial clusters in the Dutch North Sea were defined using R (version 3.4.1, R Core Team, 2017) with the additional uHMM R package (Poisson-caillault and Ternynck, 2016). The package default settings normalize data before clustering, and automatically find the number of clusters based on spectral classification and the geometry of the data. This new

methodology is more robust than the classical hierarchical and k-means technics (Rousseeuw et al., 2013, 2015). Datapoints were then per cluster labelled and plotted on a map to visually identify regions. Cruises were analysed separately and not as one continuous time series as the time gaps between the sampling cruises were large. All photophysiological colinear predictors were removed (VIF>6; Zuur et al., 2009). The FCM phytoplankton groups based on total red fluorescence were included despite colinearity.

Separate stepwise multiple regressions were performed for all months combined by stepwise deletion of insignificant predictors. Predictors were tested for colinearity and all predictors with a variance inflation factor over 6 were removed (Zuur et al., 2009). Interactions of the predictors were not included. Residuals were visually checked for normality by plotting a qqnorm plot of the residuals of the model and for homogeneity of the variances by plotting the residuals of the model against the fitted values and against each separate predictor.

Principal Component Analyses (PCA) were based on correlation matrixes with scaled parameters to correct for unequal variances and was carried out with the prcomp() function in R (version 3.4.1, R Core Team, 2017). The PCA visualization was done using the supplemental R package factoextra (Kassambra and Mundt, 2017). Maps were made using QGIS v. 2.14.2 and other figures were made with ggplot2 in R (Wickham, 2009).



## 3 Results

### 3.1 Environmental conditions

Environmental conditions in the Dutch North Sea are spatially heterogeneous and strongly influenced by seasonal dynamics. Sea surface temperature increases from $9.5 \pm 1.0$ °C in April to $19.0 \pm 0.6$ °C in August (Table 3). Seasonal variations in

salinity are small with highest monthly mean salinity was measured in April ($34.1 \pm 1.8$), while spatial variability is high with river influx decreasing the salinity down to 26 in the coastal zone. The monthly average of turbidity does show seasonal variation and was clearly higher in April ($2.3 \pm 3.0$ NTU) in comparison to other months, which was reflected in the highest $K_d$ values ($0.39 \pm 0.28$ m$^{-1}$). It needs to be noted that monthly averages are not completely comparable, because of differences in sampling route and stations (Fig. 4).

Nutrient concentrations show both high spatial and seasonal variability (supplementary Table S1). The average nutrient concentrations of all stations sampled during the different cruises is shown in Table 4. To see if nutrient concentrations were potentially limiting we used threshold concentration for DIN and Si as 2 µmol L$^{-1}$ and PO$_4^{3-}$ as 0.2 µmol L$^{-1}$ (Peperzak et al, 1991, Philippart et al., 2007), although Ly et al. (2014) showed that for Wadden Sea phytoplankton phosphate can become limiting when values become lower than 0.13-0.16 µmol L$^{-1}$. The general trend in all transects is an offshore moving

gradient of decreasing nutrient concentrations. Offshore stations (>70 km offshore west of the Netherlands, >135 km North of the Netherlands) are DIN limited year-round, while regions closer to freshwater influx (station Noordwijk 2) are DIN sufficient in all sampled months except August. This clearly reflects the input of the Rhine and its waters remained relatively close to the coast; no influence seems present at waters further than 70 km offshore. In April nutrient concentrations are on average higher and only potentially limiting in the most Southerly part of the Dutch North Sea (Walcheren transect) and

further offshore (>70 km offshore west of the Netherlands, >135 km North of the Netherlands). In later months, nitrate and silicate limitations gradually moves towards the coastal zone, with nutrient limitation at all sampled stations in August. Phosphate levels were generally quite low and possibly limiting, with exception in April north of the Wadden Islands up to 135 km offshore and Noordwijk 2, a region with high freshwater influx, in April and May. Later in the year (June and August) phosphate concentrations recovered in the Southern part of the Dutch North Sea reaching up to 0.6 µM. Silicate, an

essential element for diatoms, seemed to be present in limiting concentrations at all sampled times at the Walcheren stations, suggesting that the cruise in April was already beyond the peak of the diatom bloom. Nutrient ratios (DIN:DIP, DSi:DIP and DIN:DSi) suggest P-limitation and Si-limitation (for diatoms) in April and May. In June values were closer (but below) to the Redfield Ratio, whereas in August most of the Dutch North Sea seems N-limited. Apart from the seasonal trend of the nutrient limitation on the onshore-offshore gradient moving towards the coast, there also seems to be a south to north trend,

with the southern Dutch North Sea being depleted earlier in the season in comparison to the more northerly area.



**Table 3: Monthly averages ± SD of abiotic conditions and biological parameters. Due to differences in sampling route and stations, the monthly averages are not completely comparable. Large standard deviations are due to spatial heterogeneity, for a more detailed description of the spatial heterogeneity; see figure 4 and the supplementary material. $P_{max}$ and alpha are based on relative electron transport rates.**

|  | April | May | June | August |
|---|---|---|---|---|
| **Abiotics** |  |  |  |  |
| Salinity (‰) | 34.1 ± 1.8 | 33.6 ± 1.8 | 33.6 ± 1.8 | 34.0 ± 1.4 |
| SST (°C) | 9.5 ± 1.0 | 12.1 ± 1.1 | 15.5 ± 1.8 | 19.0 ± 0.9 |
| Turbidity (NTU) | 2.3 ± 3.0 | 1.1 ± 0.8 | 1.3 ± 1.3 | 1.2 ± 1.3 |
| $PO_4$ (μM) | 0.3 ± 0.1 | 0.2 ± 0.1 | 0.3 ± 0.1 | 0.3 ± 0.1 |
| Si (μM) | 3.2 ± 3.0 | 1.8 ± 1.5 | 1.0 ± 0.6 | 1.3 ± 1.1 |
| $NH_4$ (μM) | 1.0 ± 1.1 | 1.2 ± 0.9 | 1.5 ± 1.2 | 0.4 ± 0.4 |
| $NO_3 + NO_2$ (μM) | 10.3 ± 12.5 | 3.4 ± 5.5 | 1.0 ± 1.1 | 0.1 ± 0.2 |
| DIN:DIP | 39.0 ± 52.5 | 26.9 ± 42.1 | 7.5 ± 5.3 | 2.3 ± 2.1 |
| DSi:DIP | 6.4 ± 6.5 | 9.9 ± 9.2 | 4.1 ± 2.5 | 5.7 ± 3.9 |
| $K_d$ (m$^{-1}$) | 0.39 ± 0.28 | 0.33 ± 0.12 | 0.30 ± 0.20 | 0.25 ± 0.14 |
| **Biotics** |  |  |  |  |
| Chlorophyll $a$ (μg L$^{-1}$) | 18.32 ± 19.71 | 5.67 ± 10.39 | 4.08 ± 4.11 | 3.98 ± 3.91 |
| $F_v/F_m$ | 0.52 ± 0.04 | 0.26 ± 0.09 | 0.40 ± 0.09 | 0.48 ± 0.07 |
| $\sigma_{PSII}$ (nm$^2$ PSII$^{-1}$) | 3.67 ± 0.30 | 5.92 ± 1.35 | 4.59 ± 0.88 | 5.26 ± 1.07 |
| [RCII] (*10$^{-9}$ nmol RCII m$^{-3}$) | 32.4 ± 21.8 | 5.82 ± 10.4 | 4.31 ± 2.72 | 2.21 ± 1.84 |
| $n_{PSII}$ (*10$^{-4}$ RCII (Chl $a$)$^{-1}$) | 8.00 ± 0.58 | 5.30 ± 1.75 | 6.57 ± 1.67 | 5.95 ± 1.15 |
| $1/\tau$ (ms$^{-1}$) | 0.24 ± 0.06 | 0.52 ± 0.10 | 0.49 ± 0.07 | 0.62 ± 0.12 |
| α | 0.53 ± 0.03 | 0.25 ± 0.09 | 0.39 ± 0.08 | 0.48 ± 0.08 |
| $E_k$ | 300 ± 52.5 | 223 ± 147 | 253 ± 124 | 277 ± 137 |
| $P_{max}$ | 158 ± 30 | 56.5 ± 42.4 | 97.5 ± 47.7 | 130 ± 60.4 |
| GPP water column (mg C m$^{-2}$ h$^{-1}$) | 781 ± 409 | 207 ± 277 | 136 ± 101 | 68.4 ± 39.1 |
| GPP surface (μg C L$^{-1}$ h$^{-1}$) | 115.7 ± 58 | 27.5 ± 72 | 16.5 ± 13 | 8.7 ± 8.3 |
| O:R ratio | 0.31 ± 0.52 | 0.07 ± 0.09 | 0.20 ± 0.20 | 0.27 ± 0.16 |
| Rel. abundance microphytoplankton (%) | 0.6 ± 0.6 | 3.5 ± 4.4 | 2.0 ± 2.0 | 0.4 ± 0.6 |
| Rel. abundance Nanophytoplankton (%) | 27.6 ± 17.2 | 41.3 ± 17.8 | 21.6 ± 8.0 | 18.1 ± 8.0 |
| Rel. abundance Picophytoplankton (%) | 71.8 ± 17.5 | 55.2 ± 19.0 | 76.3 ± 9.2 | 81.5 ± 6.0 |



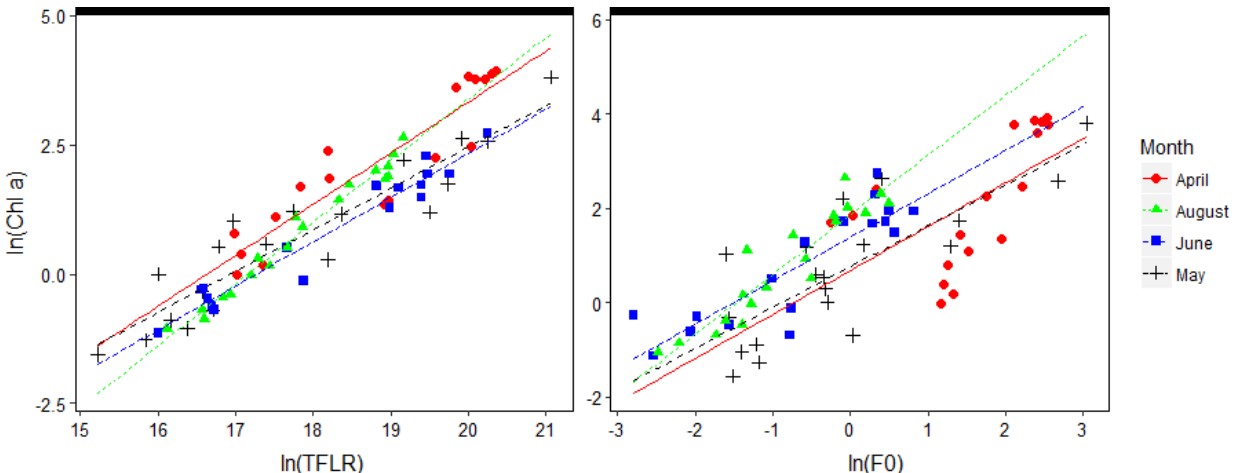

**Figure 2: linear regression of the natural logarithms of Chl *a* concentration in μg L$^{-1}$ as determined by HPLC (y-axis) and on the x-axis, FCM-derived total red fluorescence (TFLR, left panel) and FRRf-derived minimum fluorescence (F$_0$, right panel).**

## 3.2 Phytoplankton parameters

Information on total phytopankton abundance can be obtained from both FRRf and FCM (Fig. 2). The FCM provides data on abundance in the amount of cells per millilitre of seawater and gives an estimate of Chl *a* concentration based on the cumulative red fluorescence of all cells (hereafter TFLR). The FRRf also provides an estimate of Chl *a* based on the minimum fluorescence (F$_0$). Using cell count or fluorescence as predictor of phytoplankton presence yields contrasting results (Fig. 3), because of the wide range of phytoplankton cell sizes; microplankton have a substantially higher biomass, and thus fluorescence, per cell in comparison to picoplankton. So while the phytoplankton cell count is higher in June and August in comparison with April, the community in the former months consists of mainly pico-plankton which contribute little to total fluorescence resulting in a considerably lower fluorescence in June and August (Fig. 3). Fluorescence is therefore a better predictor of biomass or of Chl *a* concentration than cell counts, although Chl *a* concentration is a limited predictor of biomass because the Chl *a* concentration per cell is species-specific and is subject to phenotypic acclimation to abiotic conditions (Flynn, 1991, 2005; Geider et al., 1997).

Both the FRRf and FCM provide significant predictors of Chl a concentration (Fig. 2). When performing an ANCOVA with month as factorial predictor, natural logarithm transformations were necessary because of the highly unequal variances between months. The ANCOVA with the FRRf-derived F$_0$ as Chl *a* predictor revealed that Chl *a* concentrations significantly differed per month (p<0.01) but not the slope, and that F$_0$ was a significant predictor (p<0.01) of Chl *a* concentration (adjusted R$^2$=0.66). Yet, the FCM estimate of Chl *a* concentration (TFLR) was a better predictor (p<0.01) with an adjusted R$^2$ of 0.90. The ANCOVA with the FCM-derived TFLR as Chl *a* predictor resulted not only in a significant difference of the Chl *a* concentration per month (p<0.01) but also in a significantly different slope (p<0.05), suggesting that abiotic factors





and phytoplankton community composition are influencing the amount of fluorescence per Chl *a* molecule (Fig. 2). In April and August the slope is steeper in comparison to May and June. An explanation could be the package effect (Dubinsky et al., 1986), where stacking of Chl *a* at low light intensities causes a shading effect within the cells and a steeper slope of Chl *a* concentration with *in vivo* fluorescence. Because there is a lack of agreement in photophysiological parameters between

April and August, it is likely that the months do not have the same drivers for the steeper slope. In April the high $\sigma_{PSII}$ coincides with high $n_{PSII}$, suggesting that although the chlorophyll self-shades it does not result in a lower absorption cross section because of the high amount of RCIIs in relation to Chl *a* molecules. In contrast, in August the phytoplankton community is nutrient limited and has a higher $\sigma_{PSII}$, in correspondence with results obtained by Kolber et al. (1988) who observed that nutrient limitation increases $\sigma_{PSII}$. Self-shading in this month is more likely a result of smaller cell size as

indicated by the higher abundance of picoplankton (Table 3; Geider et al., 1986).

The FRRf yields other biomass related proxies next to the minimum fluorescence; the total absorption coefficient in the water ($a_{LHII}$ in m$^{-1}$) based on the absorption of the photosynthetic pigments pigments associated with PSII and the amount of PSII reaction centres per volume ([RCII] in nmol RCII m$^{-3}$). Both are very strongly correlated to $F_0$, although the ratio of

RCII to $a_{LHII}$ can vary by nature, affecting $n_{PSII}$ (Supplementary material). The minimum fluorescence measured with the FRRf ($F_0$) is related to the red fluorescence mearured with the FCM (TFLR; r=0.7). Interestingly, TFLR and $F_0$ are not correlated to total orange fluorescence, which indicates that the cyanobacterial picoplankton is not a fixed proportion of the total phytoplankton biomass (Supplementary material). Photosynthetic parameters are sometimes highly correlated (Supplementary material). The correlation of alpha and $F_v/F_m$, indicators for photosynthetic affinitiy and photosynthetic

efficiency, are perfectly correlated (r=1). The parameters derived from the PE-curve, $P_{max}$ and $E_k$, show high correlation. But surprisingly, α does not show any correlation with $E_k$. This suggests that the light affinity is not dependent on the level of irradiance where the PSII reaction centres become saturated, or that its value is obscured by nutrient limitation. As expected $\sigma_{PSII}$, is very strongly negatively related to $n_{PSII}$ (r=-0.9); the larger $n_{PSII}$, the smaller the number of pigment molecules associated with it.

Community composition is variable over the months with the biggest shift in community composition between May and June (Fig. 3). In May mean $F_v/F_m$ is low (0.26 ± 0.09), suggesting physiological stress (Table 3). This coincides with a shift of nutrient sufficiency in the largest part of the sample region (with only potentially limiting conditions in the most southerly part of the Dutch North Sea and further offshore) in April to a larger region of nutrient limitation in May. In June the mean

$F_v/F_m$ recovers and community composition changes. Both groups of picophytoplankton (*Synechococcus* and total) increase in relative abundance between May and June, while the nanophytoplankton shows a strong decrease (Fig. 3). Because the picophytoplankton fraction makes up for only a small part of the biomass, the microphytoplankton is the largest contributor to red fluorescence in June, although this group does not increase in relative abundance in comparison to May. After June the microplankton disappears, leaving 80% of the average community composition to picoplankton.





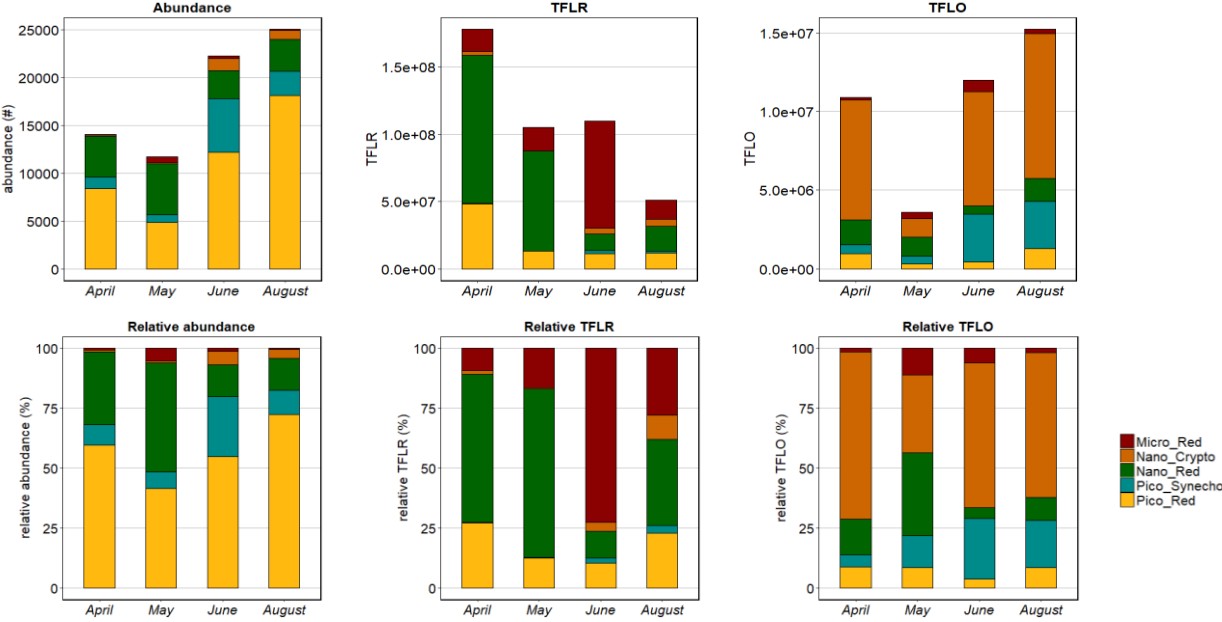

**Figure 3: Phytoplankton abundance per phytoplankton group distinghuished with the flowcytometer, shown as average (relative) abundance per month (left), total red fluorescence(middle) and total orange fluorescence (TFLO; right).The upper graphs are absolute and lower graphs relative.**

### 3.3 Spatial distributions

Both the biomass concentration and the phytoplankton community composition, expressed as percentage of the total cell numbers, showed a dynamic picture (Fig. 4). In all cases, microphytoplankton < 10% of the total cell counts, although in terms of biomass they sometimes dominate (Fig. 3).

10    In April, high biomass concentrations (using TFLR) are observed close to the Dutch Delta in the South of the Dutch EEZ and west to the island of Texel and Vlieland. Very low concentrations are found offshore, especially in the more central part towards the Doggersbank area. The north-western wedge of the Dutch North Sea was dominated by picoplankton whereas the southern part and the north coastal area of the Dutch EEZ were numerically dominated by nanophytoplankton. Orange fluorescent dominating species (like *Synechococcus* and Cryptophycea) were a small proportion of the total phytoplankton

15    community in April (generally less than 10%), however about 100km North of the island of Vlieland there was a patch of water where they seem to dominate the phytoplankton (Fig. 4e). Microphytoplankton abundance < 3%, and highest numbers were found close to the Dutch Delta and along the Noordwijk transect. In April, the lowest $F_v/F_m$ values (0.4-0.5) were found in the southern part of the Dutch EEZ. Highest values were found offshore on and towards the Terschelling transect, and at the coastal stations of the Noordwijk transect.




The situation in May is different from April (Fig. 4, second column). The higher biomass near the Texel and Delta area has mainly dissappeared, and is much more homogeously distributed while the community composition is very heterogeneous. The biomass concentration is about 50% lower than in April (Table 2, Fig. 3). Unfortunately, the RV *Zirfaea* did not sail to

the NW edge of the Dutch coastal zone (Dogger Bank), but the higher dominance of Orange fluorescent dominated species on the Terschelling transect observed in April is still visible in May (Fig. 4f). Along the Walcheren transect and at a section of the Terschelling transect (~60-135 km off the coast) the highest percentages of picophytoplankton were observed (60-80%), whereas the highest percentage of nanophytoplankton was observed north of Terschelling 100 and closer to the Frisian coast. Notice that this coast was visited twice in 2-days, and that abundance varied between those occasions (40-60% vs. 60-

80%), yet the difference between visits was only 6 %; the first time around 64 % and the second time around 58 %. The two visits did fall into different diurnal time periods, the first time was at the end of the day (around 18:00h) while the second time was early morning (around 7:00h), but both of these time periods were in the flood tide. In May, $F_v/F_m$ was in general much lower than in April (0.1-0.3) across most of the Dutch EEZ. Higher values were found in the southern coastal stations, a possible consequence of the outflow of the Scheldt River, and 70 km offshore Noordwijk. At both of these regions low

values for $F_v/F_m$ were found in April so possibly these phytoplankton communities have already acclimated to low nutrient conditions. The range in $\sigma_{PSII}$ was larger in May in comparison to April. A small area near the coast of Noordwijk showed low $\sigma_{PSII}$ values. This might reflect Rhine River waters, but the effect was not noticable further north.

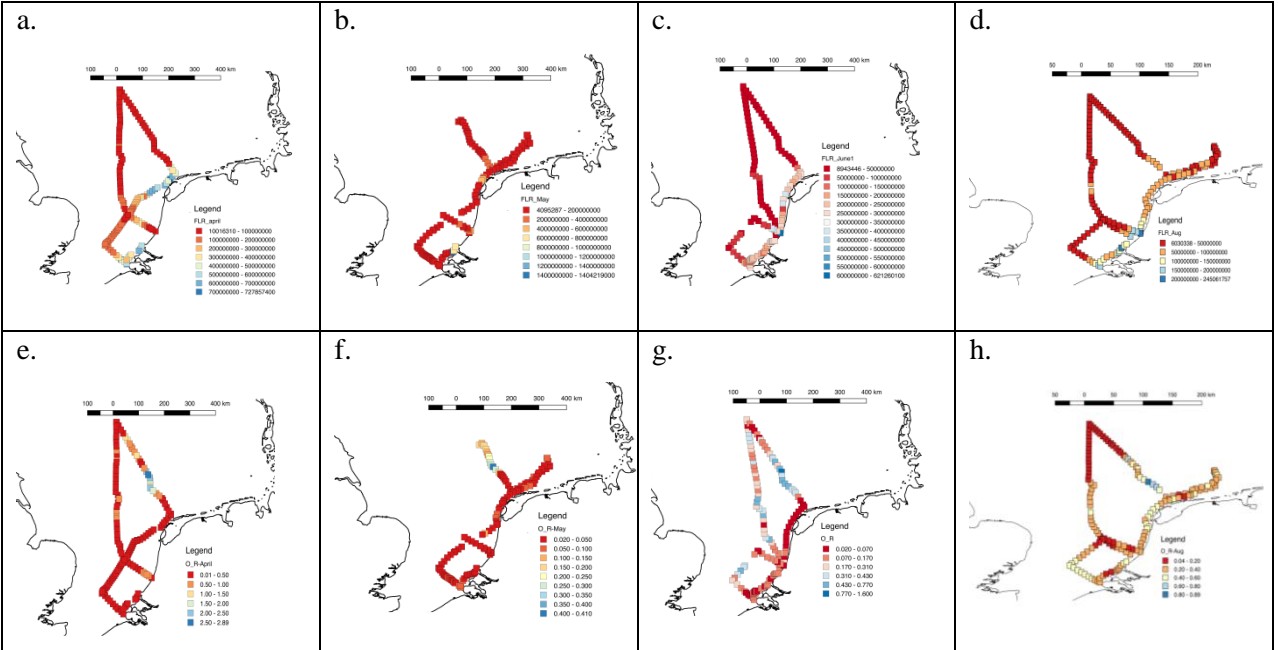





**Figure 4: Phytoplankton biomass using FCM-derived total red fluorescence (first row; a-d), O/R ratio (second row, e-h), percentage pico (third row, i-l) and percentage nano (fourth row, m-p), $F_v/F_m$ (fifth row,q-t) and $\sigma_{PSII}$ (sixth row, u-x) in April 2017 (left panels), May 2017 (middle left panels), June 2017 (middle right panels) and August 2017 (right panels). Please note the different scaling, which was necessary to optimally visualize spatial heterogeneity.**



June showed an increase in phytoplankton abundance, although the community changed toward less nanoplankton (Fig. 3). Highest biomass concentrations were found along the coast, along with highest nanophytoplankton proportional presence (Fig. 4, middle-right panels). Highest abundance of picophytoplankton was observed more offshore, although near the

Dogger Bank there was a decrease in picophytoplankton abundance. The total O/R ratio showed an increased abundance of Orange fluorescent dominating species in most of the offshore waters between 50 and 150 km, and a decrease in relative abundance near the Doggersbank area (Fig. 4, middle-right panels). Microphytoplankton abundance was less than 8%, yet they represented the largest contributors to red fluorescence (Fig. 3). They show a somewhat patchy distribution along the coast and near the Dogger Bank. The $F_v/F_m$ increased in comparison to May in the coastal region, but not in offshore regions

in the Southern North Sea (Fig. 4s).

In August the increase in phytoplankton abundance observed in front of the Dutch Delta in June was still visible, but more northerly, the biomass seemed to decrease again (Fig. 4e). Unfortunately no data are available for July to check whether this community displaced northward over time. The picophytoplankton was present at highest abundance (> 80%) and only

slightly lower values were observed (but still > 70%) along the southern Dutch coast, where the abundance of nanophytoplankton was higher. Microphytoplankton was hardly observed, only a small patch (2-3.5%) was observed near the coast of the province of North Holland. The effective absorption cross section was low in the coastal zone in comparison with offshore regions (Fig. 4x).

**3.4 Spatial clustering**

Spatial clustering (uHMM) of photophysiological characteristics and phytoplankton community composition was performed to get an overview of spatial heterogeneity and the variability over the season (Fig. 5). Colinear variables were removed based on the variable inflation factors (VIF>6), which resulted in removal of the photophysiological parameters $P_{max}$, $F_v/F_m$, $a_{LHII}$, $n_{PSII}$, and the FCM-parameter of the total red fluorescence. The five defined phytoplankton groups (Table 2) had a

higher VIF than 6 (maximum 9.6), but were retained for the sake of completeness. The remaining variables were the five FCM-defined phytoplankton groups (Table 2), the total O/R ratio and five photophysiological parameters ($1/\tau$, [RCII], $\sigma_{PSII}$, $\alpha$ and $E_k$). PCA analysis was performed to get an overview of the variables that explained most of the variation of the identified spatial clusters and to understand main drivers per region. The first two components of the PCA analyses explained 49.2% (June) to 59.7% (August) of the variance.

UHMM analysis resulted in identification of two to four spatial clusters, which were not uniform over the months. In April the most distinct phytoplankton community is found in the Northern part of the Dutch North Sea (Fig. 5). The biomass concentration in this region is low and the phytoplankton community characterized by a high O/R ratio and high





photosynthetic affinity (α). In April the whole coastal region is identified as the same spatial cluster, with high biomass concentration and a variable phytoplankton community. A small region offshore (~ 70 km) in the Southern North Sea has a very uniform phytoplankton community, which seems mainly consisiting of microphytoplankton with a low light saturation level ($E_k$) and low effective absorption cross section ($\sigma_{PSII}$). In May and June, the phytoplankton community in the Dutch

North Sea seems quite uniform with only two distinct spatial clusters. In June these clusters are clearly divided in a coastal and an offshore zone, while in May the clusters are spatially less well defined. The uniform phytoplankton community found in April in a small region offshore (~ 70 km) in the Southern North Sea remains present but in May a similar community is found in the eastern coastal region. Unfortunately, this region was not sampled in April. This spatial cluster is still very uniform, but not with distinct drivers in comparison with the other spatial cluster. It seems mainly typified by a high

effective absorption cross section and low biomass and low O/R ratio. In June a distinct separtion between coastal and offshore phytoplankton communities is present. The offshore phytoplankton community is consisting of a diverse phytoplankton community while the coastal phytoplankton community is consisting of mainly micro phytoplankton with low light saturation level and high photosynthetic affinity (α).

August was the most heterogeneous month with four different phytoplankton communities (Fig. 5). The first phytoplankton community, which covers the most Northern most part of the Dutch North Sea and the coastal region of the Northern part of the Netherlands and the offshore region of Noordwijk, were characterized by high effective absorption cross section and rate of reopening of closed RCIIs ($1/\tau$). The second phytoplankton community is corresponding to the southern coastal regions, a region with high freshwater influx, and is positively associated with most phytoplankton groups and the amount of RCII's

per volume. The third phytoplankton community is characterized by low light saturation level and high photosynthetic efficiency and low rate of reopening of closed RCIIs ($1/\tau$). Finally, the fourth spatial cluster was also found in April and May; in August it is a more a variable group of phytoplankton and the northern coastal region expands more to the south. Different spatial clusters were appointed to the same region visited within a two-day time span twice; in the north-eastern coastal region and at the transect of Noordwijk. Both times, the third cluster is one of the overlapping spatial clusters. The

third cluster corresponds to only night time sampling periods and is defined by low light saturation level and low τ, suggesting that this cluster is more a temporal than spatial cluster.









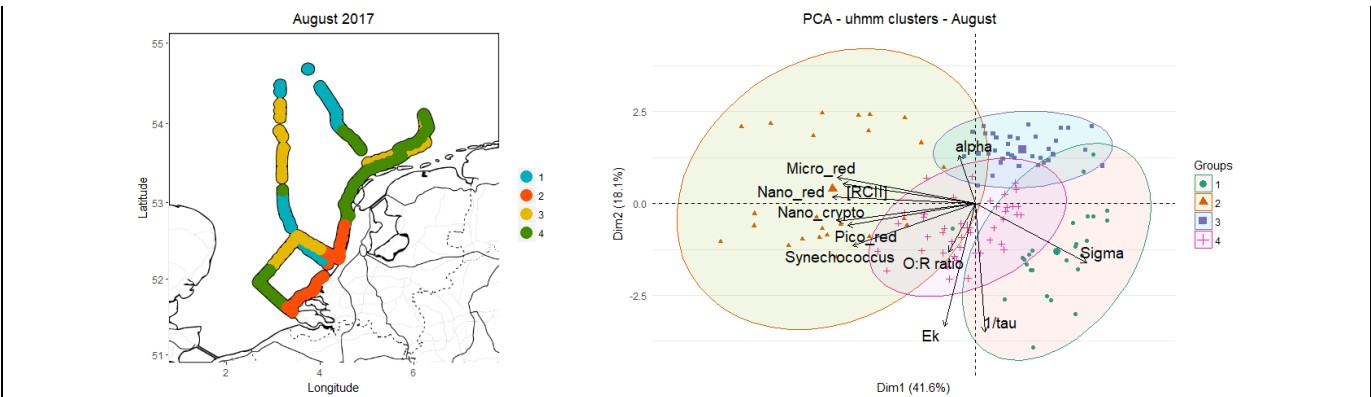

**Figure 5: Maps separated per month of spatial clusters as defined by uHMM clustering (left) and a bi-plot of the PCA of the data (right) with as variables the FCM-based parameters O/R ratio and the total red fluorescence of the five described phytoplankton groups (Table 2) and non-colinear FRRf-parameters on photophysiology ($1/\tau$, [RCII], $\sigma_{PSII}$, $\alpha$, $E_k$). In the bi-plot of the PCA colors represent assigned spatial clusters with confidence elipses (confidence 95%). Overlapping conficence elipses suggest a high similarity between groups while the size of the elipse is a measure of variability within the group.**

In April the uHMM did not visualize the phytoplankton community with distinct O/R-ratio north of Terschelling (Fig. 4e). Manual increase of the number of states in the spectral classification to four did not result in a different spatial cluster at the aforementioned location (Fig. 6), but instead split up the coastal spatial cluster with a distinct community off the coast of Noordwijk and a small patch to the north-west. Forcing another spatial cluster did result in a visualisation of the community with high O/R-ratio north of Terschelling (Fig. 6), but cluster this region with a more northern part of the transect and the coastal region of Noordwijk These regions do also show a higher O/R-ratio in the spatial map (Fig. 4e).

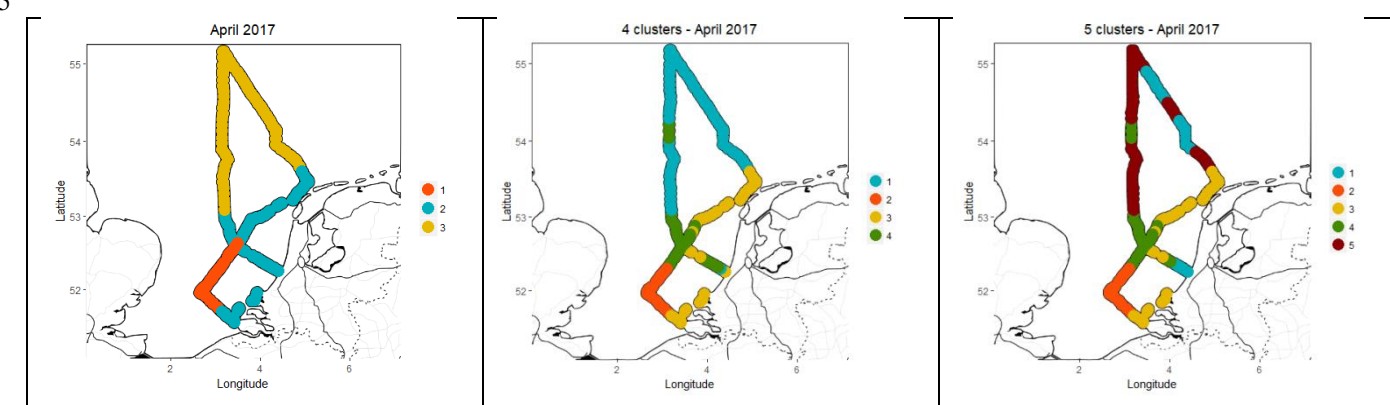

**Figure 6: Maps of the cruise in April of spatial clusters as defined by uHMM clustering, with automatic set number of states (left) and manually increasing the number of clusters to four (middle) and five (right). Cluster variables consisted of the FCM-based parameters O/R ratio and the total red fluorescence of the five described phytoplankton groups (Table 2) and non-colinear FRRf-parameters on photophysiology ($1/\tau$, [RCII], $\sigma_{PSII}$, $\alpha$, $E_k$).**





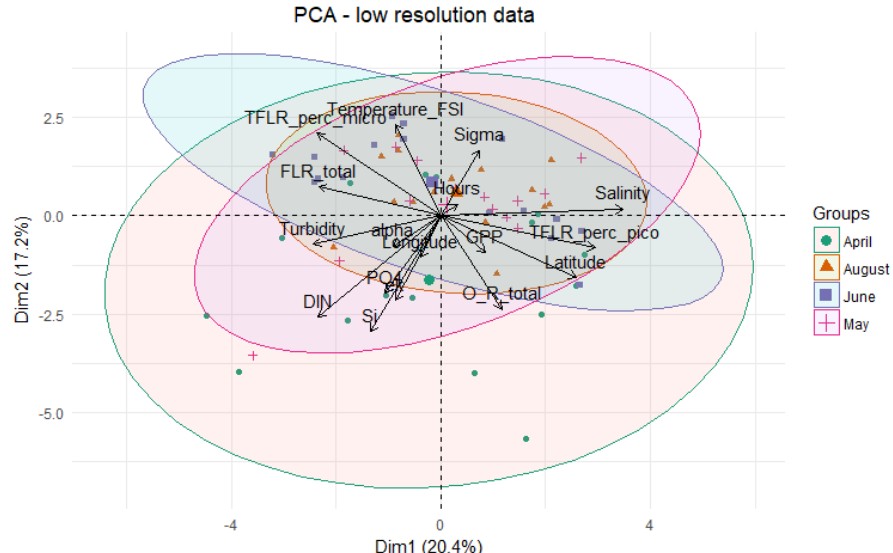

**Figure 8: bi-plot of the PCA of the combined high and low resolution data (n=61) with as variables the FCM-based parameters**
**(O/R ratio, TFLR and percentage of microphytoplankton and picophytoplanktcon to the TFLR), the non-colinear FRRf-**
**parameters (1/τ, [RCII], σ$_{PSII}$, α, E$_k$), and abiotci data (DIN, PO4, Si, salintiy, Temperature, Turbidity) spatial data (Longitude,**
**Latitude), time of the day (Hours) and the gross primary productivity (GPP). Colors represent different months with confidence**
**elipses (confidence 95%).**

### 3.5 PCA of the standard MWTL measuring points

The bi-plot of the PCA of the low resolution data combined for all months, shows that despite large differences in absolute

values of abiotic and biological parameters (Table 3), the confidence ellipses are largely overlapping, suggesting the drivers

of the different months are similar. The bi-plot further visualizes a negative relation between the nutrient concentrations

(DIN, PO$_4$ and Si) and σ$_{PSII}$ (Fig. 8) and, although σ$_{PSII}$ can also depend on the species (Kolber et al., 1988; Suggett et al.

2009), it does not seem to associate with any of the flowcytometer clusters. Furthermore, the TFLR of the picophytoplankton

is negatively associated with the total and with the microphytoplankton part of the TFLR.

### 3.6 Water column integrated primary productivity

Water column integrated gross primary productivity (GPP) over the water column was calculated with high spatial and

temporal resolution. Productivity ranged from minimum 7.5 mg C m$^{-2}$ h$^{-1}$ (June) to maxima of 2024 mg C m$^{-2}$ h$^{-1}$ in May.

The monthly average was highest in April (781 ± 409 mg C m$^{-2}$ h$^{-1}$) and lowest in August (68 ± 39 mg C m$^{-2}$ h$^{-1}$), although

these averages are not completely comparable due to different ship routes per month (Fig. 4).





Figure 7 shows the spatial heterogeneity of gross primary productivity per month. April corresponds to the month with the highest biomass concentration and higher nutrient concentration in comparison to the other months, resulting in higher GPP. In the coastal zone nutrients concentrations and turbidity are higher due to river water influx, but GPP in this region is lower. In April, phytoplankton populations were not nutrient limited in most areas which makes light availability a better predictor

5 for primary productivity. In April offshore water column productivity shows quite some spatial variability and it is higher in comparison to the coastal zone, likely due to a lower light attenuation in the offshore water column, while in the other months the opposite appears, confirming this hypothesis. GPP is highest offshore (> 800 mg C m$^{-2}$ h$^{-1}$) along the 70 km line to west of Den Helder and high near the coastal Wadden Sea. In May spatial variability is limited with some very local high GPP values (> 600 mg C m$^{-2}$ h$^{-1}$), but most values show production rates below 213 mg C m$^{-2}$ h$^{-1}$. In June the Dutch North

10 Sea showed slightly more spatial variability, but GPP was lower than in May. Highest values in June were observed (300-400 mg C m$^{-2}$ h$^{-1}$) northwest of Noordwijk, where in April the values were low. In August, a similar spatial distribution with low GPP is visible as in June with the majority of values below 100 mg C m$^{-2}$ h$^{-1}$. Yet, the GPP rates on the Terschelling transect were about twice as high as in June.

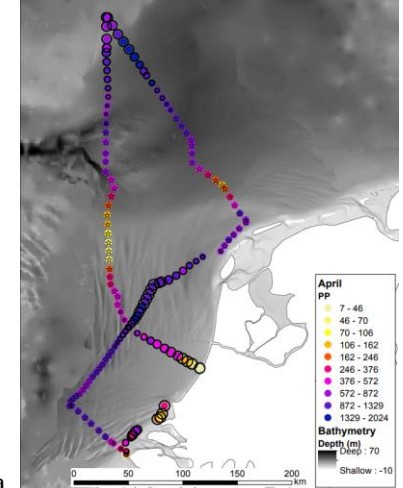

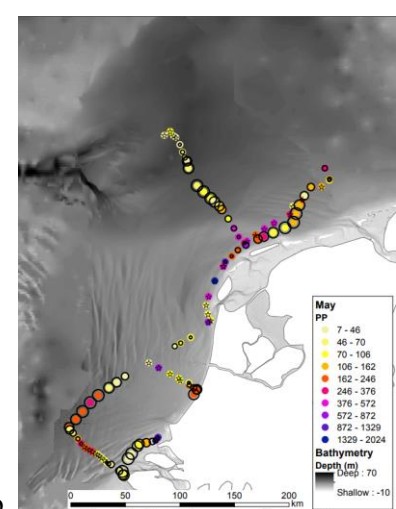

a.                                                                b.





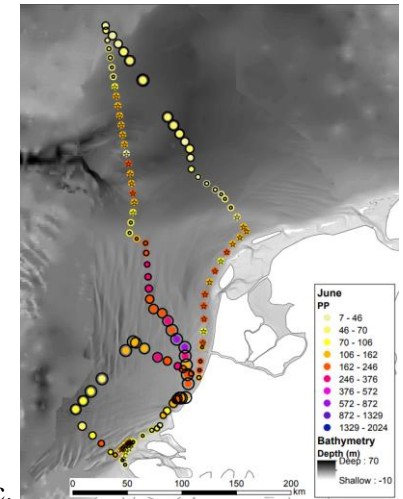
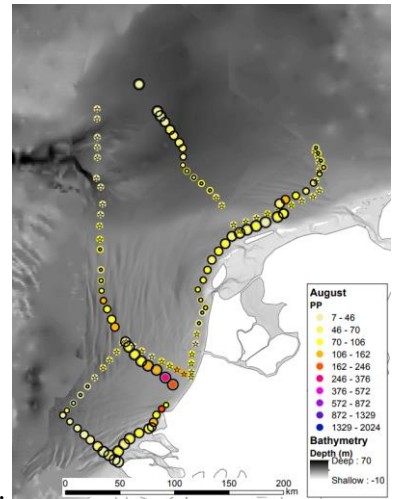

**Figure 7: Water column integrated gross primary productivity (mg C m⁻² h⁻¹) per month (a, April, b. May, c. June, d. August). Symbols: stars: measurements taken at night; filled circles: samples taken during the day. The size of the open circles relate to the modelled irradiance using the clear sky ASHRAE model by Ramsey (http://www.me.umn.edu/courses/me4131/SchematicsAndOtherFiles/Solar/ ).**

A stepwise multiple linear regression (n=61) for all months combined revealed that significant interactions included photophysiology (α), phytoplankton biomass (TFLR) and abiotic predictors (Turbidity, DIN, time of the day and temperature). A log transformation of the GPP was necessary to correct for the heteroscedasticity of the data. Colinear predictors were removed before analysis (VIF>6), which included $F_v/F_m$ (collinear with alpha), $E_k$ (collinear with $P_{max}$), $n_{PSII}$ (collinear with $\sigma_{PSII}$), [RCII] and $a_{LHII}$ (colinear with total red fluorescence). Statistically non-significant predictors included

the percentage of micro or picoplankton, maximum rate of photosynthesis, silicate and phosphate concentration, $\sigma_{PSII}$, salinity, the mean O/R ratio, and spatial predictors (latitude, longitude). Significant abiotic predictors included the time of the day (hours), turbidity, temperature and DIN (Table 4). Surprisingly, temperature and DIN concentrations are negatively correlated with the GPP, which could be a biased effect of not separating different months.

**Table 4: Coefficients of the stepwise multiple linear regression (n=61) for ln(GPP) with p<0.05 and VIF<6**

|  | coefficients |
| --- | --- |
| Intercept | 5.613 |
| alpha | 2.916 |
| Turbidity | $-9.929*10^{-2}$ |
| DIN | $-3.567*10^{-2}$ |
| Temperature | $-1.887*10^{-2}$ |
| Total red fluorescence (biomass) | $2.833*10^{-9}$ |
| Hours | $4.141*10^{-2}$ |





As shown above, variability in the data is related to the month (i.e. seasonal patterns) of sampling and the area sampled (i.e. to different abiotic and biotic factors). However, physiological activity can also be influenced by diurnal patterns, and we therefor investigated if our data show might be influenced by influenced diurnal influenced patterns in photosynthetic activity. This was done by calculating the z-scores per day and plot these as a function of the diurnal time. The results show clear diurnal trends in photosynthetic activity (Fig. 9). $P_{max}$, $E_k$, $\sigma_{PSII}$ and $1/\tau$ are all higher during the day than at night, while $F_v/F_m$ and $\alpha$ are lowest in the early afternoon (Fig. 9).

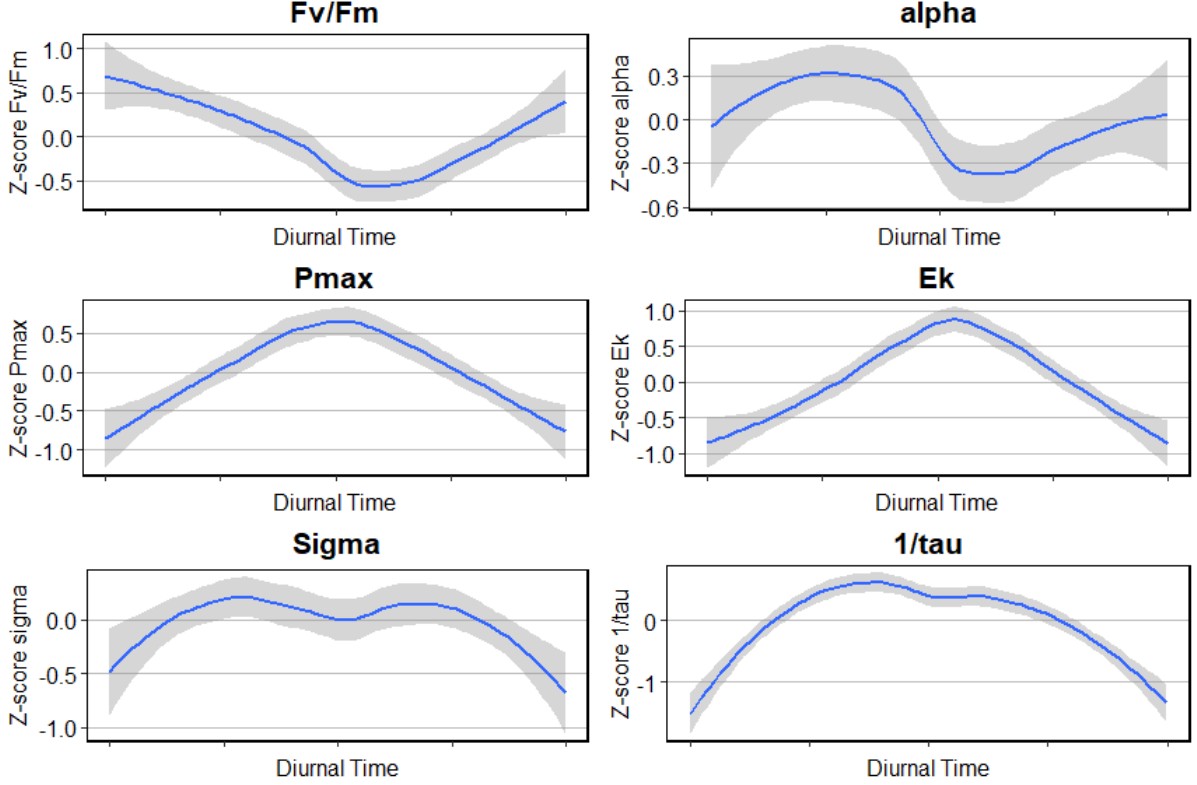

**Figure 9: Diurnal trends in photophysiological data. Z-scores were calculated by subtracting the daily mean from the value and dividing by the standard deviation of that day. Partial days were excluded because this could potentially offset the daily mean and standard deviation and would therefore not give reliable results.**





## 4 Discussion

This study examines the use of high-resolution measurements to supplement low-resolution monitoring. . Multiple parameters with high spatial resolution provide an overview of current conditions. Spatial clustering may serve as an example of how to use multiple high-resolution parameters in a monitoring program. Lastly, the water column integrated

gross primary productivity of the Dutch North Sea is estimated and its main forcing factors are identified.

Environmental conditions in the Dutch North Sea are spatially heterogeneous and strongly influenced by seasonal dynamics. The timing of the phytoplankton bloom period corresponds well to the study of Baretta-Bekker et al. (2009) on phytoplankton dynamics in the Dutch North Sea from 1991 to 2005. In April we covered a phytoplankton bloom period,

typified by high biomass concentrations, high quantum efficiencies and electron transport rates of PSII. The May cruise covered the collapse of the phytoplankton bloom period, as shown by the lower average quantum efficiency of PSII and high variability in biomass concentrations (Table 3). The cruises in June and August covered a low Chl *a* period and a second late summer bloom period was not detected. Although a second bloom period is known to occur in some regions of the Dutch North Sea, an onset later than August is not unusual (Baretta-Bekker et al., 2008). Generally, pico-autotrophs contributed

considerably to cell numbers but covered only a small fraction of the total biomass (Fig. 3). As nutrient limitation progressed from April to August, the relative abundance of picoplankton reached over 80%, which corresponded to less than 30% of the relative fluorescence. In June and August the molar nutrient N:P ratios were generally below the Redfield ratio and concentrations were in the limiting range, suggesting that phytoplankton populations were N-limited in a large part of the Dutch North Sea. This impacts the community composition: generally it is assumed that nutrient limitation favours small cell

size, because of the higher surface to volume ratio of smaller cells, and that fluctuating nutrient concentrations favour larger cells due to their greater maximum uptake rate and storage capacity (Stolte and Riegman, 1995; Giannini and Ciotti, 2016; Philippart et al., 2000), and the shift towards smaller species observed by us using FCM is thus in accordance with this theory. The change in community composition over the season has implications for the whole ecosystem, because microphytoplankton is a better food source for higher trophic levels than picophytoplankton, which is more involved in the

microbial food web, with less trophic efficiency and low contribution to carbon export (Quere et al., 2005). Nutrient limitation does not only affect community cell size but also low values of $F_v/F_m$ are often related to nutrient limitation (Kolber et al. 1988, Kolber and Falkowski 1993, Beardall et al. 2001, Ly et al. 2014), although this is not always the case, and it seems likely that after acclimation to limiting nutrient conditions $F_v/F_m$ can recover again as was seen in the current study in June (see also Kruskopf and Flynn, 2006). The $\sigma_{PSII}$ was negatively associated with DIN and turbidity in the PCA on

the low resolution data (Fig. 8), and although this value is assumed to vary per taxonomic group, it is not associated with any flowcytometer group (Kolber et al., 1988; Suggett et al., 2009), hence most of the variability seems to be driven by light and nutrient conditions. The values for the effective absorption cross section are slightly lower but in similar range to other studies (Suggett et al., 2009).





The gross primary productivity as found in the current study was both spatially and temporally variable. Average surface productivities of $44 \pm 64$ µg C $L^{-1}$ $h^{-1}$, peak primary productivity in April and lower values the rest of the year is in agreement with an earlier study in the North Sea coastal zone (Brandsma et al., 2011). To interpret water column integrated

primary productivity in an ecological or biogeochemical meaningful way, the FRR units of electrons per unit time were converted to carbon units. Gross photosynthesis correlates well with photosynthetic oxygen evolution (Suggett et al., 2003), and multiple studies have shown good correlation between [14]C-derived estimates of primary productivity and FRRf-derived estimates (Melrose et al., 2006; Kromkamp et al., 2005, 2008). In this study, the estimate of 6 moles electrons per mole carbon atom was used based on a study in the same biogeographic region by Kromkamp et al. (in prep.). This a simplified

assumption because the conversion from electron transport rate to gross primary productivity is complicated and depends on the consumption of electrons by other cell processes (Flameling and Kromkamp, 1998; Halsey and Jones, 2015; Schuback et al., 2016). The conversion factor from electron flux to carbon fixation depends on biogeographic region and taxonomy but is also subject to diurnal variation (Schuback et al., 2016; Lawrenz et al., 2012; Raateoja, 2004). The diurnal trend in coupling of electron flux and carbon fixation is dictated by cell cycle, a circadian oscillator and irradiance, as photophysiological

plasticity minimizes photodamage and optimizes growth under fluctuating light and nutrient concentrations (Claquin et al., 2014; Cohen and Golden, 2015; Schuback et al., 2016). To interpret spatial variability separately from temporal variability and to provide a more reliable estimate of gross primary productivity, Schuback et al. (2016) suggest a correction with $NPQ_{NSV}$, which needs further research.

Most photophysiological parameters we measured showed diurnal trends, but, as said, this is not only due to

photophysiological plasticity but also due to phytoplankton cell cycle (Suzuki and Johnson, 2001; Claquin et al., 2014; Schuback et al., 2016) and rhythms driven by a circadian oscillator (Cohen and Golden, 2015). The clear diurnal trends we observed are in agreement with previous studies and it is usually explained by photophysiological plasticity to minimize photodamage (Schuback et al., 2016; Behrenfeld et al., 2002). The limited dark acclimation time in our study decreases the comparability between samples because of the diurnal variation in presence of non-photochemical quenching (NPQ), which

makes the interpretation of photophysiological parameters more complicated. But although the presence of NPQ can compromise the use of fluorescence as an estimate for chlorophyll concentrations, the good relationship between the HPLC-derived Chl *a* concentration and fluorescence of the FRRf and FCM suggest that most of the NPQ is dissipated during the time in the tubing and low light pre-acclimation. Moreover, although it is difficult to separate diurnal variability from spatial variability, on a number of occasions we observed diurnal variability within the same "biogeochemical" province, as

indicated by the cluster analysis. So, it is clear that spatial patterns introduced by distinct phytoplankton communities in different biogeochemical areas were generally more prominent than the diurnal oscillations. Nevertheless, for future studies, it is advised to include a Langragian based approach where the same phytoplankton community and photosynthetic activity





can be followed during a complete light-dark cycle for a better understanding of the impact of diurnal variability in the data and the effect on the coupling between photosynthetic electron transport and C-fixation.

Primary productivity is an important parameter in system biology and climate research, but the current global annual NPP estimates are highly uncertain (Silsbe et al., 2016). Its importance is evident, being at the base of the marine food web, which

makes it worrying that in recent decades primary productivity in the North Sea seems to decline (Capuzzo et al., 2017; Cloern et al., 2014). The global declining trend in primary production has implications for the ecosystem structure and fisheries productivity and is worrying as marine ecosystems face many changes and possible threats caused by global warming and increased use of marine resources by man. This is why the improvement of estimates of ocean primary productivity is crucial. To extrapolate over a wider spatial scale remote sensing methods and models are used to estimate

primary productivity. But phytoplankton community composition or physiology is usually not included and primary productivity estimates solely based on abiotic factors in combination with Chl $a$ estimates (Cole and Cloern, 1987; Behrenfeld and Falkowski, 1997; Westberry et al., 2008; Westberry and Behrenfeld, 2013; Blauw, 2015), although some models include $P^B_{max}$ as parameters, which is parameterized from temperature only. Yet, Chl $a$ and abiotic conditions alone are limited predictors of biological processes, because the Carbon:Chl $a$ ratio is not only dependent on abiotic conditions but

also to species-specific phenotypic plasticity needed to acclimate to those abiotic conditions (Flynn, 1991, 2005; Geider et al., 1997; Alvarez-Fernandez and Riegman, 2014) and Chl $a$ is still difficult to measure in turbid case-2 waters. Therefore, *in vivo* measurements are required to calibrate remote sensing based models and we suggest that automated production measurements based on FRRf methodology can fulfil this role.

Depth integration of high-resolution measurements is depending on light penetration through the water column and assuming vertical homogeneity. For most part of the year, the assumption that the mixed layer depth (MLD) reaches below the euphotic zone and causes vertical homogeneity in photoacclimation and community composition, is a safe assumption for the Dutch North Sea, yet short-lived thermal stratification is a regional phenomenon in summer (Van Leeuwen et al., 2015). This short-lived thermal stratification can result in subsurface chlorophyll maximum layers, which, when MLD is shallower

than the euphotic zone, will result in a phytoplankton community with distinctly different photophysiological characteristics. Additionally, to calculate water column productivity, an assumption on light penetration through the water column is needed. In this study, light extinction was actively measured approximately ten times per cruise and based on the correlation with turbidity these figures are spatially interpolated using linear regression. Although the light attenuation in the water column is strongly influenced by turbidity, the situation is more complex involving not only underwater processes (absorption and

scattering) but also surface processes like reflection and refraction (Brown et al., 1984). Additionally, turbidity is measured in the near-infrared (880 nm), but different substances in the water have characteristically shaped light absorption spectra and photosynthetic active radiation spans a wide range of wavelengths (400-700 nm), this nonlinearity can make the light



attenuation coefficient based on turbidity a rough estimate (Kirk, 1994), but as we observed a good correlation between turbidity and $K_d$ ($r^2=0.77$), we assume our $K_d$ estimates are reliable.

The use of automated cluster analyses to interpret spatial heterogeneity is a necessity when dealing with the high amount of
data collected by high-resolution methods. The unsupervised Hidden Markov Model (uHMM) method used in this study was originally designed to detect phytoplankton blooms and understanding the involved dynamics, but here used to identify different phytoplankton communities at the regional scale (Rousseeuw et al., 2015). In general, we see in all months a clear distinction between phytoplankton communities of the coastal zone and off-coast regions. A further separation between the Dutch south coast and the coast off the northern Wadden Islands can usually be made. A separate off-coast area seems to be
the southernmost study area, i.e. the northern corner of the Walcheren transect. August is the most heterogeneous month, while both biomass and nutrient concentrations are low, suggesting that niche differentiation is more strongly present than in other months. Broadly, August conditions correspond to the hydrographical regions formerly identified in the Dutch North Sea (Fig. 10; Van Leeuwen et al., 2015; Capuzzo et al., 2015). However, the model was not able to automatically visualize all spatial heterogeneity. In April a distinct phytoplankton community was present off the coast from Terschelling, only
resulting in a distinct cluster when manually increasing the number of spatial clusters from three to five. Additionally, temporal variation was interfering with the spatial clustering in August. Although such models are useful for visualization and following changes in spatial heterogeneity, input and output need to be critically evaluated before implementation. To test whether the differences between months result from seasonal variation or other factors, results over multiple years and additional seasonal cruises need to be made to better characterize heterogeneity of the phytoplankton community structure.

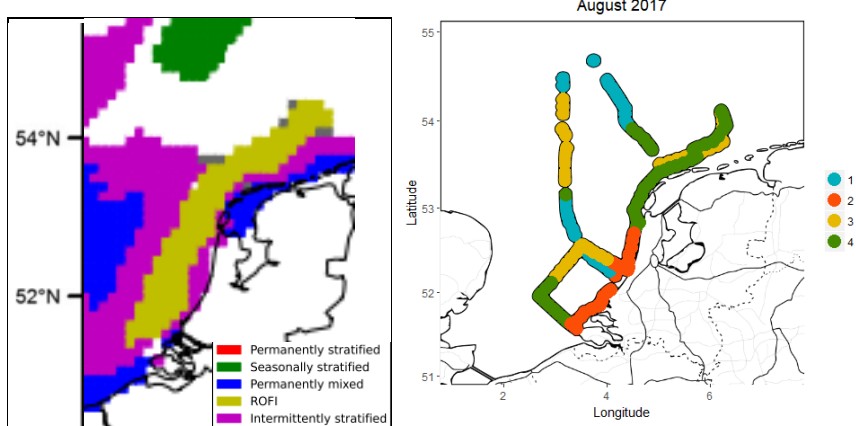

**Figure 10: hydrographical regions as defined by Van Leeuwen (2015; left) and spatial clusters by uHMM clustering in August (right).**



Currently, biological monitoring of phytoplankton in the Dutch North Sea is dictated by the requirements set by OSPAR and the EU Marine Strategy Directive and limited to HPLC analysis of Chl *a* concentration, microscopy counts of *Phaeocystis* cells, and at a few stations, coccolithophores or toxic dinoflagellates. Sampling points were reduced from almost 70 in 1984 to less than 20 today, while strong seasonal patterns, high riverine input, and tidal forces make the Dutch North Sea a region

with high spatiotemporal variability. At the same time, the Dutch North Sea is an area under high anthropogenic pressure, which has led to substantial biogeochemical changes over the past decades (Burson et al., 2016; Capuzzo et al., 2015 and 2017). These abiotic changes affect biology, with potential large implications for ecosystem function and services (Prins et al., 2012; Capuzzo et al., 2017; Burson et al., 2016). Systematic and sufficient monitoring of these changes is of crucial importance to recognize threats, and, once identified as such, develop mitigation actions. The current low-resolution

monitoring program is clearly not able to cover the entire biological variability. For instance in April, both Noordwijk 70 and Terschelling 235 km show high gross primary productivity, suggesting that production of the entire area between these points is similar, but both high and low productivity rates occur (Fig. 7). Extra sampling points in clearly deviating areas would be very useful, because only low-resolution offer the level of detail which is required to identify toxic, keystone or invasive species. Yet, adding high-resolution methods to the current monitoring program will already allow for obtaining

sensible information between sampling points. A smart monitoring system should use high-resolution methods as it delivers information which is difficult to obtain otherwise, can be used to calibrate and validate remote sensing model and can also be used to identify extra sampling points, possibly even based on real-time projections, opening up early warning methodologies.

**5 Conclusions**

The combination of FRR fluorometry and flowcytometry offers an elaborate view of the phytoplankton community. Accounting for diurnal patterns and identification of FCM clusters for functional types such as nitrogen fixers, calcifiers or DMS-producers are steps needed to increase the value for interpretation ecosystem dynamics and biogeochemical fluxes. Data interpretation may be supported by automated cluster analyses, such as the uHMM used in the current study, to interpret spatial heterogeneity and to deal with the high amount of data collected by high-resolution methods. However, our

model needs to be improved to capture more of the spatial heterogeneity present in ecology of the Dutch North Sea. Overall, the addition of high-resolution monitoring is a very useful supplement to current monitoring.

**Acknowledgements**

We want to thank the captain and crew of the RV *Zirfaea* and the shipboard Eurofins employees for their hospitality and great help during the cruises. We thank Annette Wielemaker for assistance with the GIS maps of primary production, René Geertsema for assistance with the flowcytometry data analysis and Ralf Schiebel for useful comments on the manuscript.

Furthermore we would like to thank the Rijkswaterstaat laboratory for conducting the nutrients and chlorophyll measurements and Rijkswaterstaat for the opportunity to perform measurements alongside the regular monitoring program. This project has received funding from the European Union's Horizon 2020 research and innovation programme under grant agreement No 654410 (Jericho-Next).

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
