# Peer review of "Manuscript under review for journal Ocean Sci."

_Ocean Science, 2018_

## Referee Comment (RC1) · Anonymous Referee #1 · 25 Jun 2018

The manuscript by Aardema and co-authors investigates high resolution in in situ measurements of phytoplankton photosynthetic activity and abundance in the Dutch North Sea. The main topic of this study is relevant and provides useful information, particularly when considering monitoring requirements and in defining sampling/monitoring strategies. This study is also a very good example of integrated sampling and outputs from different instruments (i.e. fRRF, flow cytometer, FerryBox).

General comments: - The introduction is focused on primary productivity (PP) but the main part of the paper investigates the photophysiological variables and phytoplankton groups with limited mention of productivity. I would suggest emphasising more the

estimates of PP throughout the ms. - Collinearity between variables: flow cytometer (FCM) phytoplankton groups were considered in the analysis even if showing collinearity (VIF>6). Statistical principles should be applied consistently across the analysis and to all the variables. If not, this should be explained clearly. - Spatial autocorrelation: transect data with high frequency sampling is likely to be spatially autocorrelated - has this been considered? If spatial autocorrelation is not considered to be a problem in this dataset, please explain why. Alternatively, presence of spatial autocorrelation could be investigated with the use of variograms. - Diurnal changes in some of the photophysiological variables: the authors clearly show that the diurnal cycle affect the clustering of observations (e.g. Page 25), so the clusters identified were not only based on changes in phytoplankton community but also in sampling activity (i.e. day vs night). As stated in the ms, it is difficult to separate the temporal from spatial variability; however, the effect of spatial variability could be investigated, for example, using measurements collected around specific time of day or night (e.g. 12:00+/4 hours) and rerunning the cluster analysis on this sub-dataset and comparing the outcome with the current clusters. In this way it would also be possible to test the suggestion in line 30-31 (page 27) that spatial patterns are more important than temporal. - Part of the text requires rewording – see technical section below for details.

Specific comments - Title – phytoplankton photosynthesis does not provide a clear idea of the content of the paper that covers different photophysiological variables as well as measurements of PP. I would suggest to being more specific. - Data analysis it would be useful if the authors could explain why clusters, stepwise regressions and PCA have been used as chosen statistical analysis and what they are you aiming to explain with these techniques? - Biomass vs chl a – repeatedly in the ms the authors refer to 'biomass', as synonymous of chl a (from validate fluorescence). Although chl a is often used as a proxy for phytoplankton biomass, they are not the same and this should clearly be stated at the start of the ms. Confusion arises from figures and tables referring to 'abundance', 'fluorescence', 'chl a', while the text refers to 'biomass'; please check for consistemcy. In addition, the implications of a variable Chl-a : C

ratio should also be considered and discussed. If the main interest is on biomass the authors could consider calculating it from the FCM measurements (for example, see DOI: 10.1016/j.dsr2.2006.05.004). - UHMM and cluster identification – it is not clear whether the clusters between the different months (Figure 5) are the same or not – in other words, is cluster 1 in April characterized (defined) by the same variables as cluster 1 in May? If not, then it may be better to separate the clusters e.g. with different numbers and/or colours in the figures - Discussion of results: results of the analysis of the photophysiological variables and of PP appear discussed separately. Outcomes from these two parts of the study should be brought (and discussed) together, where possible. - Conclusions – I would suggest to highlight the importance of this study for monitoring programme. Also, a bit more considerations on combining low and high-resolution measurements would be useful. Supplementary information – need to be linked (and referred to) in the main text of the ms, otherwise it may be difficult for the reader to know that this info is available.

Technical corrections Page 1: 23-26 – rewording is needed Page 1 30 -keywords, consider adding primary productivity Page 2: 10-12 – this sentence would fit better at the start of the paragraph. It also requires references Page 3: 5 – 'a sum': consider replacing with 'a combination' Page 3: 23 - 'pigment ratio' slightly incorrect as the ratio considered is of fluorescence Page 3: 24-25 – Aims – this statement about key driver of PP is very general and can be misinterpreted as the ms focuses on only 4 months during the growing season of a particular year. Time frame of this study should be specified Page 4: 3-5 – not clear, needs rewording Page 5: 1- would be useful to have the exact dates of the surveys. Page 5: 6 – more details on the temporal frequency indicated as 'low resolution' should be provided (e.g. how many samples per station? How many a day? How many depths?) Page 5: 27-32 – please provide more details of the methods or a published reference (for people not being able to access the internal protocols). Page 6: 16 & 18 – acronyms (e.g. NPQ and F0') should be explained when used the first time Page 8: 12-13 – formula 8 is missing Page 8: 17 – need rewording Page 8: 20-21 – it is not clear how surface irradiance was calculated; please reword
this section Page 9: 17 – was the clustering carried out by the FCM software or was it done by expert judgment manually? Also, was data cleaned from potential presence of air bubbles etc? Please provide details on these points, Page 10: 2 – outliers – specify which analysis you are referring to (e.g. outliers from the fRRF?) Page 10: 5 – provide a reference for the value of 0.65 Page 10: 12 – please specify which are the photophysiological variables considered Page 10: 13 – acronyms (VIF) should be defined here Page 11: 20 – 'nitrate': should this be 'DIN'? Page 11: 27-28 – please explain the evidence for P and Si-limitation (i.e. discuss the ratios vs expected limiting ratios in literature). Also, please specify the value of Redfield Ratio and reference Table 3 legend – 'not completely comparable': this expression doesn't have a clear statistical meaning. Please specify briefly in the legend which month had a different sampling route and station so for the reader to understand in which month the study area is not fully covered. Figure 2 provide equations of linear regressions with R2 and significance Page 14: 27 – 'suggesting physiological stress', please provide reference Page 16: 9 – it is not clear to which phytoplankton group the % are referring to. Page 16: 14 – please specify which are 'these regions' Page 16: 15-16 – this paragraph should be moved to the discussion so to allow the concept to be developed further. Page 16: 17 – please explain why low sigmaPSII may reflect Rhine River waters. Page 17 – Figure 4 – I appreciate the different scaling was necessary to 'visualize the spatial heterogeneity' however it makes very hard the comparison between figures. In fact, the reader needs to keep checking the legend, which is printed in very small characters difficult to see. I would suggest reconsidering the use of a uniform scale (at least for some of the variables, if possible). Page 18: 17 – there is limited or no comments on the results of some of the photophysiological variables such as alpha, Pmax, effective absorption cross section. Page 18: 25 – 'sake of completeness'. See general comment about collinearity, please explain why statistical principle of VIF>6 was not applied consistently to all variables Page 18: 28-29 – table should be provided (for example in the additional info) showing the contribution of each variable to the PC1 and PC2 for the 4 months, and total variance explained. Page 19: 1 – alpha is defined as Light

utilisation efficiency (Table 1) but then in the text is referred to as 'affinity'. please check for consistency. Page 21: 8-13 – consider whether to move this text in additional info (or to remove it?). It breaks the flow of the results and the addition of clusters 'manually' appears to not be meaningful and/or significant (as it doesn't adopt the same statistical robust principle). Page 22: 6 – 'abiotic' and 'salinity' misspelled Page 22: 9 – as for previous PCA, please provide variables used and information on their contribution towards variance explained. Page 23: 6-7 – this paragraph is not clear particularly what is meant with 'opposite' Figure 7 legend – Size of the open circles is a bit confusing and misleading as the reader may assume the size of the bubble refers to the amount of PP. Consider simplifying the figures and only plot productivity Page 24: 15 – please indicate how much of the variability in PP is explained by the stepwise regression (e.g. $R^2$?). Page 25: 4 – reword please. Page 26: 2-5 – require rewording particularly the need to clarify and be more specific on the work done in this study. Page 26: 5 – this sentence may be misleading. The authors calculated PP along the sampling transects but did not provide an estimate for the wider Dutch North Sea as it may appear here. Page 26: 8 & 11 – timing of the bloom is discussed in this section however it would not be possible to define the start of the bloom based on a 4-day sampling per month. Continuous observations throughout the year by an instrument buoy or remote sensing would allow to 'contextualise' the measurements within the growing season (i.e. determine when sampling was carried out within the phytoplankton growing season). Page 26: 24-25 – please reword Page 27: 8-9 – repetition of method; should be deleted. Page 29: Figure 10 legend, possibly just my issue, I don't see the similarity between the two figures. Page 30: 13 – 'low resolution'... should this be 'high-resolution'?

---

## Referee Comment (RC2) · Anonymous Referee #2 · 26 Jun 2018

This paper analyses spatial and temporal patterns in cruise data with 3 high-resolution monitoring methods: FRRF, Flow-cytometry and Ferrybox. Correlations between the observed variables are also analysed. The large dataset, including many phytoplankton and environmental variables observed together enables the authors to understand the patterns in the various phytoplankton variables. The results could guide the optimal application of such novel monitoring methods in operational monitoring for a.o. MSFD.

General comments

The paper lacks a clearly stated research question or hypothesis to be tested. Therefore, it is unclear what is the purpose of the various analyses performed and what we

can learn from the results. Based on the conclusion that this type of "high-resolution is a very useful supplement to current monitoring", I would expect a hypothesis such as "combined high-resolution monitoring of many phytoplankton variables along with environmental variables allows us to quantify seasonal and meso-scale patterns in phytoplankton biomass, species composition and primary production. The concurrent measurement of different phytoplankton variables allows us to understand the effect of phytoplankton species composition and physiological adaptation processes on the observed patterns in phytoplankton biomass and production". Then the analysis should show how the variables should be combined to provide the most reliable estimates of phytoplankton biomass and primary production.

There are many observed variables, which are not consistently named in the text, figures and tables. Therefore it is easy to get lost in the description of patterns for all individual variables. A clear definition of variables that is consistently used throughout the text would help the reader to understand the storyline. Some of the variables observed by the FRRF seem to be very similar. Which of the variables should be used as indicator and which are redundant to answer the research questions?

In the conclusions section a recommendation on next steps would be much appreciated: what would be required to use the high-resolution methods in scope to provide reliable estimates of phytoplankton biomass, production and species composition for long term monitoring? In the introduction and conclusion the species composition is defined in functional types such as nitrogen fixers, calcifiers or DMS-producers, but these do not correspond to the phytoplankton clusters used in this paper.

Specific comments

Sentences are often long: consider breaking up in multiple sentences to improve readability. Figure 1: please show only the stations (with names/ abbreviations) used in this study (see table S1) and the areas used in the text (such as Dogger Bank, Wadden, Den Helder, Rhine outflow) so the description of spatial patterns can also be understood by people that are not Dutch. Section 2.2: please refer to international protocols/ methods rather than internal protocols. Table 1: it would help to have an additional column stating the interpretation / meaning of this variable, such as total biomass, nutrient stress, maximum growth rate, efficiency of light uptake etc. Then later in the text you can use these 'meaningful' names instead of codes, to facilitate understanding of observed patterns. Also a figure illustrating the meaning of the different variables (alfa, Ek, F', Fm' etc.) could prevent getting lost in all abbreviations. Equation 9: why did you use monthly averaged irradiance if you are looking at high-resolution patterns. Why did you not use irradiances measured during the cruise? Table 2: Since you use both Length_FWS and O/R ratio as criteria to distinguish the phytoplankton groups, it would be logical to include a column for O/R ratio with the applied criteria. It is not entirely clear whether pico-red includes pico-Synecho or not. On page 14, line 30 it says: "Both groups of picophytoplankton (Synechococcus and total)", whereas table 2 and figure 3 suggest the two groups are exclusive. Section 2.4: please state with every type of analysis what is the purpose / research questions for that analysis. For example: what are you trying to predict from what and why? Section 3.1: I don't see the value of comparing averages over whole transects (with large spatial variability, which is the subject of this paper), that are not even the same, between months. The only thing you see is seasonal patterns that are well-known from other studies and that can be summarized in section 2.1 in a description of the study area. Most of this section describes the data in table S1. I would replace table 3 with table S1 and remove table S2. N/P ratios address that same question as table S1, but with an indicator that is controversial. The text in this section (and subsequent sections) is sometimes hard to follow as it is not clearly structured in time and space and variable. We go back and forth in time. Section 3.2 describes first figure 2, then figure 3 and then again figure 2 and then figure 3. I suggest to make one section about phytoplankton biomass (figure 2) and then one section on species composition (figure 3). Page 16, line 14: the southern coastal stations are more strongly affected by the Rhine outflow than the Scheldt outflows (see for example: Lacroix, G., Ruddick, K., Ozer, J., & Lancelot, C. (2004). Modelling the

impact of the Scheldt and Rhine/Meuse plumes on the salinity distribution in Belgian waters (southern North Sea). Journal of Sea Research, 52(3), 149-163.). Figure 4: Please use consistent legends for the same variable between different months, with the same colour scheme and symbols (squares vs. circles) and with blue indicating low values and red indicating high values, so the high values stand out, more than the low values. Also captions in the table per line (red fluorescence, O/R ratio etc.) and per column (april, may etc.) would help to easier understand the figure. Section 3.5: I don't see the added value of this analysis. What does it tell us? Page 24: I suggest to mention in the table all the variables that were included in the analysis and note coefficients or 'ns' for not significant and the p values per explanatory variable. Then readers don't need to reconstruct the overview from the text. Actually, the significance test is likely not valid due to strong spatial autocorrelation in the data. Discussion: Here I would expect to get some advice: How to best estimate phytoplankton biomass from these data? Should we use total red fluorescence (best R2) or F0 (least affected by NPQ)? Is there a way to combine both (with other available variables) to get an even better estimate? Can we trust GPP from FRRF as a reliable estimate of primary production or is more work needed to achieve that goal? If so, what needs to be done? It is not really clear whether the diurnal variability in the FRRF variables is a problem that needs to be solved. Are the clusters in the FCM analysis the relevant ones to provide 'useful' information to science & society? Should we / Can we move on to other clusters that are mentioned in the conclusions? Do the FCM data help to better understand the FRRF data (and vice versa)? For example do we see diatoms under light limited conditions (high F'/Fm', high alfa, low Ek) and picoplankton under nutrient limited conditions (low F'/Fm')? Other ecological niches that we know from literature? Different conditions promoting Synechococcus compared to other picoplankton?

Technical comments

Collinear should be spelled with 2 ll's throughout the whole text. Page 9, line 4 & 5: I guess um means micrometers? Page 18, line 4: middle-right, please refer to the label

a-x. Figure 5: The figure would be easier to read if the colours per group are consistent between the cluster analysis on the right and the map on the left. Labels (A-D for April to August panels) would also help. Figure 9: Please add the hours of the day on the x-axis. Page 25, line 3: the word influenced is repeated too many times and therefor should get an e in the end. Page 28, line 11: estimates are. Line 13: parameter without s. Page 31, line 8: Jerico-next, without h.

---

## Author Comment (AC1) · 15 Oct 2018

Please find our response and the revised manuscript in the attachment.

Please also note the supplement to this comment:
https://www.ocean-sci-discuss.net/os-2018-21/os-2018-21-AC1-supplement.zip

---

## Author Comment (AC2) · 15 Oct 2018

Please find our response and the revised document in the attachment.

Please also note the supplement to this comment:
https://www.ocean-sci-discuss.net/os-2018-21/os-2018-21-AC2-supplement.zip

---

## Author Response (AR1)

**Response to interactive comment of anonymous referee #1**

By Hedy M. Aardema in agreement with co-authors.

**Reviewer***: The manuscript by Aardema and co-authors investigates high resolution in in situ measurements of phytoplankton photosynthetic activity and abundance in the Dutch North Sea. The main topic of this study is relevant and provides useful information, particularly when considering monitoring requirements and in defining sampling/monitoring strategies. This study is also a very good example of integrated sampling and outputs from different instruments (i.e. fRRF, flow cytometer, FerryBox).*

**Response**: We really appreciate the elaborate and helpful comments on the manuscript. Based on this detailed and insightful review we rewrote and restructured the manuscript extensively.

**General comments**

**Reviewer:** *The introduction is focused on primary productivity (PP) but the main part of the paper investigates the photophysiological variables and phytoplankton groups with limited mention of productivity. I would suggest emphasizing more the estimates of PP throughout the ms.*

**Response**: Although the primary productivity is a very interesting parameter to calculate, the aim of the paper is to give a broader view of the phytoplankton community. Therefore, we shortened the part on primary productivity in the introduction, but did give it more attention in the results and discussion sections.

**Reviewer:** *Collinearity between variables: flow cytometer (FCM) phytoplankton groups were considered in the analysis even if showing collinearity (VIF>6). Statistical principles should be applied consistently across the analysis and to all the variables. If not, this should be explained clearly.*

**Response**: This is a good point. We reran the PCA and spatial clustering with the VIF>6 variables excluded. The Multiple Linear Regression was removed from the manuscript, because of the lack of information derived from it together with the abundance of literature already addressing the predictors of primary productivity.

**Reviewer:** *Spatial autocorrelation: transect data with high frequency sampling is likely to be spatially autocorrelated – has this been considered? If spatial autocorrelation is not considered to be a problem in this dataset, please explain why. Alternatively, presence of spatial autocorrelation could be investigated with the use of variograms.*

**Response**: As the reviewer expected, most parameters were spatially autocorrelated. We tested the spatial autocorrelation with Moran's I. This is indeed a problem for the multiple linear regression, but as mentioned previously, we removed this analysis from the manuscript. For the spectral classification clustering and PCA analysis, spatial parameters (latitude, longitude) were not included in the analysis. Without time and space in the calculation we only consider features of the data, so spatial autocorrelation does not influence the results (Demsar et al., 2013, Rousseeuw et al., 2015). Because the similarity between neighbouring points is of interest, we plotted of the spectral clusters on maps to visualize the spatial heterogeneity present.

**Reviewer:** *Diurnal changes in some of the photophysiological variables: the authors clearly show that the diurnal cycle affect the clustering of observations (e.g. Page 25), so the clusters identified were not only based on changes in phytoplankton community but also in sampling activity (i.e. day vs night). As stated in the ms, it is difficult to separate the temporal from spatial variability; however, the effect of spatial variability could be investigated, for example, using measurements collected around specific time of day or night (e.g. 12:00+/4 hours) and rerunning the cluster analysis on this sub-dataset and comparing the outcome with the current clusters. In this way it would also be possible to test the suggestion in line 30-31 (page 27) that spatial patterns are more important than temporal.*

**Response**: We performed the suggested analysis for the month of August by clustering only the measurements that fall into the 12+/-4 h timeframe (see Fig. R1b). In this timeframe the southern coastal zone is distinct from the rest of the Dutch North Sea and corresponds to cluster 10 in the analysis of the complete dataset (Fig. R1a), so this cluster is defined by spatial variability. Cluster 12 and 13 are grouped together in the 12+/-4h timeframe as cluster 1. Cluster 11 is only encountered outside the 12+/-4h timeframe, so is a temporal rather than a spatial cluster. We included Fig R1 in the supplementary material and included the following text in the manuscript: "*The third cluster corresponds to only night time sampling periods and is defined by low $E_k$ and low $1/\tau$, suggesting that this cluster is a temporal cluster instead of a spatial cluster. To test this we repeated the analysis for the month of August but only including the measurements that fall into the 8 hour timeframe around noon (12:00±4h; see supplementary material Fig. S2). Cluster 11 is not recognized as cluster within the 12+/-4h timeframe, so seems indeed controlled by temporal rather than spatial variability.*"

a).

b).

[Figure]

**Fig. R1: : Maps of clusters as defined by spectral clustering of the whole dataset (left) and only the measurements at 8h around noon (8:00h to 16:00h). Based on the FCM-based five described phytoplankton groups (Table 2) and non-collinear FRRf-parameters on photophysiology ($F_v/F_m$, $1/\tau$, [RCII], $\sigma_{PSII}$, $\alpha$, $E_k$).**

Specific comments

**Reviewer:** *Title – phytoplankton photosynthesis does not provide a clear idea of the content of the paper that covers different photophysiological variables aswell as measurements of PP. I would suggest to being more specific.*

**Response**: We prefer to stay with the chosen title. The main purpose of this study was to provide an example of high-resolution methods that could serve in a phytoplankton monitoring program. Based on the results of these methods further calculation can provide an estimate of the PP or can serve in identification of distinct biogeographical regions, of which we gave examples.

5 **Reviewer:** *Data analysis: it would be useful if the authors could explain why clusters, stepwise regressions and PCA have been used as chosen statistical analysis and what they are you aiming to explain with these techniques?*

**Response**: The main aim of the data analysis was to aid in the interpretation and visualization of the multitude of parameters derived with the high-resolution measurements. The PCA reduces the amount of parameters (or dimensions) and gives an impression on the relationship between parameters. The cluster analysis was chosen to test for spatial heterogeneity; when

10 clusters would contain measurements randomly distributed over the study area, no spatial heterogeneity is present. When clustering shows spatial structure, it is. The stepwise regression was at first used to identify drivers for primary productivity, but will be removed after realization that the dataset of this study does not add to existing knowledge on this topic.

15 **Reviewer:** *Data analysis: Biomass vs chl a – repeatedly in the ms the authors refer to 'biomass', as synonymous of chl a (from validate fluorescence). Although chl a is often used as a proxy for phytoplankton biomass, they are not the same and this should clearly be stated at the start of the ms. Confusion arises from figures and tables referring to 'abundance', 'fluorescence', 'chl a', while the text refers to 'biomass'; please check for consistency. In addition, the implications of a variable Chl-a : C ratio should also be considered and discussed. If the main interest is on biomass the authors could consider calculating it from the FCM measurements (for example, see DOI: 10.1016/j.dsr2.2006.05.004).*

20 **Response**: The authors are aware of this issue and tried to address this problem in the results section '3.2 phytoplankton parameters'. Obviously, we failed to consistently address the issue in the rest of the manuscript. To improve this, the term biomass was deleted in the manuscript. Although this is a very interesting parameter, and we are working on a method to calculate biomass based on scattering measured by the FCM. We already found good agreement between our biovolume and images obtained by the Image in Flow of the FCM

25 (unpublished). However, this relationship seems to be taxon specific, which we want to study more in depth and is beyond the scope of the current study. The method to calculate biomass of Tarran et al. (2006) assumes all cells have a spherical shape and a constant C content per biovolume. Because this is an oversimplification, we prefer to use cell counts and fluorescence in the current paper. We did include our view on biomass calculation from flowcytometer data in the discussion.

**Reviewer:** *UHMM and cluster identification – it is not clear whether the clusters between the different months (Figure 5) are the same or not – in other words, is cluster 1 in April characterized (defined) by the same variables as cluster 1 in May? If not, then it may be better to separate the clusters e.g. with different numbers and/or colours in the figures.*

**Response**: we adjusted the figure as suggested.

**Reviewer:** *Discussion of results: results of the analysis of the photophysiological variables and of PP appear discussed separately. Outcomes from these two parts of the study should be brought (and discussed) together, where possible.*

**Response**: In the result section, primary productivity and Photophysiology are now both under an own header.

40 **Reviewer:** *Conclusions – I would suggest to highlight the importance of this study for monitoring program. Also, a bit more considerations on combining low and high resolution measurements would be useful.*

**Response**: We rewrote the conclusions accordingly:

"A good monitoring program monitors the presence of nuisance phytoplankton, the carrying capacity of the ecosystem and changes in biogeochemical cycling. The objective of this study was to evaluate the use of FRR fluorometry and flowcytometry for monitoring purposes. The four conducted cruises spread over 5 months offered a wide variety of environmental conditions and phytoplankton community states, which the utilized methods were able to visualize.

Inclusion of high-resolution methods in monitoring programs allows for analysis of finer scale events. Furthermore, it allows for analysis of living phytoplankton and is thereby able to measure rates and avoid effects of preservation and storage of samples. Another advantage is that high-resolution methods allows for easier comparison between countries, once common protocols have been established. Nevertheless, low resolution methods remain a necessity for more detailed taxonomic analysis, information on vertical heterogeneity, to calibrate and to correct for blanks. Data analysis might be the biggest bottleneck of the implementation of these high-resolution methods. The cluster analysis of flowcytometric data has high potential for improvement to increase the informative value of the method. Especially identification of phytoplankton clusters with a functional quality, such as nitrogen fixers, calcifiers or DMS-producers, would be helpful for interpretation of ecosystem dynamics and biogeochemical fluxes. Regarding the FRRf, the main challenge is converting electron transport rate to gross primary productivity in carbon units. Further research in these topics would benefit implementation of these methods into monitoring protocols. Furthermore, it is important to account for diurnal patterns in monitoring set-up to be able to distinguish between diurnal and spatial variability. Possibly the diurnal variability could be modelled, but more studies with a Langragian based approach would be needed for a better understanding of the impact of diurnal variability in the data. Overall, the in this study presented high-resolution measurement set-up has large potential to improve phytoplankton monitoring in supplement to existing low-resolution monitoring programs."

**Reviewer:** *Supplementary information – need to be linked (and referred to) in the main text of the ms, otherwise it may be difficult for the reader to know that this info is available.*

**Response**: Done.

**Technical corrections**

**Reviewer:** *Page 1: 23-26 – rewording is needed*

**Response:** Rephrased to: "One of the major concerns when using these methods for monitoring purposes is the presence of a diurnal cycle concurrent to the spatial variation, especially in photophysiological parameters. This concurrent presence of spatial and temporal patterns needs to be taken into account when designing a monitoring program. Nevertheless, the richness of additional information provided by high-resolution methods, such as the FCM and FRRf, can supplement low-resolution monitoring to attain a better understanding of the phytoplankton community."

**Reviewer:** *Page 1 30 -keywords, consider adding primary productivity*

**Response:** Added.

**Reviewer:** *Page 2: 10-12 – this sentence would fit better at the start of the paragraph. It also requires references*

**Response:** Moved to beginning of the paragraph.

**Reviewer:** Page 3: 5 – 'a sum': consider replacing with 'a combination'

**Response:** Done.

**Reviewer:**  Page 3: 23 – 'pigment ratio' slightly incorrect as the ratio considered is of fluorescence

**Response:** Agreed and adopted.

**Reviewer:** Page 3: 24-25 – Aims – this statement about key driver of PP is very general and can be misinterpreted as the ms focuses on only 4 months during the growing season of a particular year. Time frame of this study should be specified

**Response:** reformulated

**Reviewer:** Page 4: 3-5 – not clear, needs rewording

**Response:** Rephrased to: "The Dutch North Sea is a shallow tidal shelf sea in the southern part of the North Sea. The main water flow is Northward flowing Atlantic water that enters the North Sea in the south through the Channel. The Atlantic water flowing around Scotland enters the North Sea and meets the Channel water and the freshwater from the rivers forming the Frisian Front."

**Reviewer:** Page 5: 1- would be useful to have the exact dates of the surveys.

**Response:** Added.

**Reviewer:** Page 5: 6 – more details on the temporal frequency indicated as 'low resolution' should be provided (e.g. how many samples per station? How many a day? How many depths?)

**Response:** Added.

**Reviewer:** Page 5: 27-32 – please provide more details of the methods or a published reference (for people not being able to access the internal protocols).

**Response:** Added.

**Reviewer:** Page 6: 16 & 18 – acronyms (e.g. NPQ and F0') should be explained when used the first time

**Response:** Added.

**Reviewer:** *Page 8: 12-13 – formula 8 is missing*

**Response:** It was removed. We changed formula 9 to formula 8.

**Reviewer:** *Page 8: 17 – need rewording*

**Response:** Rephrased as: "Volumetric $P_{max}$ and α were derived by fitting $JV_{PII}$ in µmol photons $m^{-3}$ $h^{-1}$ to equation 1 (the exponential model of Webb et al., 1974) and used to integrate productivity over depth. The light availability in the water column was estimated as […] with E(z) being the irradiance at depth z, $E_{surface}$ the incoming surface irradiance and $K_d$ the light extinction coefficient."

**Reviewer:** *Page 8: 20-21 – it is not clear how surface irradiance was calculated; please reword this section*

**Response:** We adjusted the text to the following explanation: "To avoid effects of changing incident surface irradiance ($E_{surface}$) on the spatial pattern and to be able to compare GPP between regions we used monthly average surface irradiances ($E_{surface}$) in our calculations of primary productivity. From 2010-2016 irradiance (400-700 nm) was measured at the roof of the NIOZ building in Yerseke using a LI-190 quantum PAR sensor and hourly averages stored using a LI1000 datalogger. $E_{surface}$ was then calculated by averaging all irradiance data from the years 2010-2016 for the respective month."

**Reviewer:** *Page 9: 17 – was the clustering carried out by the FCM software or was it done by expert judgment manually? Also, was data cleaned from potential presence of air bubbles etc? Please provide details on these points,*

**Response:** The chosen cluster criteria were based on expert judgement. The clustering was done by the software Easyclus 1.26 (ThomasRuttenProjects) according to these criteria. Noise, air bubbles and other potential outliers were removed after the clustering.

**Reviewer:** *Page 10: 2 – outliers –specify which analysis you are referring to (e.g. outliers from the fRRF?)*

**Response:** All data, rephrased in manuscript.

**Reviewer:** *Page 10: 5 – provide a reference for the value of 0.65*

**Response:** Added; Kolber, Z. and P. G. Falkowski. 1993. Use of active fluorescence to estimate phytoplankton photosynthesis in situ. Limnology and Oceanography. 38:1646-1665.

**Reviewer:** Page 10: 12 – please specify which are the photophysiological variables considered

**Response:** We added the following sentences to the data analysis section: "Phytoplankton parameters were first tested for collinearity and predictors with a variance inflation factor (VIF) over 6 were removed (Zuur et al., 2009). This left for the cluster analysis FCM-parameters Pico-red, Nano-red, Micro-red and *Synechococcus* and the FRRf-parameters $\sigma_{PSII}$, $F_v/F_m$, $a_{LHII}$, $1/\tau$, $E_k$."

**Reviewer:** *Page 10: 13 – acronyms (VIF) should be defined here*

**Response:** Added.

**Reviewer:** *Page 11: 20 – 'nitrate': should this be 'DIN'?*

**Response:** Yes.

**Reviewer:** *Page 11: 27-28 – please explain the evidence for P and Si-limitation (i.e. discuss the ratios vs expected limiting ratios in literature). Also, please specify the value of Redfield Ratio and reference.*

**Response:** We removed the nutrient ratios from the results. The paper only reports the nutrient values as additional background information to understand phytoplankton dynamics. A detailed analysis of concentration vs ratio is past the subject of this paper, but in the discussion nutrient limitation is now discussed.

**Reviewer:** *Table 3 legend – 'not completely comparable': this expression doesn't have a clear statistical meaning. Please specify briefly in the legend which month had a different sampling route and station so for the reader to understand in which month the study area is not fully covered.*

**Response:** True. We removed the term 'not completely comparable' from the legend and added a short explanation of the differences between months. Also, we moved the table to the supplementary information and replaced it with the nutrient concentration table.

**Reviewer:** *Figure 2 provide equations of linear regressions with R2 and significance*

**Response:** The $R^2$ and significance are now added to the legend. The linear regressions are irrelevant because the unit of the x-axis is in relative fluorescence units (RFU) and instruments will require separate calibration.

**Reviewer***: Page 14: 27 – 'suggesting physiological stress', please provide reference*

**Response:** Suggett et al., 2009.

**Reviewer:** *Page 16: 9 – it is not clear to which phytoplankton group the % are referring to.*

**Response:** The nanophytoplankton. Rephrased.

**Reviewer:** *Page 16: 14 – please specify which are 'these regions'*

**Response:** Rephrased.

**Reviewer:** *Page 16: 15-16 – this paragraph should be moved to the discussion so to allow the concept to be developed further. Page 16: 17 – please explain why low sigmaPSII may reflect Rhine River waters.*

**Response:** Moved to discussion.

**Reviewer:** *Page 17 – Figure 4 – I appreciate the different scaling was necessary to 'visualize the spatial heterogeneity' however it makes very hard the comparison between figures. In fact, the reader needs to keep checking the legend, which is printed in very small characters difficult to see. I would suggest reconsidering the use of a uniform scale (at least for some of the variables, if possible).*

**Response:** We adjusted the figure to a uniform scaling.

**Reviewer***: Page 18: 17 – there is limited or no comments on the results of some of the photophysiological variables such as alpha, Pmax, effective absorption cross section.*

**Response:** We expanded the result section on the Photophysiology.

**Reviewer:** *Page 18: 25 – 'sake of completeness'. See general comment about collinearity, please explain why statistical principle of VIF>6 was not applied consistently to all variables*

**Response:** We agree that this might not have been the best choice, we preferred to include all the phytoplankton groups. As mentioned before, we now deleted the collinear variables with VIF>6.

**Reviewer:** *Page 18: 28-29 – table should be provided (for example in the additional info) showing the contribution of each variable to the PC1 and PC2 for the 4 months, and total variance explained.*

**Response:** We added this table to the manuscript, in combination with figure 6:

|  | April | | May | | June | | August | |
|---|---|---|---|---|---|---|---|---|
|  | PC1 | PC2 | PC1 | PC2 | PC1 | PC2 | PC1 | PC2 |
| Sigma | 0.8 | **28.8** | 0.1 | **36.7** | 0.0 | 9.3 | 12.1 | 9.9 |
| $F_v/F_m$ | 13.7 | 0.6 | 0.8 | **14.5** | **27.6** | 0.1 | 0.0 | **17.5** |
| $a_{LHII}$ | **18.7** | 3.4 | 17.5 | 6.7 | **20.9** | 8.7 | **21.0** | 3.3 |
| [RCII] | **17.1** | 6.6 | **20.4** | 1.6 | **28.0** | 2.6 | **25.8** | 0.0 |
| $1/\tau$ | 9.8 | **22.7** | 4.4 | 7.5 | 0.4 | 1.7 | 0.2 | **20.6** |
| $E_k$ | 3.9 | 13.8 | 0.7 | **26.3** | 3.7 | 3.9 | 0.7 | **16.8** |
| Pico-red | 4.2 | **15.1** | **18.5** | 0.4 | 6.1 | **26.9** | 0.3 | 11.8 |
| Nano-red | **16.9** | 0.0 | **21.1** | 0.6 | 2.9 | **16.9** | 15.3 | 3.1 |
| Micro-red | 10.5 | 4.5 | 16.4 | 1.4 | 6.3 | 2.9 | **22.9** | 0.4 |
| *Synechococcus* | 4.3 | 4.4 | 0.0 | 4.3 | 4.2 | **27.0** | 1.8 | 16.7 |
| Variance explained | 45.6 % | 19.3 % | 42.5 % | 18.9 % | 29.1 % | 18.7 % | 33.9 % | 25.7 % |

**Reviewer:** *Page 19: 1 – alpha is defined as Light utilisation efficiency (Table 1) but then in the text is referred to as 'affinity'. please check for consistency.*

**Response:** Changed in table. The value for alpha is the slope of the FLC, and is a measure for photosynthetic affinity for incoming light.

**Reviewer:** *Page 21: 8-13 – consider whether to move this text in additional info (or to remove it?). It breaks the flow of the results and the addition of clusters 'manually' appears to not be meaningful and/or significant (as it doesn't adopt the same statistical robust principle).*

**Response:** It is true that it does not adopt the same statistical robust principle. However, there is spatial heterogeneity in the flowcytomer data, that are not visualized with the UHMM and this is what we wanted to explore. We do agree that the manual increase of amount of clusters might not the best way to go forward with this, so we deleted this section from the manuscript.

**Reviewer:** *Page 22: 6 – 'abiotic' and 'salinity' misspelled. Page 22: 9 – as for previous PCA, please provide variables used and information on their contribution towards variance explained.*

**Response:** this paragraph and figure were removed from the manuscript because the PCA does not provide useful insights or new information on the phytoplankton community or Dutch North Sea.

**Reviewer:** *Page 23: 6-7 – this paragraph is not clear particularly what is meant with 'opposite'*

**Response:** rephrased.

Figure 7 legend – Size of the open circles is a bit confusing and misleading as the reader may assume the size of the bubble refers to the amount of PP. Consider simplifying the figures and only plot productivity

**Response:** The figure was simplified as suggested.

**Reviewer:** *Page 24: 15 – please indicate how much of the variability in PP is explained by the stepwise regression (e.g. R2?).*

**Response:** because information on the nutrient availability was only available on a low-resolution spatial scale, the information provided by high resolution methods are not effectively used. To study the drivers of primary productivity another study design should have been chosen. Therefore, this analysis was deleted from the manuscript.

**Reviewer:** Page 25: 4 – reword please.

**Response:** rephrased

**Reviewer:** Page 26: 2-5 – require rewording particularly the need to clarify and be more specific on the work done in this study.

**Response:** removed from manuscript.

**Reviewer:** Page 26: 5 – this sentence may be misleading. The authors calculated PP along the sampling transects but did not provide an estimate for the wider Dutch North Sea as it may appear here.

**Response:** removed from manuscript.

**Reviewer:** *Page 26: 8 & 11 – timing of the bloom is discussed in this section however it would not be possible to define the start of the bloom based on a 4-day sampling per month. Continuous observations throughout the year by an instrument buoy or remote sensing would allow to 'contextualise' the measurements within the growing season (i.e. determine when sampling was carried out within the phytoplankton growing season).*

**Response:** Agreed and removed from manuscript.

**Reviewer:** *Page 26: 24-25 – please reword*

**Response:** rephrased

**Reviewer:** *Page 27: 8-9 – repetition of method; should be deleted.*

**Response:** Rephrased.

**Reviewer:** *Page 29: Figure 10 legend, possibly just my issue, I don't see the similarity between the two figures.*

**Response:** We do see a basic similarity, with the separation between the different water masses being reflected in our results. However, the similarity might not be striking enough to include the figure and therefore we leave it out of the manuscript.

**Reviewer:** *Page 30: 13 – 'low resolution': should this be 'high-resolution'?*

**Response:** no, we meant to say low-resolution. We rephrased to make it easier to follow: "Extra low-resolution sampling points in clearly deviating areas would be useful, because only low-resolution offer the level of detail which is required to identify toxic, keystone or invasive species."

**References**

Demšar, U., Harris, P., Brunsdon, C., Fotheringham, A. S., & McLoone, S. (2013). Principal Component Analysis on Spatial Data: An Overview. Annals of the Association of American Geographers, 103(1), 106–128. https://doi.org/10.1080/00045608.2012.689236

Rousseeuw, K., Poison Caillault, E., Lefebvre, A., & Hamad, D. (2015). Achimer Hybrid hidden Markov model for marine environment monitoring. IEEE JOURNAL OF SELECTED TOPICS IN APPLIED EARTH OBSERVATIONS AND REMOTE SENSING, 8(1), 204–213. https://doi.org/http://dx.doi.org/10.1109/JSTARS.2014.2341219

Suggett, D. J., C. M. Moore, A. E. Hickman, and R. J. Geider. (2009b). Interpretation of fast repetition rate (FRR) fluorescence: signatures of phytoplankton community structure versus physiological state. Marine Ecology-Progress Series **376**:1-19.

**Response to interactive comment of anonymous referee #2**

By Hedy M. Aardema in agreement with co-authors.

**Reviewer:** *This paper analyses spatial and temporal patterns in cruise data with 3 high-resolution monitoring methods: FRRF, Flow-cytometry and Ferrybox. Correlations between the observed variables are also analysed. The large dataset, including many phytoplankton and environmental variables observed together enables the authors to understand the patterns in the various phytoplankton variables. The results could guide the optimal application of such novel monitoring methods in operational monitoring for a.o. MSFD.*

**Response**: We thank the reviewer for the helpful and critical comments. We rewrote and restructured the manuscript extensively based on these comments. We are happy to hear that the reviewer sees the potential of our applied methods.

**General comments**

**Reviewer:** *The paper lacks a clearly stated research question or hypothesis to be tested. Therefore, it is unclear what is the purpose of the various analyses performed and what we can learn from the results. Based on the conclusion that this type of "high-resolution is a very useful supplement to current monitoring", I would expect a hypothesis such as "combined high-resolution monitoring of many phytoplankton variables along with environmental variables allows us to quantify seasonal and meso-scale patterns in phytoplankton biomass, species composition and primary production. The concurrent measurement of different phytoplankton variables allows us to understand the effect of phytoplankton species composition and physiological adaptation processes on the observed patterns in phytoplankton biomass and production". Then the analysis should show how the variables should be combined to provide the most reliable estimates of phytoplankton biomass and primary production.*

**Response**: Because of the exploratory nature of our research, a hypothesis was not defined. The addition of the suggested sentences does help in making the manuscript easier to follow. We therefore adopted part of the sentences and added of the following sentences to the introduction: "The aim of this study is to test the suitability of these two high-resolution methods to be developed as novel phytoplankton monitoring method. The two high-resolution methods, a flowcytometer and a FRR fluorometer, were deployed concurrently on four 4-day cruises in April, May, June and August to meet a wide range of environmental conditions and phytoplankton community states. These measurements allow for quantification of seasonal and mesoscale spatial patterns in phytoplankton abundance, photophysiology and gross primary production. In this paper we provide an overview of the acquired results, use a spectral cluster analysis to visualize spatial heterogeneity and evaluate the potential of these methods to optimize current monitoring programs."

**Reviewer:** *There are many observed variables, which are not consistently named in the text, figures and tables. Therefore, it is easy to get lost in the description of patterns for all individual variables. A clear definition of variables that is consistently used throughout the text would help the reader to understand the storyline. Some of the variables observed by the FRRF seem to be very similar. Which of the variables should be used as indicator and which are redundant to answer the research questions?*

**Response**: We corrected the inconsistent naming. The variables of the FRRf might seem similar under some conditions. However, because these variables vary depending on community composition and environmental conditions, they might deviate when conditions change (Sugget et al., 2009; Kromkamp and Forster, 2003). Therefore, care must be taken into choosing the parameters. For the current study the main interest is on monitoring the phytoplankton community, therefore we are interested in parameters that are informative on physiological adaptation or characteristic for phytoplankton taxons. Additionally, we focus on high resolution measurements, so limit the parameters to the ones attainable at high-resolution.

Based on these considerations we decided to include the current parameters, which give us a broad overview of the photophysiological status of the phytoplankton community.

**Reviewer:** *In the conclusions section a recommendation on next steps would be much appreciated: what would be required to use the high-resolution methods in scope to provide reliable estimates of phytoplankton biomass, production and species composition for long term monitoring? In the introduction and conclusion the species composition is defined in functional types such as nitrogen fixers, calcifiers or DMS-producers, but these do not correspond to the phytoplankton clusters used in this paper.*

**Response**: the conclusions were rewritten:
"A good monitoring program monitors the presence of functional types of phytoplankton, including the harmful taxons, the carrying capacity of the ecosystem and changes in biogeochemical cycling. The objective of this study was to evaluate the use of FRR fluorometry and flowcytometry for such monitoring purposes. The four conducted cruises spread over 5 months offered a wide variety of environmental conditions and phytoplankton community states, which the utilized methods were able to visualize. Inclusion of high-resolution methods in monitoring programs allows for analysis of finer scale events. Furthermore, it allows for analysis of living phytoplankton and is thereby able to measure rates and avoid effects of preservation and storage of samples. Another advantage is that high-resolution methods allows for easier comparison between countries, once common protocols have been established. Nevertheless, low resolution methods remain a necessity for more detailed taxonomic analysis, information on vertical heterogeneity, to calibrate and to correct for blanks. Data analysis might be the biggest bottleneck of the implementation of these high-resolution methods. The cluster analysis of flowcytometric data has high potential for improvement to increase the informative value of the method. Especially identification of phytoplankton clusters with a functional quality, such as nitrogen fixers, calcifiers or DMS-producers, would be helpful for interpretation of ecosystem dynamics and biogeochemical fluxes. Regarding the FRRf, the main challenge is converting electron transport rate to gross primary productivity in carbon units. Further research in these topics would benefit implementation of these methods into monitoring protocols. Furthermore, it is important to account for diurnal patterns in monitoring set-up to be able to distinguish between diurnal and spatial variability. Possibly the diurnal variability could be modelled, but more studies with a Langragian based approach would be needed for a better understanding of the impact of diurnal variability in the data. Overall, the in this study presented high-resolution measurement set-up has large potential to improve phytoplankton monitoring in supplement to existing low-resolution monitoring programs."

Specific comments

**Reviewer:** *Sentences are often long: consider breaking up in multiple sentences to improve readability.*

**Response**: We apologize for the difficulties and hope to have improved the readability in the new manuscript.

**Reviewer:** *Figure 1: please show only the stations (with names/ abbreviations) used in this study (see table S1) and the areas used in the text (such as Dogger Bank, Wadden, Den Helder, Rhine outflow) so the description of spatial patterns can also be understood by people that are not Dutch.*

**Response**: We updated the figure to the following:

[Figure]

**Reviewer:** *Section 2.2: please refer to international protocols/methods rather than internal protocols.*

**Response**: We added a more detailed description to the method section.

**Reviewer:** Table 1: it would help to have an additional column stating the interpretation / meaning of this variable, such as total biomass, nutrient stress, maximum growth rate, efficiency of light uptake etc. Then later in the text you can use these 'meaningful' names instead of codes, to facilitate understanding of observed patterns. Also a figure illustrating the meaning of the different variables (alfa,Ek, F', Fm' etc.) could prevent getting lost in all abbreviations.

**Response**: Unfortunately, the meaning of the different variables is usually not straightforward and dependent on multiple predictors (species, nutrient concentration, light availability etc.; Suggett et al., 2009). Nonetheless, we tried to make the table more information easier to understand.

**Reviewer:** *Equation 9: why did you use monthly averaged irradiance if you are looking at high-resolution patterns. Why did you not use irradiances measured during the cruise?*

**Response**: Unfortunately, we were unable to collect reliable irradiance data for all cruises. Clearly, it is preferable to have irradiance (PAR) continuously measured in parallel to the FRRF measurements when aiming to monitor current primary productivity.

**Reviewer:** *Table 2: Since you use both Length_FWS and O/R ratio as criteria to distinguish the phytoplankton groups, it would be logical to include a column for O/R ratio with the applied criteria.*

**Response**: Good idea, we added the O/R-ratio to the table.

**Reviewer:** *It is not entirely clear whether pico-red includes pico-Synecho or not. On page 14, line 30 it says: "Both groups of picophytoplankton (Synechococcus and total)", whereas table 2 and figure 3 suggest the two groups are exclusive.*

**Response**: Pico-red and Pico-synecho are two different groups, as correctly understood from table 2 and figure 3. We rephrased the sentence to: "Both groups of picophytoplankton (Synechococcus and Pico-red)", and scanned the manuscript for other mixing up.

**Reviewer:** *Section 2.4: please state with every type of analysis what is the purpose / research questions for that analysis. For example: what are you trying to predict from what and why?*

**Response**: We added the following the sentences to section 2.4: "To find regions with similar phytoplankton communities, data was spectrally clustered using the uHMM R package (Poisson-caillault and Ternynck, 2016) in the statistical software R (version 3.4.1, R Core Team, 2017)." and
"Principal Component Analyses (PCA) were performed to find which variables contributed most to the cluster results."

**Reviewer:** *Section 3.1: I don't see the value of comparing averages over whole transects (with large spatial variability, which is the subject of this paper), that are not even the same, between months. The only thing you see is seasonal patterns that are well-known from other studies and that can be summarized in section 2.1 in a description of the study area. Most of this section describes the data in table S1. I would replace table 3 with table S1 and remove table S2. N/P ratios address that same question as table S1, but with an indicator that is controversial.*

**Response**: For the authors the table helped to visualize the seasonal patterns, but we agree on the comment that this table does not add to the already existing knowledge on seasonal patterns. We therefore adopted the suggestion to replace the table with the table S1 from the supplementary material.

**Reviewer:** *The text in this section (and subsequent sections) is sometimes hard to follow as it is not clearly structured in time and space and variable. We go back and forth in time. Section 3.2 describes first figure 2, then figure 3 and then again figure 2 and then figure 3. I suggest to make one section about phytoplankton biomass (figure 2) and then one section on species composition (figure 3).*

**Response**: Rewritten

**Reviewer:** *Page 16, line 14: the southern coastal stations are more strongly affected by the Rhine outflow than the Scheldt outflows (see for example: Lacroix, G., Ruddick, K., Ozer, J., & Lancelot, C. (2004). Modelling the impact of the Scheldt and Rhine/Meuse plumes on the salinity distribution in Belgian waters (southern North Sea). Journal of Sea Research, 52(3), 149-163.).*

**Response**: We reformulated to Rhine and Scheldt river outflow.

**Reviewer:** *Figure 4: Please use consistent legends for the same variable between different months, with the same colour scheme and symbols (squares vs. circles) and with blue indicating low values and red indicating high values, so the high values stand out, more than the low values. Also captions in the table per line (red fluorescence, O/R ratio etc.) and per column (april, may etc.) would help to easier understand the figure.*

**Response**: We remade the figures, see manuscript.

**Reviewer:** *Section 3.5: I don't see the added value of this analysis. What does it tell us?*

**Response**: Agreed. We aimed to get a better understanding of the drivers of primary productivity in the Dutch North Sea. However, we realize now that the dataset is not very well suited for this and we therefore removed the analysis.

**Reviewer:** *Page 24: I suggest to mention in the table all the variables that were included in the analysis and note coefficients or 'ns' for not significant and the p values per explanatory variable. Then readers don't need to reconstruct the overview from the text. Actually, the significance test is likely not valid due to strong spatial autocorrelation in the data.*

**Response**: The Multiple Linear Regression was removed from the manuscript, because of the lack of information derived from it together with the abundance of literature already addressing the predictors of primary productivity.

**Reviewer:** *Discussion: Here I would expect to get some advice: How to best estimate phytoplankton biomass from these data? Should we use total red fluorescence (best R2) or F0 (least affected by NPQ)? Is there a way to combine both (with other available variables) to get an even better estimate?*

**Response**: We added the following paragraph to the discussion:

"Biomass might be one of the most important parameters to understand phytoplankton dynamics, but its direct measurement is not possible using high-resolution methods. Chlorophyll a concentration is often used as an estimate for biomass, although the Carbon:Chl a ratio is dependent on abiotic conditions and species-specific phenotypic plasticity (Flynn, 1991, 2005; Geider et al., 1997; Alvarez-Fernandez and Riegman, 2014; Halsey and Jones, 2015). Red fluorescence gave a good estimate of chlorophyll a concentration, both using the FRRf (adjusted $R^2$= 0.66) and FCM (adjusted $R^2$=0.90). Both the FRRf and the flowcytometer estimate the chlorophyll a concentration based upon the fluorescence in the red spectrum after excitation in the blue spectrum. There are some slight differences in the optics, the FRRf excites with a 450 nm LED and measures the fluorescence at $682 \pm 30$ nm, while the FCM excites at 488 nm and filters the red fluorescence over a longpass 650 nm filter towards the red fluorescence detector. The smaller detection range of the FRRf detector is optimized around the maximum emission of PSII and limits contamination by PSI (Franck et al., 2002; Oxborough et al., 2012). The second difference is the fluorescent state of the photosystems, the strong laser of the flowcytometer can only measure the maximum fluorescence (Fm), which is a parameter more prone to quenching than the minimum fluorescence measured by the FRRf. Yet, the biggest difference concerns the method; where the flowcytometer measures the fluorescence per particle, the FRRf does only a bulk measurement. In a bulk measurement other particles in solution scatter the excitation and emission photons, plus the emitted fluorescence of the phytoplankton is subject to reabsorption, especially at higher biomass densities. The latter seems to have the most impact on chlorophyll a concentrations, as the fit of the flowcytometer derived red fluorescence is a better than the FRRf minimum fluorescence. Other studies that use the FCM to estimate chlorophyll a concentrations also showed good relationships, but find better fits using the bulk measurements using a fluorimeter (Thyssen et al., 2015; Marrec et al., 2018). The conversion to biomass may also be done from cell abundances. Some studies use the oversimplified assumption that all cells have a spherical shape and a constant C content per biovolume (Tarran et al., 2006). With the scanning flowcytometer it is also possible to estimate biovolume based on scattering properties of the cell, but this relationship appears to be taxon specific (Rijkeboer, pers. comm.). This relationship will be further explored by comparing the calculated biovolume based on the Image in Flow pictures and the flowcytometric properties of these phytoplankters."

**Reviewer:** *Can we trust GPP from FRRF as a reliable estimate of primary production or is more work needed to achieve that goal? If so, what needs to be done?*

**Response**: We added the following paragraph to the discussion:

"The reliability of variable fluorescence as estimate of gross primary productivity is depending on many cell processes from the photon absorbance to carbon assimilation. The variable fluorescence reflects the first step of photosynthesis; the efficiency of which photons are captured and electrons produced and transferred. However, to interpret gross primary productivity in an ecological or biogeochemical meaningful way, the FRR units of electrons per unit time need to be converted to carbon units. Gross photosynthesis correlates well with photosynthetic oxygen evolution (Suggett et al., 2003), and multiple studies have shown good correlation between 14C-derived estimates of primary productivity and FRRf-derived estimates using a constant conversion factor (Melrose et al., 2006; Kromkamp et al., 2008). However, in reality this parameter is not a constant, as along the pathway from electron to carbon atom electrons are consumed by other cell processes (Flameling and Kromkamp, 1998; Halsey and Jones, 2015; Schuback et al., 2016). Therefore, a reliable GPP estimate in carbon units from FRR fluorometry requires more research and estimates provide relative rather than qualitative values. Despite its limitations the fact that the method can measure in situ, with relatively little phytoplankton manipulation before measurement, makes the method promising. Calibration with other methods, such as concurrent C14 of C13 incubations, could help to better understand the processes from electron excitation to carbon fixation. However, it should be recognized that these types of measurements come with their own problems, and measure something in between net and gross primary productivity depending on the incubation time and growth rate of the phytoplankton (Halsey and Jones, 2015). So it remains a question which method is measuring the 'real' primary productivity. Attempts to calculate primary productivity from flowcytometer data have also been made, which is actually based on the diurnal cycle in cell size caused by cell division (Marrec et al., 2018). Despite the limitations of GPP estimates by variable fluorescence, our results clearly show large spatial variability in gross primary production concurrent to

the expected strong variability during the growth season. This spatial heterogeneity is not fully captured by sampling at the standard low-resolution monitoring stations, showing the added value of our approach. Primary productivity was highest in April, and relatively large values were also observed offshore, indicating that a low phytoplankton biomass does not necessarily means that primary production is low. Our GPP rates were based on the same electron requirement for C-fixation ($\Phi e,C$). However, this is a likely oversimplification as $\Phi e,C$ is known to vary with abiotic conditions (Lawrenz et al., 2013) and the changes in nutrient conditions and temperature during the growth season are likely to affect GPP. This will be the topic of a future publication and we expect that the detection of several biogeographic regions will help us in predicting $\Phi e,C$."

**Reviewer:** *It is not really clear whether the diurnal variability in the FRRF variables is a problem that needs to be solved.*

**Response**: It is not so much a problem that needs to be solved, but it does need to be taken into account when setting up a monitoring program including FRRF variables. It is important to realize that measurements taken at different times of the day, might not be comparable. To be able to include FRRF variables in a long-term monitoring program, the included sampling points should be sampled at the same time of the day.

**Reviewer:** *Are the clusters in the FCM analysis the relevant ones to provide 'useful' information to science & society? Should we / Can we move on to other clusters that are mentioned in the conclusions?*

We added the following text to the discussion:
"To understand the role of the phytoplankton in biogeochemical cyckes, the FCM clusters would ideally reflect taxonomic or functional groups, as calcifiers, silicifiers, DMS producers (such as Phaeocystis) and nitrogen fixers (le Quéré et al., 2005). The lack of identification of distinct clusters makes this sofar impossible. Other studies manually separate up to 10 phytoplankton groups with the same instrument (Marrec et al., 2018). These groups included Prochlorococcus, which is at the absolute limit of resolving capacity of the FCM because of their small size and low fluorescence. They furthermore distinghuished the Pico-red in three groups based on FLO/FLR-ratio. Nano-cryptophytes group in high and low orange fluorescence and included a micro-eukaryotes group with a size from 10 to 20 uM. But these groups are still made up of many taxonomic genera and, apart from size, won't allow much for further interpretation of their role in the ecosystem or biogeochemical cycles. The same accounts for detection of nuisance phytoplankton; distinct clusters of toxic phytoplankton species are lacking. Although this will remain a challenge because toxicity in phytoplankton can differ within morphotypes and sometimes even differ per strain within a species (Tillman and Rick, 2003). But potentially, further research in flowcytometry can result in suspicious clusters to be flagged and further inspected by a specialist using microscopy. The potential is certainly there, as much of the information retrieved by the FCM is still unexplored; the clustering is performed on totals (area under the peak) instead of the pulse-shape. This in combination with more advanced camera options will need to further distinguish between groups in the future."

**Reviewer**: *Do the FCM data help to better understand the FRRF data (and vice versa)? For example, do we see diatoms under light limited conditions (high F'/Fm', high alfa, low Ek) and picoplankton under nutrient limited conditions (low F'/Fm')? Other ecological niches that we know from literature? Different conditions promoting Synechococcus compared to other picoplankton?*

**Response**: We tried to incorporate the link between the methods better, we added the following sentences to the manuscript:
"In this study a large part of the Dutch North Sea shifted from nutrient sufficiency to nutrient limitation between April and May, which was reflected in the low efficiency of PSII (Fv/Fm; Fig. 4). The Fv/Fm recovered between May and June, which suggest that the phytoplankton adapted to nutrient limiting conditions (Kruskopf and Flynn, 2005). However, photophysiological parameters are also varying per taxonomic group; smaller taxa typically have lower Fv/Fm values and higher σPSII values (Kolber et al., 1988; Suggett et al., 2009b). Indeed, by flowcytometry we find that the biggest shift in community composition took place between May and June from a nanophytoplankton dominated community to a picophytoplankton dominated community. These findings demonstrate how flowcytometry and fast repetition rate fluorometry can supplementary improve ecosystem understanding."

Technical comments

**Reviewer:**
- Collinear should be spelled with 2 ll's throughout the whole text.
- Page 9, line 4 & 5: I guess um means micrometers?
- Page 18, line 4: middle-right, please refer to the label C4 a-x.
- Figure 5: The figure would be easier to read if the colours per group are consistent between the cluster analysis on the right and the map on the left. Labels (A-D for April to August panels) would also help.
- Figure 9: Please add the hours of the day on the x-axis.
- Page 25, line 3: the word influenced is repeated too many times and therefor should get an e in the end.
- Page 28, line 11: estimates are.
- Line 13: parameter without s.
- Page 31, line 8: Jerico-next, without h.

**Response**: We adopted the suggested technical improvements.

Hedy M. Aardema[1,2], Machteld Rijkeboer[1], Alain Lefebvre[3], Arnold Veen[1], and Jacco C. Kromkamp[4]

[1]Laboratory for Hydrobiological Analysis, Rijkswaterstaat (RWS), Zuiderwagenplein 2, 8224 AD Lelystad, The Netherlands
[2]Department of Climate Geochemistry, Max Planck Institute for Chemistry, Hahn-Meitner-Weg 1, 55128 Mainz, Germany
[3]Ifremer, Laboratoire Environnement et Ressources, BP 699, 62321 Boulogne sur Mer, France
[3]Department of Estuarine and Delta Systems, NIOZ Royal Netherlands Institute for Sea Research and Utrecht University, P.O. box 140, 4400 AC Yerseke, The Netherlands

*Correspondence to*: Hedy M. Aardema (hedy.aardema@mpic.de)

**Abstract.** Marine waters can be highly heterogeneous both on a spatial and temporal scale, yet monitoring programs are currently  relying primarily on low- This study explores the potential of two high-to studyto studyto studybiomassMeasurementsconductedBoth FRRf and FCM data show spatial heterogeneity with monthly variation. Automated unsupervised Hidden Markov Model (uHMM) spatial clustering. Manual adjustments were necessary to optimize visualization of some distinct phytoplankton communities. Stepwise multiple linear regression (n=61) revealedthatphotophysiology (alpha), phytoplankton biomass (total red fluorescence) and abiotic predictors (Turbidity, DIN, time of the day and temperature) determined integrated water column gross primary productivity. Apart from spatial heterogeneity, the diurnal trend issignificant predictor exposing clear trends with other~~good representation of the spatial patterns, showing the added value of our approach. Still, to fully exploit the

potential of the tested high-resolution measurement set-up, some major improvements are to be made. Among which the most important are; accounting for the diurnal cycle in photophysiological parameters.  concurrent to the spatial variation, better predictions of the electron requirement for carbon fixation to estimate gross primary productivity, and the identification of more flowcytometer clusters with informative value. Nevertheless, already the richness of additional information provided by high-resolution methods such as the FCM and FRRf can improve existing low-resolution monitoring programs towards a more precise and  ecological assessment of the phytoplankton community and productivity.

[revised manuscript text omitted]

15  **Table 3: nutrient concentrations (µM) separated per month (April, May, June and August) and station. The stations are named according to name of the transects and the distance in kilometres from the coast (Fig. 1). Potentially limiting nutrient concentrations are shown in red (DIN<2 µmol L$^{-1}$, Si<1.8 µmol L$^{-1}$, $PO_4^{3-}$<0.5 µmol L$^{-1}$; Peeters and Peperzak et al, 1990). B.d: below detection limit.**

| | DIN (µM) | | | | | $PO_4^{3-}$ (µM) | | | | | Si (µM) | | | |
|---|---|---|---|---|---|---|---|---|---|---|---|---|---|---|
| Station | April | May | June | August | | April | May | June | August | | April | May | June | August |
| Walcheren 2 | 1.0 | 2.4 | 3.4 | 1.0 | | 0.2 | 0.2 | 0.4 | 0.6 | | 0.6 | 0.7 | 1.4 | 1.9 |
| Walcheren 20 | 1.2 | 3.1 | 1.1 | b.d | | 0.1 | 0.1 | 0.3 | 0.3 | | b.d | 2.7 | 0.5 | 2.0 |
| Walcheren 70 | 1.1 | 1.2 | 1.1 | b.d | | 0.2 | 0.2 | 0.2 | 0.1 | | b.d | 0.6 | 0.4 | 0.9 |
| | | | | | | | | | | | | | | |
| Noordwijk 2 | 37.5 | 21.7 | 4.9 | b.d | | 0.3 | 0.6 | 0.2 | 0.2 | | 6.7 | 3.5 | 0.8 | 1.2 |
| Noordwijk 10 | 28.5 | 15.0 | 3.1 | b.d | | 0.2 | 0.1 | 0.4 | 0.1 | | 2.9 | 3.2 | 0.7 | 1.4 |
| Noordwijk 20 | 21.6 | 4.9 | 0.9 | b.d | | 0.2 | 0.1 | 0.2 | 0.1 | | 1.3 | 0.7 | 0.8 | 0.6 |
| Noordwijk 70 | b.d | 1.0 | 0.9 | b.d | | 0.2 | 0.2 | 0.3 | 0.2 | | b.d | 1.1 | 1.7 | 0.1 |
| | - | - | - | - | | - | - | - | - | | - | - | - | - |
| Terschelling 10 | 10.1 | 1.9 | 0.9 | b.d | | 0.3 | 0.2 | 0.2 | 0.2 | | 3.0 | 2.4 | 0.5 | 0.7 |
| Terschelling 50 | 8.9 | b.d | 3.4 | 2.8 | | 0.5 | 0.2 | 0.2 | 0.3 | | 4.6 | 1.7 | 2.4 | 5.0 |
| Terschelling 100 | 12.6 | b.d | 1.9 | b.d | | 0.5 | 0.2 | 0.3 | 0.2 | | 3.9 | 0.5 | 1.1 | 1.7 |
| Terschelling 135 | 1.6 | 0.8 | 0.9 | b.d | | 0.4 | 0.1 | 0.1 | 0.3 | | 2.0 | 0.8 | 0.9 | 1.8 |
| Terschelling 175 | 0.9 | NA | 1.0 | b.d | | 0.2 | NA | 0.2 | 0.2 | | 0.6 | NA | 0.5 | b.d |
| Terschelling 235 | 1.0 | NA | 0.9 | b.d | | 0.2 | NA | 0.3 | 0.3 | | b.d | NA | 1.1 | 0.5 |

[Figure]

**Figure 2: linear regression of the natural logarithms of Chl *a* concentration in µg L$^{-1}$ as determined by HPLC (y-axis) and on the x-axis the natural logarithm of; FCM-derived total red fluorescence (in relative fluorensence units (RFU), left panel) and FRRf-derived minimum fluorescence (F$_0$ in RFU, right panel). Both FCM red fluorescence (p<0.01, adjusted R$^2$=0.90) and the FRRF F$_0$ (p<0.01, adjusted R2=0.66) are significant predictors for Chl *a* concentrations. The months (April, May, June and August) were a significant predictor of Chl a concentration for both the FRRf (p<0.05) and the FCM (p<0.01). The interaction between the x and y axis was only significant for the FCM data (p<0.05).**

**3.2 Phytoplankton**

 abundance  fluorescence ~~of all cells (hereafter TFLR). The FRRf also provides an estimate of Chl *a* based on the minimum fluorescence (F$_0$). Using cell count or fluorescence as predictor of phytoplankton presence yields contrasting results (Fig. 3), because of the wide range of phytoplankton cell sizes; microplankton have a substantially higher biomass, and thus fluorescence, per cell in comparison to picoplankton. So while the phytoplankton cell count is higher in June and August in comparison with April, the community in the former months consists of mainly pico-plankton which contribute little to total fluorescence resulting in a considerably lower fluorescence in June and August (Fig. 3). Fluorescence is therefore a better predictor of biomass or of Chl *a* concentration than cell counts, although Chl~~

High-resolution measurements of phytoplankton presence are based on either cell numbers (flowcytometers) or fluorescence (fluorometers, such as the "standard" chlorophyll sensors, FRRf, and some flowcytometers). Both parameters can yield contrasting results due to the wide range of phytoplankton cell sizes and species-specific Chl a content per cell (Falkowski and Kiefer, 1985; Kruskopf and Flynn, 2005). In this study this is clearly demonstrated by the higher phytoplankton average cell count in June in comparison to April, while the average fluorescence is higher in the latter (supplementary material; Fig. S1). This can be explained by the high relative abundance of pico-phytoplankton, which contributes little to total fluorescence.

Both the FRRf and FCM provide significant predictors of HPLC-derived Chl *a* concentration (Fig. 2). When performing an ANCOVA with month as factorial predictor, natural logarithm transformations were necessary because of the highly unequal variances between months. The ANCOVA with the FRRf-derived $F_0$ as Chl *a* predictor revealed that Chl *a* concentrations significantly differed per month ($p<0.01$) but not the slope, and that $F_0$ was a significant predictor ($p<0.01$) of Chl *a* concentration (adjusted $R^2=0.66$). Yet, the FCM estimate of Chl *a* concentration (TFLR) was a better predictor ($p<0.01$) with an adjusted $R^2$ of 0.90. The ANCOVA with the FCM-derived TFLR as Chl *a* predictor resulted not only in a significant difference of the Chl *a* concentration per month ($p<0.01$) but also in a significantly different slope ($p<0.05$), suggesting that ~~abiotic factors and phytoplankton community composition are influencing the amount of fluorescence per Chl *a* molecule (Fig. 2). In April and August the slope is steeper in comparison to May and June. An explanation could be the package effect (Dubinsky et al., 1986), where stacking of Chl *a* at low light intensities causes a shading effect within the cells and a steeper slope of Chl *a* concentration with *in vivo* fluorescence. Because there is a lack of agreement in photophysiological parameters between April and August, it is likely that the months do not have the same drivers for the steeper slope. In April the high $\sigma_{PSII}$ coincides with high $n_{PSII}$, suggesting that although the chlorophyll self shades it does not result in a lower absorption cross section because of the high amount of RCIIs in relation to Chl *a* molecules. In contrast, in August the phytoplankton community is nutrient limited and has a higher $\sigma_{PSII}$, in correspondence with results obtained by Kolber et al. (1988) who observed that nutrient limitation increases $\sigma_{PSII}$. Self-shading in this month is more likely a result of smaller cell size as indicated by the higher abundance of picoplankton (Table 3; Geider et al., 1986~~other predictors that differ per month are influencing the amount of fluorescence per Chl *a* molecule (Fig. 2).

Chl *a* concentration is a limited predictor of biomass because the Chl *a* concentration per cell is species-specific and subject to phenotypic acclimation to abiotic conditions (Falkowski and Kiefer, 1985; Kruskopf and Flynn, 2005). Therefore, the FRRf yields other biomass related proxies next to the minimum fluorescence, that allow for circumvention of the use of chlorophyll *a* to estimate primary productivity (Oxborough et al., 2012). These parameters are the total absorption coefficient in the water ($a_{LHII}$ in $m^{-1}$) based on the absorption of the photosynthetic pigments  associated with PSII and the amount of PSII reaction centres per volume ([RCII] in nmol RCII $m^{-3}$). Both are very strongly correlated to $F_0$ , although the ratio of RCII

to $a_{LHII}$ can vary by nature, affecting $n_{PSII}$. However, these three biomass related proxies show a perfect relationship to each other (Supplementary material). The minimum fluorescence measured with the FRRf ($F_0$) is related to the red fluorescence mearured with the FCM (TFLR; r=0.7). Interestingly,TFLR and $F_0$ are not correlated to total orange fluorescence, which indicates that the cyanobacterial picoplankton is not a fixed proportion of the total phytoplankton biomass (Supplementary material).; Fig. S3).

A pairplot analysis of the combined data of all cruises shows that some photosynthetic parameters are highly correlated (Supplementary material). The correlation of alpha and $F_v/F_m$, indicators for photosynthetic affinitiy and photosynthetic efficiency, are perfectly correlated (r=1). The parameters derived from the PE curve also show high correlation, being dependent on the light acclimation state of the phytoplankton trends in the maximum electron transport rate ($P_{max}$) and the light saturation parameter ($E_k$) are similar. Surprisingly, alpha does not show any correlation with Ek, suggesting that the light affinity is not dependent on the level of irradiance where the PSII reaction centres become saturated, or that its value is also affected by nutrient limitation, obscuring a relationship with $E_k$.The effective absorption cross section per photosystem, $\sigma_{PSII}$, is very strongly negatively related to $n_{PSII}$ (r= 0.9). This is to be expected; the larger $n_{PSII}$, the smaller the number of pigment molecules associated with it.

Community composition is variable over the months with the biggest shift in community composition between May and June (Fig. 3). In May mean $F_v/F_m$ is low (0.26 ± 0.09), suggesting physiological stress (Table 3). This coincides with a shift of nutrient sufficiency in the largest part of the sample region (with only potentially limiting conditions in the most southerly part of the Dutch North Sea and further offshore) in April to a larger region of nutrient limitation in May. In June the mean $F_v/F_m$ recovers and community composition changes. Both groups of picophytoplankton (*Synechococcus* and total) increase in relative abundance between May and June, while the nanophytoplankton shows a strong decrease (Fig. 3). Because the picophytoplankton fraction makes up for only a small part of the biomass, the microphytoplankton is the largest contributor to red fluorescence in June, although this group does not increase in relative abundance in comparison to May. After June the microplankton disappears, leaving 80% of the average community composition to picoplankton. The shift to smaller cell sizes at nutrient limiting conditions is not surprising because of the higher nutrient affinity of smaller cells.

[Figure]

**Figure 3: Phytoplankton abundance per phytoplankton group distinghuished with the flowcytometer, shown as average (relative) abundance per month (left), total red fluorescence(middle) and total orange fluorescence (TFLO; right).The upper graphs are absolute and lower graphs relative.**

**3.3 Spatial distributions**

Both the biomass concentration and the phytoplankton community composition, expressed as percentage of the total cell numbers, showed a dynamic picture (Fig. 4). In all cases, microphytoplankton < 10% of the total cell counts, although in terms of biomass they sometimes dominate (Fig. 3).

**In April, high biomass concentrations (using TFLR) are observed close to the Dutch Delta in the South of the Dutch EEZ and west to the island of Texel and Vlieland. Very low concentrations are found offshore, especially in the more central part towards the Doggersbank area. The north-western wedge of the Dutch North Sea was3.3 Phytoplankton community composition**

Both cell numbers and the phytoplankton community composition showed high spatial heterogeneity in the Dutch North Sea in the sampled months (Fig. 3). In cell numbers, the pico-red group was always present as the dominating group. Because of their low total biovolume, they were contributing less to total red fluorescence. The relative abundance of picophytoplankton was generally higher offshore and in the northern part of the Dutch North Sea. The pico-*Synechococcus* group showed a strong numerical presence offshore in April and in most of the Dutch North Sea in June. The nano-red group was often a dominant group, both in sense of cell numbers as contribution to total red fluorescence. The nano-cryptophytes were never abundant in cell numbers, but contributed to the total red fluorescence in the northern offshore regions. The microphytoplankton group had

a low numerical abundance and represented always less than 10% of the total cell counts. Yet in terms of red fluorescence they sometimes dominate, which occurred most frequently in coastal regions (Fig. 3).

[revised manuscript text omitted]

In May photophysiological parameters of the phytoplankton communities in the Dutch North Sea were strongly heterogeneous with only smaller scale spatial patterns (Fig. 4b,f,j). $F_v/F_m$ was in general much lower in May (0.1-0.5) than in April (>0.4) across most of the Dutch EEZ (Fig. 4b). The range in $\sigma_{PSII}$ was larger in May in comparison to April (Fig. 4f). The $\sigma_{PSII}$ was also higher across the Dutch North Sea, except from a small area near the coast of Noordwijk. A possible consequence of the outflow of the Rhine River. In the same region the $E_k$ is high (> 450 μmol photons m$^{-2}$ s$^{-1}$), but in other regions where $E_k$ is high this does not coincide with an increased $\sigma_{PSII}$. The $E_k$ across the Dutch North Sea in May is heterogeneous without large-scale spatial patterns.

In June the photophysiology of the phytoplankton in the Dutch North Sea is still as heterogeneous as in May, but larger scale spatial patterns seem to occur. The $F_v/F_m$ values recovered to above 0.4 in the coastal zone, but not in offshore regions in the Southern North Sea. The $F_v/F_m$ of the southern offshore phytoplankton, between Walcheren 70 and Noordwijk 70 (Fig. 1), remained lowest (<0.2; Fig. 4c). The $\sigma_{PSII}$ was lower than in May, apart from the southern offshore region that remained higher (Fig. 4g). In a small region around Noordwijk 70 the phytoplankton community had a particularly low $\sigma_{PSII}$ (<2.5 nm$^2$ PSII$^{-1}$) which did not present itself in anomalies in the other photophysiological parameters. The $E_k$ in May was low in the Northern coastal zone and higher in offshore regions (Fig. 4k).

In August the $F_v/F_m$ recovered across the Dutch North Sea (Fig. 4d). The $\sigma_{PSII}$ was high in northern offshore region, and comparable to June in the rest of the Dutch North Sea (Fig. 4h). The $E_k$ shows some interesting variability in August. The regions off the Noordwijk coast and the of the Wadden Island coast were sampled twice, on two different times. These double

measurements resulted in strongly different $E_k$, suggesting that time is a more important predictor in comparison to spatial variability.

To further investigate possible daily patterns we calculated standardized daily anomalies (z-scores). These show a clear diurnal trend in photosynthetic activity (Fig. 5). $F_v/F_m$ is lowest during the middle of the day, while Ek, $\sigma_{PSII}$ and $1/\tau$ peak during the day. As $E_k$ is strongly correlated to $P_{max}$ (Fig. S3); a clear diurnal pattern is also present in the photosynthetic electron transport rate.

[Figure]

**Figure 4: Maps of the photophysiological parameters Fv/Fm (a-d), $\sigma_{PSII}$ (e-h; in nm$^2$ PSII$^{-1}$) and $E_k$ (i-l; in µmol photons m$^{-2}$ s$^{-1}$) per month (from left to right: April, May, June and August). For more details on the location see Fig. 1.**

[Figure]

**Figure 5: Standardized daily anomalies (z-scores) of $F_v/F_m$, $E_k$, $\sigma_{PSII}$ and $1/\tau$ showing the diurnal trends in photophysiological data. On the x-axis the time of the day and on the y-axis the z-score.**

**3.5 Gross primary productivity**

Gross primary productivity ranged from minimum 0.35 µg C L$^{-1}$ h$^{-1}$ in June to peak productivities of 602 µg C L$^{-1}$ h$^{-1}$ in the coastal zone in May (Fig. 6). The average GPP was highest in April and lowest in August. Monthly averages ranged from 116 ± 59 µg C L$^{-1}$ h$^{-1}$ in April and 8.7 ± 8.3 µg C L$^{-1}$ h$^{-1}$ in August, although these averages are not completely comparable due to different ship routes per month (Fig. 6). In April spatial heterogeneity in GPP was low. Highest rates in April were measured offshore (> 250 µg C L$^{-1}$ h$^{-1}$) and in the coastal regions close to the Wadden Islands (Terschelling 10 in Fig. 1). In May, the GPP is heterogeneous without clear spatial pattern. Most production rates stay below 30 µg C L$^{-1}$ h$^{-1}$, with local GPP peak rates over 600 µg C L$^{-1}$ h$^{-1}$ in the southern coastal zone. In June the Dutch North Sea was on average lower than in May, and showed slightly more large-scale spatial patterning. Highest values in June were observed (30-40 µg C L$^{-1}$ h$^{-1}$) northwest of Noordwijk. In August GPP was low throughout the Dutch North Sea with the majority of water-column productivity rates staying below 10 µg C L$^{-1}$ h$^{-1}$. In the southern coastal zone slightly higher rates were found, reaching up to 50 µg C L$^{-1}$ h$^{-1}$.

[Figure]

**Figure 6: Gross primary productivity of the surface (a-d; in µg C L⁻¹ h¹)  per month (from left to right: April, May, June and August). Colors represent rates, where blue is low and red is high (see legend).**

**3.5 Spatial clustering**

Strong collinearity between measured parameters was present. For spatial clustering these were removed based on the variable inflation factor (VIF>6; see supplementary material for pairplots), which resulted in removal of the photophysiological parameters $P_{max}$, α, $a_{LHII}$, $n_{PSII}$, the FCM-parameter of the total red fluorescence and the GPP. From the five defined phytoplankton groups (Table 2), the nano-crypto group was not used in the clustering because of collinearity (VIF>6). The remaining variables were the abundance of the remaining four FCM-defined phytoplankton groups (Pico-Red, Pico-Synecho, Nano-Red and Micro-Red), the total O/R ratio and five photophysiological parameters ($F_v/F_m$, $σ_{PSII}$, $1/τ$, [RCII], and $E_k$). For an overview of the collinearity between variables see the pairplots in the supplementary material.

Spectral cluster analysis resulted in identification of two to four clusters in each cruise. Most of these clusters were spatially separated and can therefore be seen as regions with distinct phytoplankton communities (Fig. 7). In April the clustering resulted in three clusters with a clear spatial pattern. In the PCA the variables that contributed most to the first principal component were all biomass related; [RCII] and $a_{LHII}$, related to the photosynthetic capacity per reaction center and per volume, and the abundance of the Nano-red group. The second principal component has photosynthetic parameters as two main contributors

[Figure]

| | PC1 | PC2 |
|---|---|---|
| $\sigma_{PSII}$ | 0.8 | **28.8** |
| $F_v/F_m$ | 13.7 | 0.6 |
| $a_{LHII}$ | **18.7** | 3.4 |
| [RCII] | 17.1 | 6.6 |
| $1/\tau$ | 9.8 | **22.7** |
| $E_k$ | 3.9 | 13.8 |
| Pico-red | 4.2 | **15.1** |
| Nano-red | **16.9** | 0.0 |
| Micro-red | 10.5 | 4.5 |
| *Synechococcus* | 4.3 | 4.4 |
| Variance explained (%): | 45.6 | 19.3 |

| | PC1 | PC2 |
|---|---|---|
| $\sigma_{PSII}$ | 0.1 | **36.7** |
| $F_v/F_m$ | 0.8 | **14.5** |
| $a_{LHII}$ | 17.5 | 6.7 |
| [RCII] | **20.4** | 1.6 |
| $1/\tau$ | 4.4 | 7.5 |
| $E_k$ | 0.7 | **26.3** |
| Pico-red | **18.5** | 0.4 |
| Nano-red | **21.1** | 0.6 |
| Micro-red | 16.4 | 1.4 |
| *Synechococcus* | 0.0 | 4.3 |
| Variance explained (%): | 42.5 | 18.9 |

**Formatted Table**

[Figure]

| | PC1 | PC2 |
|---|---|---|

[revised manuscript text omitted]
 separtionis presentis consisting of mainly micro phytoplanktonlow light saturation levelphotosynthetic affinity (α)~~.

$a_{LHII}$ and [RCII]. The four clusters identified in August are spatially separated, but with some complications (Fig. ~~5). The first phytoplankton community, which covers the most Northern most part of the Dutch North Sea and the coastal region of the Northern part of the Netherlands and the offshore region of Noordwijk, were characterized by high effective absorption cross section and rate of reopening of closed RCIIs (1/τ). The second phytoplankton community is corresponding to the southern coastal regions, a region with high freshwater influx, and is positively associated with most phytoplankton groups and the amount of RCII's per volume. The third phytoplankton community is characterized by low light saturation level and high photosynthetic efficiency and low rate of reopening of closed RCIIs (1/τ). Finally, the fourth spatial cluster was also found in April and May; in August it is a more a variable group of phytoplankton and the northern coastal region expands more to the south.north easternthe thirdThe third clusternight timelight saturation levelsuggestingthismorethan~~cluster instead of a spatial cluster.

[Figure]

[Figure]

[revised manuscript text omitted]
 was to supplement low-resolution monitoring. Multiple investigate spatial and seasonal patterns in photophysiological parameters and phytoplankton community composition with high spatial resolution. If successful, the method employed here can be further developed as novel monitoring method to improve existing monitoring programmes towards a more precise and ecosystemic ecological assessment (OSPAR,

15    MSFD).

Previous studies found that the strong seasonal dynamics in the Dutch North Sea affect the spatial distribution and community composition of the phytoplankton community (Baretta-Bekker et al., 2009; Brandsma et al., 2011). provide an overview of current The high resolution methods used in this study, the FRRf and FCM, were able to visualize this spatial and seasonal variability of the phytoplankton community in the Dutch North Sea in a supplementary way. The typical spring bloom was partly captured by the cruise of April; photophysiology was uniform and primary productivity high. Between April and May, the efficiency of PSII ($F_v/F_m$; Fig. 4) decreased throughout the Dutch North Sea. A decreasing $F_v/F_m$ is generally associated with limiting nutrient conditions. Spatial clustering may serve as an example of how to use multiple high-resolution or other abiotic stressors (Suggett et al., 2009b; Kolber et al. 1988; Kolber and Falkowski, 1993; Beardall et al. 2001; Ly et al. 2014), but can also reflect a change in community composition. Photophysiological parameters in a monitoring program. Lastly, the water column integrated gross primary productivity of the Dutch North Sea is estimated and its main forcing factors are vary per taxonomic group; smaller taxa typically have lower $F_v/F_m$ values and higher $\sigma_{PSII}$ values (Kolber et al., 1988; Suggett et al., 2009b). No major shift in community composition was identified. by flowcytometry between April and May. This suggests that an abiotic stressor, such as the nutrient limiting conditions in a large part of the Dutch North Sea, instead of the community composition was driving the decrease in efficiency of PSII. In contrast, the recovery of the $F_v/F_m$ between May and June did coincide with a shift in community composition. In May the phytoplankton communities were mostly nanophytoplankton-dominated, while in June the communities were dominated by picophytoplankton (offshore) and microphytoplankton (coastal). So, although a recovery of the $F_v/F_m$ can also occur as adaptation of the phytoplankton to nutrient limiting conditions (Kruskopf and Flynn, 2005), it seems that the shift in community composition was the major driver for the recovery of the $F_v/F_m$ between May and June. These findings are a good example of how concurrent measurements by flowcytometry and fast repetition rate fluorometry can supplementary improve ecosystem understanding.

Environmental conditions in the Dutch North Sea are spatially heterogeneous and strongly influenced by seasonal dynamics. The timing of the phytoplankton bloom period corresponds well to the study of Baretta-Bekker et al. (2009) on phytoplankton dynamics in the Dutch North Sea from 1991 to 2005. In April we covered a phytoplankton bloom period, typified by high biomass concentrations, high quantum efficiencies and electron transport rates of PSII. The May cruise covered the collapse of the phytoplankton bloom period, as shown by the lower average quantum efficiency of PSII and high variability in biomass concentrations (Table 3). The cruises in June and August covered a low Chl *a* period and a second late summer bloom period was not detected. Although a second bloom period is known to occur in some regions of the Dutch North Sea, an onset later than August is not unusual (Baretta-BekkerThe identification of only 5 distinct phytoplankton groups by flowcytometry has limited informative value. Yet, size distribution does affect the carrying capacity of the ecosystem, microphytoplankton are a food source for higher trophic levels than picophytoplankton. Picophytoplankton is part of the microbial food web, with less trophic efficiency and low contribution to carbon export (Azam et al., 1983; Finkel et al., 2010). The shift from nanophytoplankton-dominated communities in April to picophytoplanktondominated communities in August, therefore implicates that over the season the tropic efficiency and carbon export decrease. To increase the informational value of the flowcytometry data beyond size, the FCM clusters would need to reflect taxonomic or functionally relevant groups. Interesting groups include calcifiers, silicifiers, DMS producers (such as *Phaeocystis*) or nitrogen fixers (le Quéré et al., 2005). The lack of identification of distinct clusters makes this sofar impossible. Marrec et al. (2018) manually separate up to 10 phytoplankton groups from the data of the Cytosense flowcytometer. Yet, most of these groups still comprise many taxonomic genera and, apart from size, do not allow much for further interpretation of their role in the ecosystem or biogeochemical cycles. Also for detection of nuisance phytoplankton, distinct clusters of toxic species are lacking. Yet, toxicity in phytoplankton can differ even between strains within one species, so finding a distinct cluster by flowcytometry is problematic (Tillman and Rick, 2003). However, much of the information retrieved by the FCM is still unexplored; the clustering is performed on totals (area under the peak) instead of the entire pulse-shape. Identification of 'suspicious' clusters with potential toxic species could be helpful. These suspicious clusters can flag sampling points to be further inspected by a specialist using microscopy. Combination of flowcytometry with an Image-in-flow camera may open up the possibility to identify groups with more informative value.

Biomass might be one of the most important parameters to understand phytoplankton dynamics, but its direct measurement is not possible using high-resolution methods. Chlorophyll *a* concentration is often used as an estimate for biomass, although the Carbon:Chl *a* ratio is dependent on abiotic conditions and species-specific phenotypic plasticity (Flynn, 1991, 2005; Geider et al., 1997; Alvarez-Fernandez and Riegman, 2014; Halsey and Jones, 2015). In this study, chlorophyll concentrations was estimated by red fluorescence, which resulted in a good fit both using the FRRf (adjusted $R^2 = 0.66$) and FCM (adjusted $R^2 = 0.90$). Both the FRRf and the flowcytometer estimate the chlorophyll a concentration based upon the fluorescence in the red spectrum after excitation in the blue spectrum. There are some slight differences in the optics, the FRRf excites with a 450 nm LED and measures the fluorescence at $682 \pm 30$ nm, while the FCM excites at 488 nm and filters the red fluorescence over a longpass 650 nm filter towards the red fluorescence detector. The smaller detection range of the FRRf detector is optimized around the maximum emission of PSII and limits contamination by PSI (Franck et al., 2002; Oxborough et al., 2012). The second difference is the fluorescent state of the photosystems, the strong laser of the flowcytometer can only measure the maximum fluorescence ($F_m$), which is a parameter more prone to quenching than the minimum fluorescence measured by the FRRf. Yet, the biggest difference concerns the method; where the flowcytometer measures the fluorescence per particle, the FRRf does only a bulk measurement. In a bulk measurement other particles in solution scatter the excitation and emission photons, plus the emitted fluorescence of the phytoplankton is subject to reabsorption, especially at higher biomass densities. The latter seems to have the most impact on chlorophyll *a* concentrations, as the fit of the flowcytometer derived red fluorescence is a better than the FRRf minimum fluorescence. Other studies that use the FCM to estimate chlorophyll *a* concentrations also showed good relationships, but find better fits using the bulk measurements using a fluorimeter (Thyssen et al., 2015; Marrec et al., 2018). An alternative to the controversial use of chlorophyll *a* as estimation for biomass is the biomass estimation from cell abundances. Although this requires assumptions on cell shape and a constant C content per

biovolume (Tarran et al., 2006). Yet another alternative to explore is to estimate biovolume based on scattering properties of the cell using a pulse shape recording flowcytometer. This relationship appears to be taxon specific (Rijkeboer, pers. comm.) and needs to be further explored by comparison of calculated biovolume (based on the Image in Flow pictures) and the flowcytometric properties of the cell.

Phytoplankton biomass does not necessarily reflect primary productivity, as high grazing pressure can keep biomass low while production is high. This is clearly visualized by the lack of resemblance between patterns in cell numbers (Fig. 3 a-d) and gross primary productivity (Fig. 6). The reliability of variable fluorescence as estimate of , et al., 2008). Generally, pico-autotrophs contributed considerably to cell numbers but covered only a small fraction of the total biomass (Fig. 3). As nutrient limitation progressed from April to August, the relative abundance of picoplankton reached over 80%, which corresponded to less than 30% of the relative fluorescence. In June and August the molar nutrient N:P ratios were generally below the Redfield ratio and concentrations were in the limiting range, suggesting that phytoplankton populations were N-limited in a large part of the Dutch North Sea. This impacts the community composition: generally it is assumed that nutrient limitation favours small cell size, because of the higher surface to volume ratio of smaller cells, and that fluctuating nutrient concentrations favour larger cells due to their greater maximum uptake rate and storage capacity (Stolte and Riegman, 1995; Giannini and Ciotti, 2016; Philippart et al., 2000), and the shift towards smaller species observed by us using FCM is thus in accordance with this theory. The change in community composition over the season has implications for the whole ecosystem, because microphytoplankton is a better food source for higher trophic levels than picophytoplankton, which is more involved in the microbial food web, with less trophic efficiency and low contribution to carbon export (Quere et al., 2005). Nutrient limitation does not only affect community cell size but also low values of $F_v/F_m$ are often related to nutrient limitation (Kolber et al. 1988, Kolber and Falkowski 1993, Beardall et al. 2001, Ly et al. 2014), although this is not always the case, and it seems likely that after acclimation to limiting nutrient conditions $F_v/F_m$ can recover again as was seen in the current study in June (see also Kruskopf and Flynn, 2006). The $\sigma_{PSII}$ was negatively associated with DIN and turbidity in the PCA on the low resolution data (Fig. 8), and although this value is assumed to vary per taxonomic group, it is not associated with any flowcytometer group (Kolber et al., 1988; Suggett et al., 2009), hence most of the variability seems to be driven by light and nutrient conditions. The values for the effective absorption cross section are slightly lower but in similar range to other studies (Suggett et al., 2009).

The gross primary productivity as found in the current study was both spatially and temporally variable. Average surface productivity of 44 ± 64 µg C L⁻¹ h⁻¹ and peak primary productivity in April and lower values the rest of the year is in agreement with earlier studies in the North Sea coastal zone (is depending on many cell processes from the photon absorbance to carbon assimilation. The variable fluorescence reflects the first step of photosynthesis; the efficiency of which photons are captured and electrons produced and transferred. However, to Brandsma et al., 2011). To interpret water column integrated gross primary productivity in an ecological or biogeochemical meaningful way, the FRR units of electrons per unit

time need to be converted to carbon units. Gross photosynthesis correlates well with photosynthetic oxygen evolution (Suggett et al., 2003), and multiple studies have shown good correlation between $^{14}$C-derived estimates of primary productivity and FRRf-derived estimates using a constant conversion factor (Melrose et al., 2006; Kromkamp et al., 2008). However, in reality this parameter is not a constant, as along the pathway from electron to carbon atom  electrons are consumed by other cell processes (Flameling and Kromkamp, 1998; Halsey and Jones, 2015; Schuback et al., 2016). Therefore, a reliable GPP estimate in carbon units from FRR fluorometry requires more research and estimates provide relative rather than qualitative values. Despite its limitations the fact that the method can measure *in situ*, with relatively little phytoplankton manipulation before measurement, makes the method promising. Calibration with other methods, such as concurrent C14 of C13 incubations, could help to better understand the processes from electron excitation to carbon fixation. However, it should be recognized that these methods introduce other uncertainties; they measure something in between net and gross primary productivity, depending on the incubation time and growth rate of the phytoplankton (Halsey and Jones, 2015). Thus, which method is measuring the 'true' primary productivity remains controversial and should be interpreted with care.

When including photophysiology (or photophysiology based GPP estimates) in a monitoring program, it is critical to consider diurnal variability. Diurnal trends make extrapolation of rates obtained at a specific timepoint to daily rates difficult. Most photophysiological parameters we measured showed diurnal trends (Fig. 5). The diurnal trend is dictated by the phytoplankton cell cycle, a circadian oscillator and photophysiological response to varying irradiance (Suzuki and Johnson, 2001; Cohen and Golden, 2015; Schuback et al., 2016). Phytoplankton use photophysiological plasticity to minimize photodamage and optimize growth under fluctuating irradiance (Schuback et al., 2016; Behrenfeld et al., 2002).  The electron requirement for carbon fixation  is also subject to diurnal variation (Schuback et al., 2016; Lawrenz et al., 2013; Raateoja, 2004). ~~As we passed several "biogeochemical" provinces, indicated by the cluster analysis, it is difficult to separate diurnal variability from variability introduced by phytoplankton in different biogeochemical areas. However, as we observed this diurnal variability also in the same clusters on a number of occasions, it is clear that diurnal variability is inherent in our analysis. For future studies it is advised to include Langragian based approach where the same phytoplankton community can be followed during a complete light-dark cycle. The diurnal trend in coupling of electron flux and carbon fixation is dictated by cell cycle, a circadian oscillator and irradiance, and photophysiological plasticity minimizes photodamage and optimizes growth under fluctuating light and nutrient concentrations (Claquin et al., 2014; Cohen and Golden, 2015; Schuback et al., 2016).reliable estimate of gross primary productivity, Schuback et al. (2016) suggest a correction with NPQ which needs further research in order to get a~~

more reliable GPP value. The presence of non-photochemical quenching (NPQ) makes the interpretation of most photophysiological parameters complicated in our study because the lack of dark acclimation time decreases the comparability between samples. Most photophysiological parameters we measured showed diurnal trends, although, but as said, this is likely not only due to NPQ but also to phytoplankton cell cycle (Claquin et al., 2014; Schuback et al., 2016) and rhythms

5  driven by a circadian oscillator (Cohen and Golden, 2015). But although the presence of NPQ compromises the use of fluorescence as estimate for chlorophyll concentrations, the good relationship between the HPLC-derived Chl a concentration and fluorescence of the FRRf and FCM suggest that most of the NPQ is dissipated during the time in the tubing and low light acclimation. The clear diurnal trends we observed are in agreement with previous studies and is usually explained by photophysiological plasticity to minimize photodamage (Schuback et al., more reliable estimate of

10  gross primary productivity, Schuback et al. (2016) suggest a correction with normalized Stern-Volmer quenching (NPQ$_{NSV}$). This approach needs further research, for example by using a Lagrangian approach where the photosynthetic activity of the same population is followed during the day. Until a reliable correction method has been established, a monitoring program including photophysiology should account for diurnal variability, for instance by using only measurements collected in a certain timeframe. Despite the limitations of GPP estimates by variable fluorescence, our results clearly show large spatial

15  variability in gross primary production that is not explained by diurnal variability. This spatial heterogeneity is not fully captured by sampling at the standard low-resolution monitoring stations, showing the added value of our approach. , 2016; Behrenfeld et al., 2002). This makes interpreting spatial patterns difficult as temporal and spatial patterns occur simultaneously, yet, spatial patterns were generally more prominent than the diurnal oscillations.

20  Primary productivity is an important but difficult to estimate parameter. Its importance is evident, being at the base of the marine food web. In recent decades primary productivity in the North Sea seem to decline, with implications for the ecosystem structure and fisheries productivity. Capuzzo et al. (2017) and Cloern et al. (2014) see a global declining trend in primary production measurements. This is worrying as marine ecosystems face many changes and possible threats caused by global warming and increased use of marine resources by man. Remote sensing methods and models

25  are used to estimate primary productivity, but despite improvements in satellite capabilities and ocean colour analyses, the current global annual NPP estimates are uncertain (Silsbe et al., 2016). One of the reasons is that for satellite estimates or modelling purposes variation in phytoplankton community composition or physiology are usually not included. Primary productivity is then estimated solely based on abiotic factors in combination with Chl a estimates (Cole and Cloern, 1987; Behrenfeld and Falkowski, 1997; Westberry et al., 2008; Westberry and Behrenfeld, 2013),

30  although some models include P$^B_{max}$ as parameters, which is parameterized from temperature only. Yet, Chl a and abiotic conditions alone are limited predictors of biological processes, because the Carbon:Chl a ratio is not only dependent on abiotic conditions but also to species-specific phenotypic plasticity needed to acclimate to those abiotic conditions (Flynn, 1991, 2005; Geider et al., 1997; Alvarez-Fernandez and Riegman, 2014) and Chl a is still difficult to

measure in turbid case 2 waters. Therefore, *in vivo* measurements are required to calibrate remote sensing based models while *in vivo* high resolution methods require remote sensing methods to extrapolate over a wider spatial and temporal scale and we suggest that automated production measurements based on FRRf methodology can fulfil this role.

5      Depth integration of high-resolution measurements is a complicated estimate, depending on light penetration through the water column and assuming vertical homogeneity. For most part of the year, the assumption that the mixed layer depth (MLD) reaches below the euphotic zone and causes vertical homogeneity in photoacclimation and community composition, is a safe assumption for the Dutch North Sea, yet short-lived thermal stratification is a regional phenomenon in summer (Van Leeuwen et al., 2015). This short-lived thermal stratification can result in subsurface

10     chlorophyll maximum layers, which, when MLD is shallower than the euphotic zone, will result in a phytoplankton community with distinctly different photophysiological characteristics. Additionally, to calculate water column productivity, an assumption on light penetration through the water column is needed. In this study, light extinction was actively measured approximately ten times per cruise and based on the correlation with turbidity these figures are spatially interpolated using linear regression. Although the light attenuation in the water column is strongly

15     influenced by turbidity, the situation is more complex involving not only underwater processes (absorption and scattering) but also surface processes like reflection and refraction (Brown et al., 1984). Additionally, turbidity is measured in the near-infrared (880 nm), but different substances in the water have characteristically shaped light absorption spectra and photosynthetic active radiation spans a wide range of wavelengths (400-700 nm), this nonlinearity can make the light attenuation coefficient based on turbidity a rough estimate (Kirk, 1994), but as we

20     observed a good correlation between turbidity and $K_d$ ($r^2$=0.77), we assume our $K_d$ estimates are reliable.

The use of automated cluster analyses to interpret spatial heterogeneity is a necessity when dealing with the high amount of data collected by high-resolution methods. The unsupervised Hidden Markov Model (uHMM)**Biogeographic regions**

25     Our GPP rates were based on the same electron requirement for C-fixation ($\Phi_{e,C}$). However, this is an oversimplification as $\Phi_{e,C}$ is known to vary with abiotic conditions (Lawrenz et al., 2013). Therefore, the changes in nutrient conditions and temperature during the growth season are likely to affect GPP. This will be the topic of a future publication and we expect that the detection of several biogeographic regions will help us in predicting $\Phi_{e,C}$. The in this study applied automated cluster methods allowed for identification of distinct phytoplankton communities or biogeographic regions. The spectral clustering

30     method used in this study was originally designed to detect phytoplankton blooms and understanding the involved dynamics, but here used (Rousseeuw et al., 2015; Lefebvre and Poisson-Caillault, in press). In this study this method was applied to identify different phytoplankton communities atand observe spatial patterns. In some months, like April and June, it was indeed possible to identify regions with distinct phytoplankton communities. In other months, such as May, the clustering was not clearly regional scale (Rousseeuw et al., 2015). In general, we see in all months about heterogeneous over the whole

35     Dutch North Sea. A clear distinction between phytoplankton communities of the coastal zone and off-coast regions. A further

separation between the Dutch south coast and the coast off the northern Wadden Islands can usually be made. A separate off-coast area seems to be the southernmost study area, i.e. the northern corner of the Walcheren transect. August is the most heterogeneous month, while both biomass and nutrient concentrations are low, suggesting that niche differentiation is more strongly present than in other months. Broadly, August conditions correspond to the hydrographical regions formerly identified in the Dutch North Sea (Fig. 10; Van Leeuwen et al., 2015; Capuzzo et al., 2015). Howevershore regions could be made in all months, except May. Unfortunately, the model was not able to automatically visualize all spatial heterogeneity.

[Figure]

**Figure 10: hydrographical regions as defined by Van Leeuwen (2015; left) and spatial clusters by uHMM clustering in August (right).**

In April a distinct phytoplankton community was presentFor instance, in April off the coast from Terschelling, only we found a distinct community with high cryptophyte abundance not resulting in a distinctseparate cluster when manually increasing the number of spatial clusters from three to five.. Additionally, temporal variation (i.e. day-night differences) was interfering with the spatial clustering in August. AlthoughSo although such models are useful for visualization and following changes in spatial heterogeneity, input and output need to be critically evaluated before implementation. in monitoring programs. To test whether the differences between months result from seasonal variation or other factors, results over multiple years and additional seasonal cruises need to be made to better characterize heterogeneity of the phytoplankton community structure.

Currently, biological monitoring of phytoplankton in the Dutch North Sea is dictated by the requirements set by OSPAR and the EU Marine Strategy Directive and limited to HPLC analysis of Chl *a* concentration, microscopy counts of *Phaeocystis* cells, and at a few stations, coccolithophores or toxic dinoflagellates. Unfortunately, sampling points were reduced from almost 70 in 1984 to less than 20 today, while strong seasonal patterns, high riverine input, and tidal forces make the Dutch North Sea

a region with high spatiotemporal variability. At the same time, the Dutch North Sea is an area under high anthropogenic pressure, which has led to substantial biogeochemical changes over the past decades (Burson et al., 2016; Capuzzo et al., 2015 and 2017). These abiotic changes affect biology, with potential large implications for ecosystem function and services (Prins et al., 2012; Capuzzo et al., 2017; Burson et al., 2016).

5 **Designing 'smart' phytoplankton monitoring**

A smart monitoring program combines high and low resolution methods in a supplementary way. No method or parameters will offer clear-cut answers, low-resolution nor high-resolution methods alone. Low resolution methods remain a necessity to support the proposed measurements set-up for three reasons: the practical requirement for calibration and blank correction, to retrieve more detailed taxonomical information and to capture the variability in the water column. Firstly, FRRf measurements

10 are affected by interference of colored dissolved matter which can lead to under or overestimation of some parameters (like $F_v/F_m$; Cullen and Davis, 2003). The blank correction is still manual and should be done at least when abiotic conditions change. Secondly, regular measurements of the whole water column remain a necessity to retrieve information on the vertical heterogeneity and the light extinction in the water column. Surface water measurements are only a good reflection of the water column when mixed layer depth is deeper than the euphotic zone. Stratification or mixed layer depth shallower than the

15 euphotic zone can result in subsurface chlorophyll maximum layers and significantly different phytoplankton community (Latasa et al., 2017). Extrapolation of surface measurements to water column estimates is required to assess the carrying capacity of the ecosystem and the contributions to biogeochemical cycles. Only frequent CTD casts equipped with PAR sensor can determine the mixed layer depth and the light extinction in the water column. Thirdly, the level of detail required to identify harmful, keystone or invasive species is only achieved by microscopy analysis. But once identified, flowcytometry is

20 much more suitable for counting the organisms. Another potential combination of high and low resolution methods would be to use high-resolution methods to identify extra sampling points based on real-time projections, opening up early warning methodologies. For example, in the April cruise both Noordwijk 70 and Terschelling 235 km show high gross primary productivity, but in between both high and low productivity rates occur which are not detected with the current sampling program (Fig. 6). The combination of high-resolution *in situ* methods with remote sensing has potential to further increase the

25 spatial and temporal scale. Estimating biological parameters using remote sensing is still difficult, especially in turbid, coastal, case-2 waters (Gohin et al., 2005; van der Woerd et al., 2008). Therefore, *in vivo* measurements are required to calibrate remote sensing based models and we suggest that automated flowcytometry and production measurements based on FRRf methodology can fulfil this role.

Systematic and sufficient monitoring of these changes is of crucial importance to recognize threats, and, once identified as

30 such, develop mitigation actions. The current low resolution monitoring program is clearly not able to cover the entire biological variability. For instance in April, both Noordwijk 70 and Terschelling 235 km show high gross primary productivity, suggesting that production of the entire area between these points is similar, but both high and low productivity rates occur (Fig. 7). Although extra sampling points in clearly deviating areas would be very useful, because only low-resolution offer the level of detail which is required to identify toxic, keystone or invasive species, adding high-resolution methods to the current

monitoring program will already allow for obtaining sensible information between sampling points. A smart monitoring system should use high-resolution methods as it delivers information which is difficult to obtain otherwise, can be used to calibrate and validate remote sensing model and can also be used to identify extra sampling points, possibly even based on real-time projections, opening up early warning methodologies.

**5 Conclusions**

The combination of FRR fluorometry and flowcytometry offers an elaborate view of the phytoplankton community. Accounting for diurnal patterns and identification of FCM clusters for functional types such as nitrogen fixers, calcifiers or DMS-producers are steps needed to increase the value for interpretation ecosystem dynamics and biogeochemical fluxes.

Data interpretation may be supported by automated cluster analyses, such as the uHMM used in the current study, to interpret spatial heterogeneity and to deal with the high amount of data collected by high-resolution methods. However, our model needs to be improved to capture more of the spatial heterogeneity present in ecology of the Dutch North Sea. Overall, the addition of high-resolution monitoring is a very useful supplement to current monitoring to improve.

A good monitoring program monitors the presence of functional types of phytoplankton, including the harmful taxons, the carrying capacity of the ecosystem and changes in biogeochemical cycling. The objective of this study was to evaluate the use of FRR fluorometry and flowcytometry for such monitoring purposes. The four conducted cruises spread over 5 months offered a wide variety of environmental conditions and phytoplankton community states, which the utilized methods were able to visualize. Inclusion of high-resolution methods in monitoring programs allows for analysis of finer scale events. Furthermore, it allows for analysis of living phytoplankton and is thereby able to measure rates and avoid effects of preservation and storage of samples. Another advantage is that high-resolution methods allows for easier comparison between countries, once common protocols have been established. Nevertheless, low resolution methods remain a necessity for more detailed taxonomic analysis, information on vertical heterogeneity, to calibrate and to correct for blanks. Data analysis might be the biggest bottleneck of the implementation of these high-resolution methods. The cluster analysis of flowcytometric data has high 
[revised manuscript text omitted]

|---|---|

---

## Editor Decision (ED1)

Suggestions to authors

**Title**

As suggested by referee 1, the present title is probably not optimal relative to the content of the manuscript. I have seen your answer on this specific point but still suggest that the title could be rephrased to better suit the objective listed at the end of the introduction.

**Abstract**

L14 remove « in time and space »

L15 and 16 (see also L27): Abbreviations ae generally not used in an abstract

L22 « do » should become « does »

L22 Remove the first « Still »

Introduction

P2, L11 I would suggest to modify the order temperature, sea level, acidification

P2, L24 Could be interesting to provide infrmation on the time frequency as well. I know that this is provided later in the manuscript but is nevertheless missing at this level

P2 L30 Could be interesting to discuss of possible consequences associated with the reduction in resolution in the assessment of community composition as well

**Materials and Methods**

P5 L8 « was situated » should become « is situated » or remove « On the RW… »

P5 L23 freezing temperature ??

P5 L26  Unless I am wrong « RWS » has not been defined before

P5 L29 ammonium, calcium, magnesium

P6 Fluorescence Light Curve

P6 L29 Non-Photochemical Quenching

P8 L12 Reaction Centres. What is the signification of H in RCH ?

P9 L7 Ligh Scatter, Sideward Light Scatter

P9 L22-23 « having an angle of inclination of almost 1 ». Not clear. What do you mean : slope ?. If yes yous shoul also consider the intercept (close to 0 ?)

P10 L2-5. May be too qualitative. Precision on the different thresholds could be provided

P10 L18 « scaled » do you mean centered and reduced as is the usual procedure for PCA ?

**Results**

General : Better care should be taken with utilization of the present and the preterite. Generally the preterite is used in results sections. The whole section should be checked an corrected frr that.

General : The sampling took place between April ad May. This is not enough to refer to seasonal changes. I thus recommand that « seasonal » be replaced by « between cruises ». This should be checked and corrected throughout the whole manuscript

P13 L19-24 Unless I am wrong, the ANCOVA has not been presented in the M&M section, which should be corrected. Moreover, readers may not be familiar with this procedure which is aiming at testing for significant differences between regression lines (i.e., in the present ases relationships observed during the 4 sampling months). This could be more clearly stated. Moreover, the wording of the results is not clear as it stands. I would like remind that the procediure is a three step process :

1. Checking for the homogeneity of the residuals between the 4 models (if this condition is not met, then the procedure is not possible)
2. Comparing the slopes (if the slopes are different then the models are different and there is no need/sense to compare the intercepts)
3. Comparing the intercepts if the slopes are not different (if the intercepts are different then the regression models are different although their slopes ae not)

I suggest that the paragraph could be rewriten based on this sequence.

P14 L2-3 Does the first sentence really belongs to A results section ?

P14 L12 Remove « present as »

P14 L12-13 There is a problem of singular and pluraial between the two sentences

P16 L12 Usually sentences do no start with an abbreviation « $E_k$ » should thus become « The $E_k$ » as used afterward

P18-L17 I suggest to simplify the sentence to « Gross pimary productivity ranged from XX in June to XX in the coastal zone during May »

P19 L14 « the identification »

P19 L15 « seen » should be replaced by « considered »

Discussion

General : I suggest to rearrange the discussion by stating at the beginning of each paragraph/section the results from the study that are discussed. This will help to reduce he confusion presently generated by the mixture of general statements and results from the study

General : For each paragraph/section, specific inputs from the study could/should be better put in evidence and their consequences stated more clearly

P22 L3 and 5 : There is a confusion here due to the use of singular (L3) and plural (L5) for method. Maybe one way of avoiding that would be to replace « method » by approach L3

P22 L9 « with and » ??

P22 L10 replace « in both the » by « during both »

P22 L24 « separate » should become « separated »

P23 L5-27 care should be taken to put the a of chlorophyll or chl in italics. Moreover, a clearer conclusion should emerge of this paragraph.

References

General : Please check this whole section for typing mystakes (e.g. p28 L31-35, p29 L2….)

---

## Author Response (AR3)

**Response to interactive comment of anonymous referee #1**

By Hedy M. Aardema in agreement with co-authors.

**Reviewer***: The manuscript by Aardema and co-authors investigates high resolution in in situ measurements of phytoplankton photosynthetic activity and abundance in the Dutch North Sea. The main topic of this study is relevant and provides useful information, particularly when considering monitoring requirements and in defining sampling/monitoring strategies. This study is also a very good example of integrated sampling and outputs from different instruments (i.e. fRRF, flow cytometer, FerryBox).*

**Response**: We really appreciate the elaborate and helpful comments on the manuscript. This detailed and insightful review has allowed us to improve the manuscript considerably.

General comments

**Reviewer:** *The introduction is focused on primary productivity (PP) but the main part of the paper investigates the photophysiological variables and phytoplankton groups with limited mention of productivity. I would suggest emphasizing more the estimates of PP throughout the ms.*

**Response**: Although the primary productivity is a very interesting parameter to calculate, the aim of the paper is to give a broader view of the phytoplankton community. Therefore, we shortened the part on primary productivity in the introduction, but did give it more attention in the results and discussion sections.

**Reviewer:** *Collinearity between variables: flow cytometer (FCM) phytoplankton groups were considered in the analysis even if showing collinearity (VIF>6). Statistical principles should be applied consistently across the analysis and to all the variables. If not, this should be explained clearly.*

**Response**: This is a good point. We re-ran the PCA and spatial clustering and excluded variabiles with the VIF>6. The Multiple Linear Regression was removed from the manuscript, because of the lack of information derived from it together with the abundance of literature already addressing the predictors of primary productivity.

**Reviewer:** *Spatial autocorrelation: transect data with high frequency sampling is likely to be spatially autocorrelated – has this been considered? If spatial autocorrelation is not considered to be a problem in this dataset, please explain why. Alternatively, presence of spatial autocorrelation could be investigated with the use of variograms.*

**Response**: As the reviewer expected, most parameters were spatially autocorrelated. We tested the spatial autocorrelation with Moran's I. This is indeed a problem for the multiple linear regression, but as mentioned previously, we removed this analysis from the manuscript. For the spectral classification clustering and PCA analysis, spatial parameters (latitude, longitude) were not included in the analysis. Without time and space in the calculation we only consider features of the data, so spatial autocorrelation does not influence the results (Demsar et al., 2013, Rousseeuw et al., 2015). Because the similarity between neighbouring points is of interest, we plotted of the spectral clusters on maps to visualize the spatial heterogeneity present.

**Reviewer:** *Diurnal changes in some of the photophysiological variables: the authors clearly show that the diurnal cycle affect the clustering of observations (e.g. Page 25), so the clusters identified were not only based on changes in phytoplankton community but also in sampling activity (i.e. day vs*

*night). As stated in the ms, it is difficult to separate the temporal from spatial variability; however, the effect of spatial variability could be investigated, for example, using measurements collected around specific time of day or night (e.g. 12:00+/4 hours) and rerunning the cluster analysis on this sub-dataset and comparing the outcome with the current clusters. In this way it would also be possible to test the suggestion in line 30-31 (page 27) that spatial patterns are more important than temporal.*

**Response**: We performed the suggested analysis for the month of August by clustering only the measurements that fall into the 12+/-4 h timeframe (see Fig. R1b). In this timeframe the southern coastal zone is distinct from the rest of the Dutch North Sea and corresponds to cluster 10 in the analysis of the complete dataset (Fig. R1a), so this cluster is defined by spatial variability. Cluster 12 and 13 are grouped together in the 12+/-4h timeframe as cluster 1. Cluster 11 is only encountered outside the 12+/-4h timeframe, so is a temporal rather than a spatial cluster. We added this information to the text and added the figure below to the supplementary material.

a).                                                    b).

[Figure]

**Fig. R1: : Maps of clusters as defined by spectral clustering of the whole dataset (left) and only the measurements at 8h around noon (8:00h to 16:00h). Based on the FCM-based five described phytoplankton groups (Table 2) and non-collinear FRRf-parameters on photophysiology ($F_v/F_m$, $1/\tau$, [RCII], $\sigma_{PSII}$, $\alpha$, $E_k$).**

Specific comments

**Reviewer:** *Title – phytoplankton photosynthesis does not provide a clear idea of the content of the paper that covers different photophysiological variables as well as measurements of PP. I would suggest to being more specific.*

**Response**: We prefer to stay with the chosen title. The main purpose of this study was to provide an example of high-resolution methods that could serve in a phytoplankton monitoring program. Based on the results of these methods further calculation can provide an estimate of the PP or can serve in identification of distinct biogeographical regions, of which we gave examples.

**Reviewer:** *Data analysis: it would be useful if the authors could explain why clusters, stepwise regressions and PCA have been used as chosen statistical analysis and what they are you aiming to explain with these techniques?*

**Response**: The main aim of the data analysis was to aid in the interpretation and visualization of the multitude of parameters derived with the high-resolution measurements. The PCA reduces the amount of parameters (or dimensions) and gives an impression on the relationship between parameters. The cluster analysis was chosen to test for spatial heterogeneity; when clusters would contain measurements randomly distributed over the study area, no spatial heterogeneity is present. When clustering shows spatial structure, it is. The stepwise regression was at first used to identify drivers for

primary productivity, but will be removed after realization that the dataset of this study does not add to existing knowledge on this topic.

**Reviewer:** *Data analysis: Biomass vs chl a – repeatedly in the ms the authors refer to 'biomass', as synonymous of chl a (from validate fluorescence). Although chl a is often used as a proxy for phytoplankton biomass, they are not the same and this should clearly be stated at the start of the ms. Confusion arises from figures and tables referring to 'abundance', 'fluorescence', 'chl a', while the text refers to 'biomass'; please check for consistency. In addition, the implications of a variable Chl-a : C ratio should also be considered and discussed. If the main interest is on biomass the authors could consider calculating it from the FCM measurements (for example, see DOI: 10.1016/j.dsr2.2006.05.004).*

**Response**: The authors are aware of this issue and tried to address this problem in the results section '3.2 phytoplankton parameters' where we state: "*Both parameters can yield contrasting results due to the wide range of phytoplankton cell sizes and species-specific Chl a content per cell (Falkowski and Kiefer, 1985; Kruskopf and Flynn, 2005).*" This is repeated in the discussion where we write: „*Chlorophyll a concentration is often used as an estimate for biomass, although the Carbon:Chl a ratio is dependent on abiotic conditions and species-specific phenotypic plasticity (Flynn, 1991, 2005; Geider et al., 1997; Alvarez-Fernandez and Riegman, 2014; Halsey and Jones. 2014)*". So we think we clearly stated this. However, to further improve on this point, the term biomass was deleted in the manuscript. Although this is a very interesting parameter, and we are working on a method to calculate biomass based on scattering measured by the FCM. We already found good agreement between our biovolume and images obtained by the Image in Flow of the FCM (unpublished). However, this relationship seems to be taxon specific, which we want to study more in depth and is beyond the scope of the current study. The method to calculate biomass of Tarran et al. (2006) assumes all cells have a spherical shape and a constant C content per biovolume. Because this is an oversimplification, we prefer to use cell counts and fluorescence in the current paper. We did include our view on biomass calculation from flowcytometer data in the discussion.

**Reviewer:** *UHMM and cluster identification – it is not clear whether the clusters between the different months (Figure 5) are the same or not – in other words, is cluster 1 in April characterized (defined) by the same variables as cluster 1 in May? If not, then it may be better to separate the clusters e.g. with different numbers and/or colours in the figures.*

**Response**: we adjusted the figure as suggested.

**Reviewer:** *Discussion of results: results of the analysis of the photophysiological variables and of PP appear discussed separately. Outcomes from these two parts of the study should be brought (and discussed) together, where possible.*

**Response**: In the result section, primary productivity and Photophysiology are now both under an own header.

**Reviewer:** *Conclusions – I would suggest to highlight the importance of this study for monitoring program. Also, a bit more considerations on combining low and high resolution measurements would be useful.*

**Response**: We rewrote the conclusions accordingly:
"A good monitoring program monitors the presence of nuisance phytoplankton, the carrying capacity of the ecosystem and changes in biogeochemical cycling. The objective of this study was to evaluate the use of FRR fluorometry and flowcytometry for monitoring purposes. The four conducted cruises spread over 5 months offered a wide variety of environmental conditions and phytoplankton community states, which the utilized methods were able to visualize.
Inclusion of high-resolution methods in monitoring programs allows for analysis of finer scale events. Furthermore, it allows for analysis of living phytoplankton and is thereby able to measure rates and avoid effects of preservation and storage of samples. Another advantage is that high-resolution

methods allows for easier comparison between countries, once common protocols have been established. Nevertheless, low resolution methods remain a necessity for more detailed taxonomic analysis, information on vertical heterogeneity, to calibrate and to correct for blanks. Data analysis might be the biggest bottleneck of the implementation of these high-resolution methods. The cluster analysis of flowcytometric data has high potential for improvement to increase the informative value of the method. Especially identification of phytoplankton clusters with a functional quality, such as nitrogen fixers, calcifiers or DMS-producers, would be helpful for interpretation of ecosystem dynamics and biogeochemical fluxes. Regarding the FRRf, the main challenge is converting electron transport rate to gross primary productivity in carbon units. Further research in these topics would benefit implementation of these methods into monitoring protocols. Furthermore, it is important to account for diurnal patterns in monitoring set-up to be able to distinguish between diurnal and spatial variability. Possibly the diurnal variability could be modelled, but more studies with a Langragian based approach would be needed for a better understanding of the impact of diurnal variability in the data. Overall, the in this study presented high-resolution measurement set-up has large potential to improve phytoplankton monitoring in supplement to existing low-resolution monitoring programs."

**Reviewer:** *Supplementary information – need to be linked (and referred to) in the main text of the ms, otherwise it may be difficult for the reader to know that this info is available.*

**Response**: Done.

**Technical corrections**

**Reviewer:** *Page 1: 23-26 – rewording is needed*

**Response:** Rephrased to: "One of the major concerns when using these methods for monitoring purposes is the presence of a diurnal cycle concurrent to the spatial variation, especially in photophysiological parameters. This concurrent presence of spatial and temporal patterns needs to be taken into account when designing a monitoring program. Nevertheless, the richness of additional information provided by high-resolution methods, such as the FCM and FRRf, can supplement low-resolution monitoring to attain a better understanding of the phytoplankton community."

**Reviewer:** *Page 1 30 -keywords, consider adding primary productivity*
**Response:** Added.

**Reviewer:** *Page 2: 10-12 – this sentence would fit better at the start of the paragraph. It also requires references*
**Response:** Moved to beginning of the paragraph.

**Reviewer:** Page 3: 5 – 'a sum': consider replacing with 'a combination'
**Response:** Done.

**Reviewer:** Page 3: 23 – 'pigment ratio' slightly incorrect as the ratio considered is of fluorescence
**Response:** Agreed and adopted.

**Reviewer:** Page 3: 24-25 – Aims – this statement about key driver of PP is very general and can be misinterpreted as the ms focuses on only 4 months during the growing season of a particular year. Time frame of this study should be specified
**Response:** reformulated

**Reviewer:** Page 4: 3-5 – not clear, needs rewording
**Response:** Rephrased to: "The Dutch North Sea is a shallow tidal shelf sea in the southern part of the North Sea. The main water flow is Northward flowing Atlantic water that enters the North Sea in the

south through the Channel. The Atlantic water flowing around Scotland enters the North Sea and meets the Channel water and the freshwater from the rivers forming the Frisian Front."

**Reviewer:** Page 5: 1- would be useful to have the exact dates of the surveys.
**Response:** Added.

**Reviewer:** Page 5: 6 – more details on the temporal frequency indicated as 'low resolution' should be provided (e.g. how many samples per station? How many a day? How many depths?)
**Response:** Added.

**Reviewer:** Page 5: 27-32 – please provide more details of the methods or a published reference (for people not being able to access the internal protocols).
**Response:** Added.

**Reviewer:** Page 6: 16 & 18 – acronyms (e.g. NPQ and F0') should be explained when used the first time
**Response:** Added.

**Reviewer:** *Page 8: 12-13 – formula 8 is missing*
**Response:** It was removed. We changed formula 9 to formula 8.

**Reviewer:** *Page 8: 17 – need rewording*
**Response:** Rephrased as: "Volumetric $P_{max}$ and α were derived by fitting $JV_{PII}$ in µmol photons $m^{-3}$ $h^{-1}$ to equation 1 (the exponential model of Webb et al., 1974) and used to integrate productivity over depth. The light availability in the water column was estimated as […] with E(z) being the irradiance at depth z, $E_{surface}$ the incoming surface irradiance and $K_d$ the light extinction coefficient."

**Reviewer***: Page 8: 20-21 – it is not clear how surface irradiance was calculated; please reword this section*
**Response:** We adjusted the text to the following explanation: "To avoid effects of changing incident surface irradiance ($E_{surface}$) on the spatial pattern and to be able to compare GPP between regions we used monthly average surface irradiances ($E_{surface}$) in our calculations of primary productivity. From 2010-2016 irradiance (400-700 nm) was measured at the roof of the NIOZ building in Yerseke using a LI-190 quantum PAR sensor and hourly averages stored using a LI1000 datalogger. $E_{surface}$ was then calculated by averaging all irradiance data from the years 2010-2016 for the respective month."

**Reviewer:** *Page 9: 17 – was the clustering carried out by the FCM software or was it done by expert judgment manually? Also, was data cleaned from potential presence of air bubbles etc? Please provide details on these points,*
**Response:** The chosen cluster criteria were based on expert judgement. The clustering was done by the software Easyclus 1.26 (ThomasRuttenProjects) according to these criteria. Noise, air bubbles and other potential outliers were removed after the clustering.

**Reviewer***: Page 10: 2 – outliers –specify which analysis you are referring to (e.g. outliers from the fRRF?)*
**Response:** All data, rephrased in manuscript.

**Reviewer:** *Page 10: 5 – provide a reference for the value of 0.65*
**Response:** Added; Kolber, Z. and P. G. Falkowski. 1993. Use of active fluorescence to estimate phytoplankton photosynthesis in situ. Limnology and Oceanography. 38:1646-1665.

**Reviewer:** Page 10: 12 – please specify which are the photophysiological variables considered

**Response:** We added the following sentences to the data analysis section: "Phytoplankton parameters were first tested for collinearity and predictors with a variance inflation factor (VIF) over 6 were removed (Zuur et al., 2009). This left for the cluster analysis FCM-parameters Pico-red, Nano-red, Micro-red and *Synechococcus* and the FRRf-parameters $\sigma_{PSII}$, $F_v/F_m$, $a_{LHII}$, $1/\tau$, $E_k$."

**Reviewer:** *Page 10: 13 – acronyms (VIF) should be defined here*

**Response:** Added.

**Reviewer:** *Page 11: 20 – 'nitrate': should this be 'DIN'?*

**Response:** Yes.

**Reviewer***: Page 11: 27-28 – please explain the evidence for P and Si-limitation (i.e. discuss the ratios vs expected limiting ratios in literature). Also, please specify the value of Redfield Ratio and reference.*

**Response:** We removed the nutrient ratios from the results. The paper only reports the nutrient values as additional background information to understand phytoplankton dynamics. A detailed analysis of concentration vs ratio is past the subject of this paper, but in the discussion nutrient limitation is now discussed.

**Reviewer:** *Table 3 legend – 'not completely comparable': this expression doesn't have a clear statistical meaning. Please specify briefly in the legend which month had a different sampling route and station so for the reader to understand in which month the study area is not fully covered.*

**Response:** True. We removed the term 'not completely comparable' from the legend and added a short explanation of the differences between months. Also, we moved the table to the supplementary information and replaced it with the nutrient concentration table.

**Reviewer:** *Figure 2 provide equations of linear regressions with R2 and significance*

**Response:** The $R^2$ and significance are now added to the legend. The linear regressions are irrelevant because the unit of the x-axis is in relative fluorescence units (RFU) and instruments will require separate calibration.

**Reviewer***: Page 14: 27 – 'suggesting physiological stress', please provide reference*

**Response:** Suggett et al., 2009.

**Reviewer:** *Page 16: 9 – it is not clear to which phytoplankton group the % are referring to.*

**Response:** The nanophytoplankton. Rephrased.

**Reviewer:** *Page 16: 14 – please specify which are 'these regions'*

**Response:** Rephrased.

**Reviewer:** *Page 16: 15-16 – this paragraph should be moved to the discussion so to allow the concept to be developed further.*
*Page 16: 17 – please explain why low sigmaPSII may reflect Rhine River waters.*

**Response:** Moved to discussion.

**Reviewer:** *Page 17 – Figure 4 – I appreciate the different scaling was necessary to 'visualize the spatial heterogeneity' however it makes very hard the comparison between figures. In fact, the reader needs to keep checking the legend, which is printed in very small characters difficult to see. I would suggest reconsidering the use of a uniform scale (at least for some of the variables, if possible).*

**Response:** We adjusted the figure to a uniform scaling.

**Reviewer**: *Page 18: 17 – there is limited or no comments on the results of some of the photophysiological variables such as alpha, Pmax, effective absorption cross section.*
**Response:** We expanded the result section on the Photophysiology.

**Reviewer:** *Page 18: 25 – 'sake of completeness'. See general comment about collinearity, please explain why statistical principle of VIF>6 was not applied consistently to all variables*
**Response:** We agree that this might not have been the best choice, we preferred to include all the phytoplankton groups. As mentioned before, we now deleted the collinear variables with VIF>6.

**Reviewer:** *Page 18: 28-29 – table should be provided (for example in the additional info) showing the contribution of each variable to the PC1 and PC2 for the 4 months, and total variance explained.*
**Response:** We added this table to the manuscript, in combination with figure 6:

| | April | | May | | June | | August | |
|---|---|---|---|---|---|---|---|---|
| | PC1 | PC2 | PC1 | PC2 | PC1 | PC2 | PC1 | PC2 |
| Sigma | 0.8 | **28.8** | 0.1 | **36.7** | 0.0 | 9.3 | 12.1 | 9.9 |
| $F_v/F_m$ | 13.7 | 0.6 | 0.8 | **14.5** | 27.6 | 0.1 | 0.0 | **17.5** |
| $a_{LHII}$ | **18.7** | 3.4 | 17.5 | 6.7 | **20.9** | 8.7 | **21.0** | 3.3 |
| [RCII] | **17.1** | 6.6 | **20.4** | 1.6 | **28.0** | 2.6 | **25.8** | 0.0 |
| $1/\tau$ | 9.8 | **22.7** | 4.4 | 7.5 | 0.4 | 1.7 | 0.2 | **20.6** |
| $E_k$ | 3.9 | 13.8 | 0.7 | **26.3** | 3.7 | 3.9 | 0.7 | **16.8** |
| Pico-red | 4.2 | **15.1** | **18.5** | 0.4 | 6.1 | **26.9** | 0.3 | 11.8 |
| Nano-red | **16.9** | 0.0 | **21.1** | 0.6 | 2.9 | **16.9** | 15.3 | 3.1 |
| Micro-red | 10.5 | 4.5 | 16.4 | 1.4 | 6.3 | 2.9 | **22.9** | 0.4 |
| *Synechococcus* | 4.3 | 4.4 | 0.0 | 4.3 | 4.2 | **27.0** | 1.8 | 16.7 |
| Variance explained | 45.6 % | 19.3 % | 42.5 % | 18.9 % | 29.1 % | 18.7 % | 33.9 % | 25.7 % |

**Reviewer:** *Page 19: 1 – alpha is defined as Light utilisation efficiency (Table 1) but then in the text is referred to as 'affinity'. please check for consistency.*
**Response:** Changed in table. The value for alpha is the slope of the FLC, and is a measure for photosynthetic affinity for incoming light.

**Reviewer:** *Page 21: 8-13 – consider whether to move this text in additional info (or to remove it?). It breaks the flow of the results and the addition of clusters 'manually' appears to not be meaningful and/or significant (as it doesn't adopt the same statistical robust principle).*
**Response:** It is true that it does not adopt the same statistical robust principle. However, there is spatial heterogeneity in the flowcytomer data, that are not visualized with the UHMM and this is what we wanted to explore. We do agree that the manual increase of amount of clusters might not the best way to go forward with this, so we deleted this section from the manuscript.

**Reviewer:** *Page 22: 6 – 'abiotic' and 'salinity' misspelled. Page 22: 9 – as for previous PCA, please provide variables used and information on their contribution towards variance explained.*
**Response:** this paragraph and figure were removed from the manuscript because the PCA does not provide useful insights or new information on the phytoplankton community or Dutch North Sea.

**Reviewer**: *Page 23: 6-7 – this paragraph is not clear particularly what is meant with 'opposite'*
**Response:** rephrased.

Figure 7 legend – Size of the open circles is a bit confusing and misleading as the reader may assume the size of the bubble refers to the amount of PP. Consider simplifying the figures and only plot productivity

**Response:** The figure was simplified as suggested.

**Reviewer:** *Page 24: 15 – please indicate how much of the variability in PP is explained by the stepwise regression (e.g. R2?).*

**Response:** because information on the nutrient availability was only available on a low-resolution spatial scale, the information provided by high resolution methods are not effectively used. To study the drivers of primary productivity another study design should have been chosen. Therefore, this analysis was deleted from the manuscript.

**Reviewer:** Page 25: 4 – reword please.

**Response:** rephrased

**Reviewer:** Page 26: 2-5 – require rewording particularly the need to clarify and be more specific on the work done in this study.

**Response:** removed from manuscript.

**Reviewer:** Page 26: 5 – this sentence may be misleading. The authors calculated PP along the sampling transects but did not provide an estimate for the wider Dutch North Sea as it may appear here.

**Response:** removed from manuscript.

**Reviewer:** *Page 26: 8 & 11 – timing of the bloom is discussed in this section however it would not be possible to define the start of the bloom based on a 4-day sampling per month. Continuous observations throughout the year by an instrument buoy or remote sensing would allow to 'contextualise' the measurements within the growing season (i.e. determine when sampling was carried out within the phytoplankton growing season).*

**Response:** Agreed and removed from manuscript.

**Reviewer:** *Page 26: 24-25 – please reword*

**Response:** rephrased

**Reviewer:** *Page 27: 8-9 – repetition of method; should be deleted.*

**Response:** Rephrased.

**Reviewer:** *Page 29: Figure 10 legend, possibly just my issue, I don't see the similarity between the two figures.*

**Response:** We do see a basic similarity, with the separation between the different water masses being reflected in our results. However, the similarity might not be striking enough to include the figure and therefore we leave it out of the manuscript.

**Reviewer:** *Page 30: 13 – 'low resolution': should this be 'high-resolution'?*

**Response:** no, we meant to say low-resolution. We rephrased to make it easier to follow: "Extra low-resolution sampling points in clearly deviating areas would be useful, because only low-resolution offer the level of detail which is required to identify toxic, keystone or invasive species."

**References**

Demšar, U., Harris, P., Brunsdon, C., Fotheringham, A. S., & McLoone, S. (2013). Principal Component Analysis on Spatial Data: An Overview. Annals of the Association of American Geographers, 103(1), 106–128. https://doi.org/10.1080/00045608.2012.689236

Rousseeuw, K., Poison Caillault, E., Lefebvre, A., & Hamad, D. (2015). Achimer Hybrid hidden Markov model for marine environment monitoring. IEEE JOURNAL OF SELECTED TOPICS IN APPLIED EARTH OBSERVATIONS AND REMOTE SENSING, 8(1), 204–213. https://doi.org/http://dx.doi.org/10.1109/JSTARS.2014.2341219

Suggett, D. J., C. M. Moore, A. E. Hickman, and R. J. Geider. (2009b). Interpretation of fast repetition rate (FRR) fluorescence: signatures of phytoplankton community structure versus physiological state. Marine Ecology-Progress Series **376**:1-19.

**Response to interactive comment of anonymous referee #2**

By Hedy M. Aardema in agreement with co-authors.

**Reviewer:** *This paper analyses spatial and temporal patterns in cruise data with 3 high-resolution monitoring methods: FRRF, Flow-cytometry and Ferrybox. Correlations between the observed variables are also analysed. The large dataset, including many phytoplankton and environmental variables observed together enables the authors to understand the patterns in the various phytoplankton variables. The results could guide the optimal application of such novel monitoring methods in operational monitoring for a.o. MSFD.*

**Response**: We thank the reviewer for the helpful and critical comments. We rewrote and restructured the manuscript extensively based on these comments. We are happy to hear that the reviewer sees the potential of our applied methods.

General comments

**Reviewer:** *The paper lacks a clearly stated research question or hypothesis to be tested. Therefore, it is unclear what is the purpose of the various analyses performed and what we can learn from the results. Based on the conclusion that this type of "high-resolution is a very useful supplement to current monitoring", I would expect a hypothesis such as "combined high-resolution monitoring of many phytoplankton variables along with environmental variables allows us to quantify seasonal and meso-scale patterns in phytoplankton biomass, species composition and primary production. The concurrent measurement of different phytoplankton variables allows us to understand the effect of phytoplankton species composition and physiological adaptation processes on the observed patterns in phytoplankton biomass and production". Then the analysis should show how the variables should be combined to provide the most reliable estimates of phytoplankton biomass and primary production.*

**Response**: Because of the exploratory nature of our research, a hypothesis was not defined. The addition of the suggested sentences does help in making the manuscript easier to follow. We therefore adopted part of the sentences and added of the following sentences to the introduction: "The aim of this study is to test the suitability of these two high-resolution methods to be developed as novel phytoplankton monitoring method. The two high-resolution methods, a flowcytometer and a FRR fluorometer, were deployed concurrently on four 4-day cruises in April, May, June and August to

meet a wide range of environmental conditions and phytoplankton community states. These measurements allow for quantification of seasonal and mesoscale spatial patterns in phytoplankton abundance, photophysiology and gross primary production. In this paper we provide an overview of the acquired results, use a spectral cluster analysis to visualize spatial heterogeneity and evaluate the potential of these methods to optimize current monitoring programs."

**Reviewer:** *There are many observed variables, which are not consistently named in the text, figures and tables. Therefore, it is easy to get lost in the description of patterns for all individual variables. A clear definition of variables that is consistently used throughout the text would help the reader to understand the storyline. Some of the variables observed by the FRRF seem to be very similar. Which of the variables should be used as indicator and which are redundant to answer the research questions?*

**Response**: We corrected the inconsistent naming. The variables of the FRRf might seem similar under some conditions. However, because these variables vary depending on community composition and environmental conditions, they might deviate when conditions change (Sugget et al., 2009; Kromkamp and Forster, 2003). Therefore, care must be taken into choosing the parameters. For the current study the main interest is on monitoring the phytoplankton community, therefore we are interested in parameters that are informative on physiological adaptation or characteristic for phytoplankton taxons. Additionally, we focus on high resolution measurements, so limit the parameters to the ones attainable at high-resolution. Based on these considerations we decided to include the current parameters, which give us a broad overview of the photophysiological status of the phytoplankton community.

**Reviewer:** *In the conclusions section a recommendation on next steps would be much appreciated: what would be required to use the high-resolution methods in scope to provide reliable estimates of phytoplankton biomass, production and species composition for long term monitoring? In the introduction and conclusion the species composition is defined in functional types such as nitrogen fixers, calcifiers or DMS-producers, but these do not correspond to the phytoplankton clusters used in this paper.*

**Response**: the conclusions were rewritten:
"A good monitoring program monitors the presence of functional types of phytoplankton, including the harmful taxons, the carrying capacity of the ecosystem and changes in biogeochemical cycling. The objective of this study was to evaluate the use of FRR fluorometry and flowcytometry for such monitoring purposes. The four conducted cruises spread over 5 months offered a wide variety of environmental conditions and phytoplankton community states, which the utilized methods were able to visualize. Inclusion of high-resolution methods in monitoring programs allows for analysis of finer scale events. Furthermore, it allows for analysis of living phytoplankton and is thereby able to measure rates and avoid effects of preservation and storage of samples. Another advantage is that high-resolution methods allows for easier comparison between countries, once common protocols have been established. Nevertheless, low resolution methods remain a necessity for more detailed taxonomic analysis, information on vertical heterogeneity, to calibrate and to correct for blanks. Data analysis might be the biggest bottleneck of the implementation of these high-resolution methods. The cluster analysis of flowcytometric data has high potential for improvement to increase the informative value of the method. Especially identification of phytoplankton clusters with a functional quality, such as nitrogen fixers, calcifiers or DMS-producers, would be helpful for interpretation of ecosystem dynamics and biogeochemical fluxes. Regarding the FRRf, the main challenge is converting electron transport rate to gross primary productivity in carbon units. Further research in these topics would benefit implementation of these methods into monitoring protocols. Furthermore, it is important to account for diurnal patterns in monitoring set-up to be able to distinguish between diurnal and spatial variability. Possibly the diurnal variability could be modelled, but more studies with a Langragian based approach would be needed for a better understanding of the impact of diurnal variability in the data. Overall, the in this study presented high-resolution measurement set-up has large potential to improve phytoplankton monitoring in supplement to existing low-resolution monitoring programs."

Specific comments

**Reviewer:** *Sentences are often long: consider breaking up in multiple sentences to improve readability.*

**Response**: We apologize for the difficulties and hope to have improved the readability in the new manuscript.

**Reviewer:** *Figure 1: please show only the stations (with names/ abbreviations) used in this study (see table S1) and the areas used in the text (such as Dogger Bank, Wadden, Den Helder, Rhine outflow) so the description of spatial patterns can also be understood by people that are not Dutch.*

**Response**: We updated the figure to the following:

[Figure]

**Reviewer:** *Section 2.2: please refer to international protocols/methods rather than internal protocols.*
**Response**: We added a more detailed description to the method section.

**Reviewer:** Table 1: it would help to have an additional column stating the interpretation / meaning of this variable, such as total biomass, nutrient stress, maximum growth rate, efficiency of light uptake etc. Then later in the text you can use these 'meaningful' names instead of codes, to facilitate understanding of observed patterns. Also a figure illustrating the meaning of the different variables (alfa,Ek, F', Fm' etc.) could prevent getting lost in all abbreviations.

**Response**: Unfortunately, the meaning of the different variables is usually not straightforward and dependent on multiple predictors (species, nutrient concentration, light availability etc.; Suggett et al., 2009). Nonetheless, we tried to make the table more information easier to understand.

**Reviewer:** *Equation 9: why did you use monthly averaged irradiance if you are looking at high-resolution patterns. Why did you not use irradiances measured during the cruise?*

**Response**: Unfortunately, we were unable to collect reliable irradiance data for all cruises. Clearly, it is preferable to have irradiance (PAR) continuously measured in parallel to the FRRF measurements when aiming to monitor current primary productivity.

**Reviewer:** *Table 2: Since you use both Length_FWS and O/R ratio as criteria to distinguish the phytoplankton groups, it would be logical to include a column for O/R ratio with the applied criteria.*

**Response**: Good idea, we added the O/R-ratio to the table.

**Reviewer:** *It is not entirely clear whether pico-red includes pico-Synecho or not. On page 14, line 30 it says: "Both groups of picophytoplankton (Synechococcus and total)", whereas table 2 and figure 3 suggest the two groups are exclusive.*

**Response**: Pico-red and Pico-synecho are two different groups, as correctly understood from table 2 and figure 3. We rephrased the sentence to: "Both groups of picophytoplankton (Synechococcus and Pico-red)", and scanned the manuscript for other mixing up.

**Reviewer:** *Section 2.4: please state with every type of analysis what is the purpose / research questions for that analysis. For example: what are you trying to predict from what and why?*

**Response**: We added the following the sentences to section 2.4: "To find regions with similar phytoplankton communities, data was spectrally clustered using the uHMM R package (Poisson-caillault and Ternynck, 2016) in the statistical software R (version 3.4.1, R Core Team, 2017)." and "Principal Component Analyses (PCA) were performed to find which variables contributed most to the cluster results."

**Reviewer:** *Section 3.1: I don't see the value of comparing averages over whole transects (with large spatial variability, which is the subject of this paper), that are not even the same, between months. The only thing you see is seasonal patterns that are well-known from other studies and that can be summarized in section 2.1 in a description of the study area. Most of this section describes the data in table S1. I would replace table 3 with table S1 and remove table S2. N/P ratios address that same question as table S1, but with an indicator that is controversial.*

**Response**: For the authors the table helped to visualize the seasonal patterns, but we agree on the comment that this table does not add to the already existing knowledge on seasonal patterns. We therefore adopted the suggestion to replace the table with the table S1 from the supplementary material.

**Reviewer:** *The text in this section (and subsequent sections) is sometimes hard to follow as it is not clearly structured in time and space and variable. We go back and forth in time. Section 3.2 describes first figure 2, then figure 3 and then again figure 2 and then figure 3. I suggest to make one section about phytoplankton biomass (figure 2) and then one section on species composition (figure 3).*

**Response**: Rewritten

**Reviewer:** *Page 16, line 14: the southern coastal stations are more strongly affected by the Rhine outflow than the Scheldt outflows (see for example: Lacroix, G., Ruddick, K., Ozer, J., & Lancelot, C. (2004). Modelling the impact of the Scheldt and Rhine/Meuse plumes on the salinity distribution in Belgian waters (southern North Sea). Journal of Sea Research, 52(3), 149-163.).*

**Response**: We reformulated to Rhine and Scheldt river outflow.

**Reviewer:** *Figure 4: Please use consistent legends for the same variable between different months, with the same colour scheme and symbols (squares vs. circles) and with blue indicating low values and red indicating high values, so the high values stand out, more than the low values. Also captions in the table per line (red fluorescence, O/R ratio etc.) and per column (april, may etc.) would help to easier understand the figure.*

**Response**: We remade the figures, see manuscript.

**Reviewer:** *Section 3.5: I don't see the added value of this analysis. What does it tell us?*

**Response**: Agreed. We aimed to get a better understanding of the drivers of primary productivity in the Dutch North Sea. However, we realize now that the dataset is not very well suited for this and we therefore removed the analysis.

**Reviewer:** *Page 24: I suggest to mention in the table all the variables that were included in the analysis and note coefficients or 'ns' for not significant and the p values per explanatory variable. Then readers don't need to reconstruct the overview from the text. Actually, the significance test is likely not valid due to strong spatial autocorrelation in the data.*

**Response**: The Multiple Linear Regression was removed from the manuscript, because of the lack of information derived from it together with the abundance of literature already addressing the predictors of primary productivity.

**Reviewer:** *Discussion: Here I would expect to get some advice: How to best estimate phytoplankton biomass from these data? Should we use total red fluorescence (best R2) or F0 (least affected by NPQ)? Is there a way to combine both (with other available variables) to get an even better estimate?*

**Response**: We added the following paragraph to the discussion:
"Biomass might be one of the most important parameters to understand phytoplankton dynamics, but its direct measurement is not possible using high-resolution methods. Chlorophyll a concentration is often used as an estimate for biomass, although the Carbon:Chl a ratio is dependent on abiotic conditions and species-specific phenotypic plasticity (Flynn, 1991, 2005; Geider et al., 1997; Alvarez-Fernandez and Riegman, 2014; Halsey and Jones, 2015). Red fluorescence gave a good estimate of chlorophyll a concentration, both using the FRRf (adjusted $R^2$= 0.66) and FCM (adjusted $R^2$=0.90). Both the FRRf and the flowcytometer estimate the chlorophyll a concentration based upon the fluorescence in the red spectrum after excitation in the blue spectrum. There are some slight differences in the optics, the FRRf excites with a 450 nm LED and measures the fluorescence at $682 \pm 30$ nm, while the FCM excites at 488 nm and filters the red fluorescence over a longpass 650 nm filter towards the red fluorescence detector. The smaller detection range of the FRRf detector is optimized around the maximum emission of PSII and limits contamination by PSI (Franck et al., 2002; Oxborough et al., 2012). The second difference is the fluorescent state of the photosystems, the strong laser of the flowcytometer can only measure the maximum fluorescence (Fm), which is a parameter more prone to quenching than the minimum fluorescence measured by the FRRf. Yet, the biggest difference concerns the method; where the flowcytometer measures the fluorescence per particle, the FRRf does only a bulk measurement. In a bulk measurement other particles in solution scatter the excitation and emission photons, plus the emitted fluorescence of the phytoplankton is subject to reabsorption, especially at higher biomass densities. The latter seems to have the most impact on chlorophyll a concentrations, as the fit of the flowcytometer derived red fluorescence is a better than the FRRf minimum fluorescence. Other studies that use the FCM to estimate chlorophyll a concentrations also showed good relationships, but find better fits using the bulk measurements using a fluorimeter (Thyssen et al., 2015; Marrec et al., 2018). The conversion to biomass may also be done from cell abundances. Some studies use the oversimplified assumption that all cells have a spherical shape and a constant C content per biovolume (Tarran et al., 2006). With the scanning flowcytometer it is also possible to estimate biovolume based on scattering properties of the cell, but this relationship appears to be taxon specific (Rijkeboer, pers. comm.). This relationship will be further explored by comparing the calculated biovolume based on the Image in Flow pictures and the flowcytometric properties of these phytoplankters."

**Reviewer:** *Can we trust GPP from FRRF as a reliable estimate of primary production or is more work needed to achieve that goal? If so, what needs to be done?*

**Response**: We added the following paragraph to the discussion:
"The reliability of variable fluorescence as estimate of gross primary productivity is depending on many cell processes from the photon absorbance to carbon assimilation. The variable fluorescence reflects the first step of photosynthesis; the efficiency of which photons are captured and electrons

produced and transferred. However, to interpret gross primary productivity in an ecological or biogeochemical meaningful way, the FRR units of electrons per unit time need to be converted to carbon units. Gross photosynthesis correlates well with photosynthetic oxygen evolution (Suggett et al., 2003), and multiple studies have shown good correlation between 14C-derived estimates of primary productivity and FRRf-derived estimates using a constant conversion factor (Melrose et al., 2006; Kromkamp et al., 2008). However, in reality this parameter is not a constant, as along the pathway from electron to carbon atom electrons are consumed by other cell processes (Flameling and Kromkamp, 1998; Halsey and Jones, 2015; Schuback et al., 2016). Therefore, a reliable GPP estimate in carbon units from FRR fluorometry requires more research and estimates provide relative rather than qualitative values. Despite its limitations the fact that the method can measure in situ, with relatively little phytoplankton manipulation before measurement, makes the method promising. Calibration with other methods, such as concurrent C14 of C13 incubations, could help to better understand the processes from electron excitation to carbon fixation. However, it should be recognized that these types of measurements come with their own problems, and measure something in between net and gross primary productivity depending on the incubation time and growth rate of the phytoplankton (Halsey and Jones, 2015). So it remains a question which method is measuring the 'real' primary productivity. Attempts to calculate primary productivity from flowcytometer data have also been made, which is actually based on the diurnal cycle in cell size caused by cell division (Marrec et al., 2018). Despite the limitations of GPP estimates by variable fluorescence, our results clearly show large spatial variability in gross primary production concurrent to the expected strong variability during the growth season. This spatial heterogeneity is not fully captured by sampling at the standard low-resolution monitoring stations, showing the added value of our approach. Primary productivity was highest in April, and relatively large values were also observed offshore, indicating that a low phytoplankton biomass does not necessarily means that primary production is low. Our GPP rates were based on the same electron requirement for C-fixation ($\Phi e, C$). However, this is a likely oversimplification as $\Phi e, C$ is known to vary with abiotic conditions (Lawrenz et al., 2013) and the changes in nutrient conditions and temperature during the growth season are likely to affect GPP. This will be the topic of a future publication and we expect that the detection of several biogeographic regions will help us in predicting $\Phi e, C$."

**Reviewer:** *It is not really clear whether the diurnal variability in the FRRF variables is a problem that needs to be solved.*

**Response**: It is not so much a problem that needs to be solved, but it does need to be taken into account when setting up a monitoring program including FRRF variables. It is important to realize that measurements taken at different times of the day, might not be comparable. To be able to include FRRF variables in a long-term monitoring program, the included sampling points should be sampled at the same time of the day.

**Reviewer:** *Are the clusters in the FCM analysis the relevant ones to provide 'useful' information to science & society? Should we / Can we move on to other clusters that are mentioned in the conclusions?*

We added the following text to the discussion:
"To understand the role of the phytoplankton in biogeochemical cyckes, the FCM clusters would ideally reflect taxonomic or functional groups, as calcifiers, silicifiers, DMS producers (such as Phaeocystis) and nitrogen fixers (le Quéré et al., 2005). The lack of identification of distinct clusters makes this sofar impossible. Other studies manually separate up to 10 phytoplankton groups with the same instrument (Marrec et al., 2018). These groups included Prochlorococcus, which is at the absolute limit of resolving capacity of the FCM because of their small size and low fluorescence. They furthermore distinghuished the Pico-red in three groups based on FLO/FLR-ratio. Nano-cryptophytes group in high and low orange fluorescence and included a micro-eukaryotes group with a size from 10 to 20 uM. But these groups are still made up of many taxonomic genera and, apart from size, won't allow much for further interpretation of their role in the ecosystem or biogeochemical cycles. The same accounts for detection of nuisance phytoplankton; distinct clusters of toxic phytoplankton species are lacking. Although this will remain a challenge because toxicity in

phytoplankton can differ within morphotypes and sometimes even differ per strain within a species (Tillman and Rick, 2003). But potentially, further research in flowcytometry can result in suspicious clusters to be flagged and further inspected by a specialist using microscopy. The potential is certainly there, as much of the information retrieved by the FCM is still unexplored; the clustering is performed on totals (area under the peak) instead of the pulse-shape. This in combination with more advanced camera options will need to further distinguish between groups in the future."

**Reviewer***: Do the FCM data help to better understand the FRRF data (and vice versa)? For example, do we see diatoms under light limited conditions (high F'/Fm', high alfa, low Ek) and picoplankton under nutrient limited conditions (low F'/Fm')? Other ecological niches that we know from literature? Different conditions promoting Synechococcus compared to other picoplankton?*

**Response**: We tried to incorporate the link between the methods better, we added the following sentences to the manuscript:
"In this study a large part of the Dutch North Sea shifted from nutrient sufficiency to nutrient limitation between April and May, which was reflected in the low efficiency of PSII (Fv/Fm; Fig. 4). The Fv/Fm recovered between May and June, which suggest that the phytoplankton adapted to nutrient limiting conditions (Kruskopf and Flynn, 2005). However, photophysiological parameters are also varying per taxonomic group; smaller taxa typically have lower Fv/Fm values and higher σPSII values (Kolber et al., 1988; Suggett et al., 2009b). Indeed, by flowcytometry we find that the biggest shift in community composition took place between May and June from a nanophytoplankton dominated community to a picophytoplankton dominated community. These findings demonstrate how flowcytometry and fast repetition rate fluorometry can supplementary improve ecosystem understanding."

Technical comments

**Reviewer:**
- Collinear should be spelled with 2 ll's throughout the whole text.
- Page 9, line 4 & 5: I guess um means micrometers?
- Page 18, line 4: middle-right, please refer to the label C4 a-x.
- Figure 5: The figure would be easier to read if the colours per group are consistent between the cluster analysis on the right and the map on the left. Labels (A-D for April to August panels) would also help.
- Figure 9: Please add the hours of the day on the x-axis.
- Page 25, line 3: the word influenced is repeated too many times and therefor should get an e in the end.
- Page 28, line 11: estimates are.
- Line 13: parameter without s.
- Page 31, line 8: Jerico-next, without h.

**Response**: We adopted the suggested technical improvements.

**Abstract**

**Editor:** L14 remove « in time and space »

L15 and 16 (see also L27): Abbreviations ae generally not used in an abstract

L22 « do » should become « does »

L22 Remove the first « Still »

**Response**: Corrected

**Introduction**

**Editor:** P2, L11 I would suggest to modify the order temperature, sea level, acidification

**Response**: Corrected

**Editor:** P2, L24 Could be interesting to provide information on the time frequency as well. I know that this is provided later in the manuscript but is neverthelees missing at this level

**Response**: Included << Monitoring cruises take samples in the Dutch North Sea between March and October every two or four weeks. >>

**Editor:** P2 L30 Could be interesting to discuss of possible consequences associated with the reduction in resolution in the assessment of community composition as well

**Response**: Added to the introduction: << Underway measurements are not able to replace some more detailed low-resolution measurements, but their higher spatial and temporal resolutions provide the possibility to identify short-lived events, detect small-scale changes in phytoplankton dynamics, evaluate consequences of possible (spatial) undersampling, and act as an early warning system. >>

**Materials and Methods**

**Editor:** P5 L8 « was situated » should become « is situated » or remove « On the RW… »

**Response**: Removed << On the RV Zirfaea >> and added <<of the underway system >>

**Editor:** P5 L23 freezing temperature??

**Response**: -18°C. Added to text.

**Editor:** P5 L26 Unless I am wrong « RWS » has not been defined before

**Response**: Rijkswaterstaat. Added to text.

**Editor:** P5 L29 ammonium, calcium, magnesium

P6 Fluorescence Light Curve

P6 L29 Non-Photochemical Quenching

**Response**: corrected

**Editor:** P8 L12 Reaction Centres. What is the signification of H in RCH ?

**Response**: The H is actually two II's. The two II's refer to photosystem II, which was added to the text: << The amount of reaction centres of PSII per cubic metre ([RCII]) was calculated as >>

**Editor:** P9 L7 Light Scatter, Sideward Light Scatter

**Response**: corrected

**Editor:** P9 L22-23 « having an angle of inclination of almost 1 ». Not clear. What do you mean : slope ?. If yes you s should also consider the intercept (close to 0 ?)

**Response**: Replaced by:

<< The acceleration of the particles in the sheath fluid positions the cells along their long axis, which allows for size estimation based on the FWS pulse shape. A linear relation was found between Length FWS and measured length of diverse phytoplankton species (Length FWS = 0.92*Measured length – 1.57; $R^2$=0.98; Rijkeboer, 2018). Size estimation is limited by the width of the laser beam (5 µm) so estimations of cell sizes smaller than 5 µm is not possible based on the FWS. >>

**Editor:** P10 L2-5. May be too qualitative. Precision on the different thresholds could be provided

**Response**: Inlcuded into Methods following description:

<< For the FRRf data, quality control of the FLC fits was done based on the quality ratio of the induction curve fit per FLC light step and the $r^2$ of the FLC fit. The quality ratio of the induction curve fit was calculated as the ratio of Fv or Fv' to the standard error (SE) of the linear regression of the saturation phase. FLC fits with an $r^2$ < 0.75, or with over 30% of the data points with a quality ratio below 6, were visually inspected and removed based on expert judgement. This led to removal of 1% to 7% of the FLC fits per month>>

**Editor:** P10 L18 « scaled » do you mean centered and reduced as is the usual prdure for PCA ?

**Response**: Not completely. Scaling does include centering, but also standardizes the deviation from the center to equal units, so that relative difference from the mean, rather than absolute deviatons are used for the PCA.

Results

**Editor:** *General: Better care should be taken with utilization of the present and the preterite. Generally, the preterite is used in results sections. The whole section should be checked an corrected for that.*

**Response**: corrected.

**Editor:** *General: The sampling took place between April ad May. This is not enough to refer to seasonal changes. I thus recommend that « seasonal » be replaced by « between cruises ». This should be checked and corrected throughout the whole manuscript*

**Response**: Corrected.

**Editor:** *Unless I am wrong, the ANCOVA has not been presented in the M&M section, which should be corrected. Moreover, readers may not be familiar with this procedure which is aiming at testing for significant differences between regression lines (i.e., in the present ases relationships observed during the 4 sampling months). This could be more clearly stated. Moreover, the wording of the results is not clear as it stands. I would like remind that the procedure is a three step process :*

*1. Checking for the homogeneity of the residuals between the 4 models (if this condition is not met, then the procedure is not possible)*

*2. Comparing the slopes (if the slopes are different then the models are different and there is no need/sense to compare the intercepts)*

*3. Comparing the intercepts if the slopes are not different (if the intercepts are different then the regression models are different although their slopes are not)*

*I suggest that the paragraph could be rewritten based on this sequence.*

**Response**:

Added to the m&m:

<< To test whether environmental conditions (as present in the different months) had a significant effect on fluorescence as predictor for Chl *a* concentration, an ANCOVA was performed with the month as a factorial predictor.>>

In the methods section replaced the respective chapter by:

<< Before the ANCOVA analysis, natural logarithm transformations were required to correct for inhomogeneity of the residuals and unequal variances between months. Both the FRRf $F_0$ ($p<0.01$, adjusted $R^2=0.66$) and FCM total red fluorescence ($p<0.01$, adjusted $R^2=0.90$) provided significant predictors of HPLC-derived Chl *a* concentration (Fig. 2). The ANCOVA with the FRRf-derived $F_0$ as Chl *a* predictor revealed that the slope did not differ per month, but the intercept did ($p<0.01$). The ANCOVA with the FCM-derived TFLR as Chl *a* predictor resulted not only in a significant difference of the Chl *a* concentration per month ($p<0.01$) but also in a significantly different slope ($p<0.05$), suggesting that other predictors that differ per month were influencing the fluorescence per Chl *a* molecule (Fig. 2). >>

**Editor:** P14 L2-3 Does the first sentence really belongs to A results section ?.

**Response**: Moved paragraph to discussion.

**Editor:** P14 L12 Remove « present as »

**Response**: Corrected.

**Editor:** P14 L12-13 There is a problem of singular and plurial between the two sentences

**Response**: These sentences were removed from the method section as response to an earlier comment.

**Editor:** P16 L12 Usually sentences do no start with an abbreviation « Ek » should thus become « The Ek » as used afterward

**Response**: Included 'The'

**Editor:** P18-L17 I suggest to simplify the sentence to « Gross pimary productivity ranged from XX in June to XX in the coastal zone during May »

**Response**: Agreed and corrected.

**Editor:** P19 L14 « the identification »

**Response**: Corrected.

**Editor:** P19 L15 « seen » should be replaced by « considered »

**Response**: Corrected.

**Discussion**

**Editor:** *General: I suggest to rearrange the discussion by stating at the beginning of each paragraph/section the results from the study that are discussed. This will help to reduce he confusion presently generated by the mixture of general statements and results from the study*

**Response:** Rearranged discussion.

**Editor:** *General: For each paragraph/section, specific inputs from the study could/should be better put in evidence and their consequences stated more clearly*

**Response:** Rewritten discussion.

**Editor:** *P22 L3 and 5 : There is a confusion here due to the use of singular (L3) and plural (L5) for method. Maybe one way of avoiding that would be to replace « method » by approach L3*

**Response:**

**Editor:** P22 L9 « with and » ??

**Response:** Checked, but can't find.

**Editor:** P22 L10 replace « in both the » by « during both »

**Response:** Checked, but can't find.

**Editor:** P23 L5-27 care should be taken to put the a of chlorophyll or chl in italics. Moreover, a clearer conclusion should emerge of this paragraph.

**Response:** Corrected the italics of Chl *a*. Added in the paragraph: << The lower sensitivity to environmental conditions implies that the FRRf is better suited to estimate chlorophyll *a* concentration in comparison to the FCM..>> and added to the end of the paragraph: << As long as there is no uncontroversial method to derive phytoplankton biomass, calculation of multiple parameters and critical evaluation remains necessary.>>

**References**

**Editor:** General : Please check this whole section for typing mystakes (e.g. p28 L31-35, p29 L2….)

**Response:** Checked, but can't find.

[revised manuscript text omitted]

A pairplot analysis of the combined data of all cruises shows that some photosynthetic parameters are highly correlated (Supplementary material). The correlation of alpha and $F_v/F_m$, indicators for photosynthetic affinitiy and photosynthetic efficiency, are perfectly correlated (r=1). The parameters derived from the PE-curve also show high correlation, being dependent on the light acclimation state of the phytoplankton trends in the maximum electron transport rate ($P_{max}$) and the light saturation parameter ($E_k$) are similar. Surprisingly, alpha does not show any correlation with Ek, suggesting that the light affinity is not dependent on the level of irradiance where the PSII reaction centres become saturated, or that its value is also affected by nutrient limitation, obscuring a relationship with $E_k$. The effective absorption cross section per photosystem, $\sigma_{PSII}$, is very strongly negatively related to $n_{PSII}$ (r=-0.9). This is to be expected; the larger $n_{PSII}$, the smaller the number of pigment molecules associated with it.

 **3.3 Phytoplankton community composition**

In April the northern  $_v$$_m$  part of the Dutch North Sea $_v$$_m$ ~~recovers and community composition changes. Both groups of picophytoplankton (*Synechococcus* and total) increase in relative abundance between May and June, while the nanophytoplankton shows a strong decrease (Fig. 3). Because the picophytoplankton fraction makes up for only a small part of the biomass, the microphytoplankton is the largest contributor to red fluorescence in June, although this group does not increase in relative abundance in comparison to May. After June the microplankton disappears, leaving 80% of the average community composition to picoplankton. The shift to smaller cell sizes at nutrient limiting conditions is not surprising because of the higher nutrient affinity of smaller cells.~~

[revised manuscript text omitted]
 variability introduced by phytoplankton in different biogeochemical areas. However, as we observed this diurnal variability also in the same clusters on a number of occasions, it is clear that diurnal variability is inherent in our analysis. For future studies it is advised to include Langragian based approach where the same phytoplankton community can be followed during a complete light-dark cycle. The diurnal trend in coupling of electron flux and carbon fixation is dictated by cell cycle, a circadian oscillator and irradiance, and photophysiological plasticity minimizes photodamage and optimizes growth under fluctuating light and nutrient concentrations (Claquin et al., 2014; Cohen and Golden, 2015; Schuback et al., 2016). To interpret spatial variability separately from temporal variability and to provide a reliable estimate of gross primary productivity, Schuback et al. (2016) suggest a correction with NPQ$_{NSV}$, which needs further research in order to get

a more reliable GPP value. The presence of non-photochemical quenching (NPQ) makes the interpretation of most photophysiological parameters complicated in our study because the lack of dark acclimation time decreases the comparability between samples. Most photophysiological parameters we measured showed diurnal trends, although, but as said, this is likely not only due to NPQ but also to phytoplankton cell cycle (Claquin et al., 2014; Schuback et al., 2016) and rhythms driven by a circadian oscillator (Cohen and Golden, 2015). But although the presence of NPQ compromises the use of fluorescence as estimate for chlorophyll concentrations, the good relationship between the HPLC-derived Chl a concentration and fluorescence of the FRRf and FCM suggest that most of the NPQ is dissipated during the time in the tubing and low light acclimation. The clear diurnal trends we observed are in agreement with previous studies and is usually explained by photophysiological plasticity to minimize photodamage (Schuback et al., 2016; Behrenfeld et al., 2002). This makes interpreting spatial patterns difficult as temporal and spatial patterns occur simultaneously, yet, spatial patterns were generally more prominent than the diurnal oscillations.

Primary productivity is an important but difficult to estimate parameter. Its importance is evident, being at the base of the marine food web. In recent decades primary productivity in the North Sea seem to decline, with implications for the ecosystem structure and fisheries productivity. Capuzzo et al. (2017) and Cloern et al. (2014) see a global declining trend in primary production measurements. This is worrying as marine ecosystems face many changes and possible threats caused by global warming and increased use of marine resources by man. Remote sensing methods and models are used to estimate primary productivity, but despite improvements in satellite capabilities and ocean colour analyses, the current global annual NPP estimates are uncertain (Silsbe et al., 2016). One of the reasons is that for satellite estimates or modelling purposes variation in phytoplankton community composition or physiology are usually not included. Primary productivity is then estimated solely based on abiotic factors in combination with Chl $a$ estimates (Cole and Cloern, 1987; Behrenfeld and Falkowski, 1997; Westberry et al., 2008; Westberry and Behrenfeld, 2013), although some models include $P^B_{max}$ as parameters, which is parameterized from temperature only. Yet, Chl $a$ and abiotic conditions alone are limited predictors of biological processes, because the Carbon:Chl $a$ ratio is not only dependent on abiotic conditions but also to species-specific phenotypic plasticity needed to acclimate to those abiotic conditions (Flynn, 1991, 2005; Geider et al., 1997; Alvarez-Fernandez and Riegman, 2014) and Chl $a$ is still difficult to measure in turbid case-2 waters. Therefore, $in\ vivo$ measurements are required to calibrate remote sensing based models while $in\ vivo$ high resolution methods require remote sensing methods to extrapolate over a wider spatial and temporal scale 
[revised manuscript text omitted]

| $F_q^{(t)}/F_m^{(t)}$ | Fluorescence parameter providing an estimate of PSII efficiency under ambient light (under ambient light) | Dimensionless |
| $F_v/F_m$ | Quantum efficiency of PSII | Dimensionless |
| $\sigma_{PSII}$ | Absorption cross section of PSII photochemistry | $nm^2 \, PSII^{-1}$ |
| [RCII] | Concentration of functional RCII | $nmol \, RCII \, m^{-3}$ |
| $a_{LHII}$ | Absorption coefficient of PSII light harvesting | $m^{-1}$ |
| α | Light utilisation efficiency | $\mu mol \, electrons \, (\mu mol \, photons)^{-1}$ |
| $E_k$ | Minimum saturating irradiance of fluorescence light curve | $\mu mol \, photons \, m^{-2} \, s^{-1}$ |
| $P_{max}$ | Maximum photosynthetic electron transport rate | $\mu mol \, electrons \, m^{-2} \, s^{-1}$ |
| $JV_{PII}$ | PSII flux per unit volume | $mol \, electrons \, (PSII \, m^{-3}) \, d^{-1}$ |
| GPP | Gross Primary Productivity | $mg \, C \, m^{-2} \, h^{-1}$ |
| $n_{PSII}$ | Number of [RCII] per mole Chl $a$ | $mol \, RCII \, mol^{-1} \, chla$ |
| 1/τ | Rate of re-opening of a closed RCII with an empty $Q_B$ site | $ms^{-1}$ |
| $K_a$ | Instrument type-specific constant allowing for direct calculation of [RCII] and $JV_{PII}$ from FRR data | $m^{-1}$ |

Table 2: The phytoplankton groups distinguished in the current study.

| | Length FWS | Main corresponding taxonomic group |
| --- | --- | --- |
| Pico-Red | <4 µm* | Pico-eukaryotes |
| Pico-Synecho | <4 µm* | e.g. Synechococcus |
| Nano-Crypto | 4-20 µm | Cryptophycea |
| Nano-Red | 4-20 µm | Diatoms, Haptophytes |
| Micro-Red | >20 µm | Diatoms, Haptophytes |

*In june <6 µm

~~Table 3: Monthly averages ± SD of abiotic conditions and biological parameters. Due to differences in sampling route and stations, the monthly averages are not completely comparable. Large standard deviations are due to spatial heterogeneity, for a more detailed description of the spatial heterogeneity; see figure 4 and the supplementary material. $P_{max}$ and alpha are based on relative electron transport rates.~~

| | April | May | June | August |
|---|---|---|---|---|
|  | | | | |
|  |  |  |  |  |
|  |  |  |  |  |
|  |  |  |  |  |
|  |  |  |  |  |
|  |  |  |  |  |
|  |  |  |  |  |
|  |  |  |  |  |
|  |  |  |  |  |
|  |  |  |  |  |
|  |  |  |  |  |
|  | | | | |
|  |  |  |  |  |
|  |  |  |  |  |
|  |  |  |  |  |
|  |  |  |  |  |
|  |  |  |  |  |
|  |  |  |  |  |
|  |  |  |  |  |
|  |  |  |  |  |
|  |  |  |  |  |
|  |  |  |  |  |
|  |  |  |  |  |
|  |  |  |  |  |
|  |  |  |  |  |
|  |  |  |  |  |
|  |  |  |  |  |

Table 4: Coefficients of the stepwise multiple linear regression (n=61) for ln(GPP) with p<0.05 and VIF<6

|  | coefficients |
|---|---|
| Intercept | 5.613 |
| alpha | 2.916 |
| Turbidity | $-9.929 \times 10^{-2}$ |
| DIN | $-3.567 \times 10^{-2}$ |
| Temperature | $-1.887 \times 10^{-2}$ |
| Total red fluorescence (biomass) | $2.833 \times 10^{-9}$ |
| Hours | $4.141 \times 10^{-2}$ |